# ACTIVATION STEERING WITH A FEEDBACK CONTROLLER

**Dung V. Nguyen**[*]
Department of Mathematics
National University of Singapore
dungnv@u.nus.edu

**Nhi Y. Pham**[*]
Center for AI Research
VinUniversity, Hanoi, Vietnam
nhi.py2@vinuni.edu.vn

**Hieu M. Vu**[*]
Torilab
vmhieu17@gmail.com

**Lei Zhang**[†]
Department of Mathematics
National University of Singapore
matzhlei@nus.edu.sg

**Tan M. Nguyen**[†]
Department of Mathematics
National University of Singapore
tanmn@nus.edu.sg

## ABSTRACT

Controlling the behaviors of large language models (LLM) is fundamental to their safety alignment and reliable deployment. However, existing steering methods are primarily driven by empirical insights and lack theoretical performance guarantees. In this work, we develop a control-theoretic foundation for activation steering by showing that popular steering methods correspond to the proportional (P) controllers, with the steering vector serving as the feedback signal. Building on this finding, we propose Proportional-Integral-Derivative (PID) Steering, a principled framework that leverages the full PID controller for activation steering in LLMs. The proportional (P) term aligns activations with target semantic directions, the integral (I) term accumulates errors to enforce persistent corrections across layers, and the derivative (D) term mitigates overshoot by counteracting rapid activation changes. This closed-loop design yields interpretable error dynamics and connects activation steering to classical stability guarantees in control theory. Moreover, PID Steering is lightweight, modular, and readily integrates with state-of-the-art steering methods. Extensive experiments across multiple LLM families and benchmarks demonstrate that PID Steering consistently outperforms existing approaches, achieving more robust and reliable behavioral control. The code is publicly available at: https://github.com/dungnvnus/pid-steering.

## 1 INTRODUCTION

Large language models (LLMs) have demonstrated remarkable capabilities across diverse domains, yet ensuring that their outputs align with desired behaviors remains a central challenge (Houlsby et al., 2019; Sclar et al., 2023; Kotha et al., 2024; Luo et al., 2025). Common post-training approaches (Wei et al., 2022; Ouyang et al., 2022) have proven effective for improving alignment. However, these techniques demand substantial computational resources (Houlsby et al., 2019) and require weight updates with new training data, which can unintentionally degrade fluency or performance on unrelated tasks (Kotha et al., 2024; Templeton et al., 2024; Luo et al., 2025).

An increasingly popular alternative is *activation steering*, which modifies a model's internal activations directly at inference time, avoiding costly retraining (Turner et al., 2024a;b; Li et al., 2023; Rimsky et al., 2024; Lee et al., 2025; Teo et al., 2025; Rodriguez et al., 2025; Vu & Nguyen, 2025). This approach has been employed both to probe internal representations (Geiger et al., 2024; von Rütte et al.,

---

[*]Co-first authors. [†]Co-last authors. Correspondence to: dungnv@u.nus.edu & tanmn@nus.edu.sg

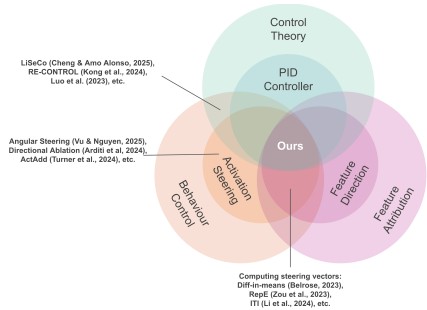
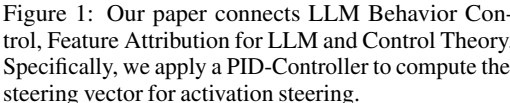

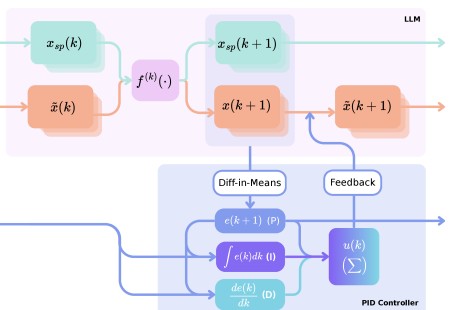

Figure 1: Our paper connects LLM Behavior Control, Feature Attribution for LLM and Control Theory. Specifically, we apply a PID-Controller to compute the steering vector for activation steering.

Figure 2: PID Steering: To compute the steering vector $u(k)$: a PID controller is applied at every layer $f^{(k)}(\cdot)$, using the diff-in-means between 2 contrastive data $x_{sp}(k)$ and $x(k)$ as the error signal $e(k)$.

2024; Vu & Nguyen, 2025; Teo et al., 2025) and to enable fine-grained behavioral control (Zou et al., 2023a; Li et al., 2023; Rimsky et al., 2024; Turner et al., 2024b; Vu & Nguyen, 2025; Rodriguez et al., 2025). Recent work demonstrates that steering along carefully chosen low-dimensional directions can effectively alter model behavior (Zou et al., 2023a; Turner et al., 2024b; Rimsky et al., 2024; Arditi et al., 2024; Vu & Nguyen, 2025), highlighting its potential as a lightweight yet powerful alignment strategy.

**Steering through the Lens of Dynamical Systems.** Recent methods leverage the geometric structure of the activation space (Marks & Tegmark, 2024; Park et al., 2024) using linear algebraic techniques (Zou et al., 2023a; Turner et al., 2024b; Rimsky et al., 2024; Arditi et al., 2024; Vu & Nguyen, 2025) to compute the steering vectors. While effective, these works oversimplify the complex, dynamic behavior arising from the auto-regressive nature of LLMs. When viewed through this dynamical lens, activation steering can be interpreted as guiding the model's trajectory through activation space, from a region encoding one concept to another, analogous to steering a dynamical system from one state to a desired target state.

**Contribution.** Building on the aforementioned dynamical system insight, our work departs from the prevailing algebraic framing and instead adopts a control-theoretic perspective on activation steering. Although recent studies (Luo et al., 2023; Soatto et al., 2023; Kong et al., 2024) have begun exploring this direction, their focus has primarily remained at the level of the token-level generation process, treating high-level behaviors as control signals. In contrast, we take into account the internal mechanisms of LLMs by modeling the layer-wise construction of feature directions (Bricken et al., 2023; Park et al., 2024) as a dynamical system. These feature directions are then used as steering vectors (Zou et al., 2023a; Turner et al., 2024b; Rimsky et al., 2024; Arditi et al., 2024; Vu & Nguyen, 2025). Specifically, we show that existing steering methods relying on difference-of-means feature directions (Rimsky et al., 2024), including Activation Addition (ActAdd) (Turner et al., 2024b), Directional Ablation (Arditi et al., 2024), and Mean Activation Transport (Mean-AcT) (Rodriguez et al., 2025), can be interpreted as instances of a *proportional (P) controller*, thus suffering from the steady-state error due to the disturbance to the state of the system (Åström & Hägglund, 1995b). This new perspective enables the application of principled control-theoretic strategies for extracting effective feature directions and computing steering vectors, thereby offering stronger robustness and performance guarantees for activation steering methods. An overview of our approach is shown in Fig. 1 and 2. In this paper, we use the terms feature direction and steering vector interchangeably, noting that steering vectors represent a practical application of feature directions in activation steering. Our contribution is three-fold:

1. **Control-Theoretic Formulation for Feature Direction:** We develop a new control-theoretic framework for constructing feature directions/steering vectors along the layers of an LLM.

2. **PID-Based Steering:** We propose the novel *Proportional-Integral-Derivative (PID) Steering*, a control-theoretic framework for computing feature directions using a PID controller to reduce the steady-state error inherent in existing activation steering methods (see Fig. 2).

3. **Unified Theoretical Framework:** We demonstrate that common activation steering methods correspond to proportional (P) controllers. This connection enables a theoretical analysis that highlights PID Steering's advantages in reducing steady-state error and oscillations

We comprehensively validate our PID Steering across diverse *modalities* (text and image), *downstream applications* (toxicity mitigation, jailbreaking attack, and image style control), *steering paradigms* (ActAdd, Mean-AcT, and Angular Steering (Vu & Nguyen, 2025)), *model families* (Qwen2.5 (Yang et al., 2024), Gemma2 (Gemma Team et al., 2024), Llama3 (Dubey et al., 2024), SDXL-Lightning (Lin et al., 2024), and Flux (Labs, 2024)), and *model scales* (3B-14B for language models and 3.5B-12B for diffusion models).

**Organization.** We organize the paper as follows: Section 2 reviews background; Section 3 links activation steering to P control and introduces PID Steering; Section 4 presents its theoretical analysis; Section 5 provides empirical validation; Appendix A discusses related work; and Section 6 concludes. Proofs, derivations, and additional experiments are in the Appendix.

## 2 BACKGROUND

### 2.1 TRANSFORMERS

Decoder-only transformers (Vaswani et al., 2017) take an input token sequence $\boldsymbol{q} = [q_1,...,q_n]$ and map it to initial embeddings $\boldsymbol{x}(1) = [\boldsymbol{x}_1(1),...,\boldsymbol{x}_n(1)]^\top = \text{Embed}(\boldsymbol{q})$. The embeddings are then propagated through $K$ layers. At each layer $k$, the residual activation $\boldsymbol{x}_i(k)$ for token $p_i$ is updated by self-attention and an MLP block, with normalization applied before (and sometimes after) these modules:

$$\boldsymbol{x}_{i,\text{post-attn}}(k) = \boldsymbol{x}_i(k) + \text{SelfAttn}^{(k)}(\text{Norm}(\boldsymbol{x}_i(k)))$$

$$\boldsymbol{x}_i(k+1) = \boldsymbol{x}_{i,\text{post-attn}}(k) + \text{MLP}^{(k)}(\text{Norm}(\boldsymbol{x}_{i,\text{post-attn}}(k))).$$

In this paper, for notational brevity, we summarize the layered processing above as $\boldsymbol{x}_i(k+1) = f_i^{(k)}(\boldsymbol{x}(k))$, $i = 1,...,n$, where $f_i^{(k)}$ encapsulates both the Self-Attention mechanism and Multi-Layer Perceptron at layer $k$. Finally, the output activations from the last layer, $\boldsymbol{x}_i(L+1)$, are decoded over the model's vocabulary to get the next token $y_i = \text{Decode}(\boldsymbol{x}_i(L+1))$ for subsequent generation.

### 2.2 ACTIVATION STEERING

Features such as behaviors or concepts are hypothesized to align with (approximately) orthogonal directions in activation space (Elhage et al., 2022; Park et al., 2024; Bereska & Gavves, 2024). Activation steering leverages this by modifying hidden states at inference to amplify or suppress specific features (Konen et al., 2024; Li et al., 2023; Marks et al., 2025; Templeton et al., 2024; Bayat et al., 2025). Recent approaches operationalize this by constructing feature directions, which act as *steering vectors* $\boldsymbol{r}$ for adjusting hidden states. These steering vectors are computed as layerwise differences in mean activations between datasets with contrasting concepts (e.g., harmful vs. harmless), a *difference-in-means* approach (Rimsky et al., 2024), shown to effectively isolate salient feature directions (Turner et al., 2024a;b; Arditi et al., 2024).

#### 2.2.1 APPLYING THE STEERING VECTORS

Two popular activation steering methods using steering vectors are: *Activation Addition* (Turner et al., 2024b), and *Directional Ablation* (Arditi et al., 2024). Both modify the token activation $\boldsymbol{x}(k)$ using the steering vector $\boldsymbol{r}(k)$ at layer $k$ such that the activation expresses the target concept or behavior. By setting $\boldsymbol{x}(1, \boldsymbol{q}) = \text{Embed}(\boldsymbol{q})$ and $\boldsymbol{r}(1) = 0$, these methods apply the steering vectors $\boldsymbol{r}(k)$ to the activation $\boldsymbol{x}(k)$, $k = [K]$, at each layer via a steering function $\rho_{\text{steer}}$ as follows:

$$\boldsymbol{x}(k-1, \boldsymbol{q}) = \rho_{\text{steer}}(\boldsymbol{x}(k-1, \boldsymbol{q}), \boldsymbol{r}(k-1)), \text{for } \boldsymbol{q} \in \mathcal{D}_{\text{source}} \tag{1}$$

$$\boldsymbol{x}(k, \boldsymbol{q}) = f^{(k)}(\boldsymbol{x}(k-1, \boldsymbol{q})), \text{for } \boldsymbol{q} \in \mathcal{D}_{\text{source}} \cup \mathcal{D}_{\text{target}}. \tag{2}$$

We discuss here the details on how to design the steering function $\rho_{\text{steer}}$ for each method.

**Activation Addition (ActAdd).** ActAdd and sets $\rho_{\text{steer}}(\boldsymbol{x}(k), \boldsymbol{r}(k)) = \boldsymbol{x}(k) + \alpha \boldsymbol{r}(k)$, where the coefficient $\alpha$ controls the strength of the effect.

**Directional Ablation (DirAblate).** DirAblate removes the feature by projecting the token activation onto the orthogonal complement, $\rho_{\text{steer}}(\boldsymbol{x}(k), \boldsymbol{r}(k)) = \boldsymbol{x}(k) - \boldsymbol{r}(k)\boldsymbol{r}(k)^\top \boldsymbol{x}(k)$.

#### 2.2.2 COMPUTING THE STEERING VECTORS

**Non-sequential Mapping.** Let us use the jailbreaking task as an example. In this task, we apply activation steering to force the LLM to respond to harmful prompts (Arditi et al., 2024; Vu & Nguyen, 2025). In order to compute the steering vectors, i.e., refusal direction, for each layer $k \in [K]$ and

post-instruction token position $i \in I$, we calculate the mean activation $\boldsymbol{\mu}_{i,\text{target}}(k)$ for harmless prompts from $\mathcal{D}_{\text{target}}^{(\text{train})}$ and $\boldsymbol{\mu}_{i,\text{source}}(k)$ for harmful prompts from $\mathcal{D}_{\text{source}}^{(\text{train})}$:

$$\boldsymbol{\mu}_{i,\text{target}}(k) = \frac{1}{|\mathcal{D}_{\text{target}}^{(\text{train})}|} \sum_{\boldsymbol{q} \in \mathcal{D}_{\text{target}}^{(\text{train})}} \boldsymbol{x}_i(k,\boldsymbol{q}), \qquad \boldsymbol{\mu}_{i,\text{source}}(k) = \frac{1}{|\mathcal{D}_{\text{source}}^{(\text{train})}|} \sum_{\boldsymbol{q} \in \mathcal{D}_{\text{source}}^{(\text{train})}} \boldsymbol{x}_i(k,\boldsymbol{q}). \tag{3}$$

We then compute the difference-in-means vectors, $\boldsymbol{r}_i(k) = \boldsymbol{\mu}_{i,\text{target}}(k) - \boldsymbol{\mu}_{i,\text{source}}(k)$, and use them as steering vectors. Optionally, among the difference-in-means vector $\boldsymbol{r}_i(k)$ for each post-instruction token position $i \in I$ at layer $k$, we can select the single most effective vector $\boldsymbol{r}(k) = \text{Select}(\{\boldsymbol{r}_i(k)\}_{i \in I})$ from this set by evaluating each candidate vector over validation sets $\mathcal{D}_{\text{source}}^{(\text{val})}$ and $\mathcal{D}_{\text{target}}^{(\text{val})}$.

**Sequential Mapping.** A non-sequential mapping neglects the causal dependency across activations, where outputs from one layer are passed to the next, i.e., $\boldsymbol{x}_i(k+1) = f_i^{(k)}(\boldsymbol{x}(k))$. Consequently, any intervention applied at one layer must be accounted for before introducing an intervention at the subsequent layer. To capture this causal structure, *Mean Activation Transport (Mean-AcT)* in (Rodriguez et al., 2025) estimates the steering vectors incrementally at each layer as follows:

$$\boldsymbol{x}_i(k-1,\boldsymbol{q}) = \rho_{\text{steer}}(\boldsymbol{x}_i(k-1,\boldsymbol{q}), \boldsymbol{r}(k-1)), \text{for } \boldsymbol{q} \in \mathcal{D}_{\text{source}} \tag{4}$$

$$\boldsymbol{x}_i(k,\boldsymbol{q}) = f_i^{(k)}(\boldsymbol{x}(k-1,\boldsymbol{q})), \text{for } \boldsymbol{q} \in \mathcal{D}_{\text{source}} \cup \mathcal{D}_{\text{target}} \tag{5}$$

$$\boldsymbol{\mu}_{\text{target}}(k) = \frac{1}{|\mathcal{D}_{\text{target}}^{(\text{train})}|} \sum_{i \in I, \boldsymbol{q} \in \mathcal{D}_{\text{target}}^{(\text{train})}} \boldsymbol{x}_i(k,\boldsymbol{q}), \qquad \boldsymbol{\mu}_{\text{source}}(k) = \frac{1}{|\mathcal{D}_{\text{source}}^{(\text{train})}|} \sum_{i \in I, \boldsymbol{q} \in \mathcal{D}_{\text{source}}^{(\text{train})}} \boldsymbol{x}_i(k,\boldsymbol{q})$$

$$\boldsymbol{r}(k) = \boldsymbol{\mu}_{\text{target}}(k) - \boldsymbol{\mu}_{\text{source}}(k). \tag{6}$$

Like ActAdd, Mean-AcT sets $\rho_{\text{steer}}(\boldsymbol{x}(k), \boldsymbol{r}(k)) = \boldsymbol{x}(k) + \alpha \boldsymbol{r}(k)$.

## 2.3 PROPORTIONAL–INTEGRAL–DERIVATIVE CONTROLLER

Proportional-Integral-Derivative (PID) control is a feedback mechanism extensively used in control systems (Minorsky, 1922). It is valued for its simplicity, robustness, and effectiveness in a broad range of applications, from industrial automation to robotics and aerospace systems (Visioli, 2006; Borase et al., 2021; Nguyen et al., 2024). The core idea behind PID control is to compute a control signal based on the error between a target reference signal and the actual output of a system. Specifically, consider a continuous-time dynamical system governed by a state space model

$$\dot{\boldsymbol{x}}(t) = g(\boldsymbol{x}(t), \boldsymbol{u}(t), t), \boldsymbol{y}(t) = h(\boldsymbol{x}(t), \boldsymbol{u}(t), t), \tag{7}$$

where $\boldsymbol{x}(t) \in \mathbb{R}^d$ denotes the state variable, $\boldsymbol{u}(t) \in \mathbb{R}^m$ is the control variable, and $\boldsymbol{y}(t) \in \mathbb{R}^{d'}$ represents the measured output signal. Here, $g : \mathbb{R}^d \times \mathbb{R}^m \to \mathbb{R}^d$ specifies the system dynamics, and $h : \mathbb{R}^d \times \mathbb{R}^m \to \mathbb{R}^{d'}$ is an output mapping. A PID controller applies the control variable $\boldsymbol{u}(t)$ to minimize the discrepancy between a target reference, or also known as the setpoint in the literature of PID control, $\boldsymbol{y}_{sp}(t)$ and the actual output $\boldsymbol{y}(t)$. This discrepancy, called control error, is defined as

$$\boldsymbol{e}(t) = \boldsymbol{y}_{sp}(t) - \boldsymbol{y}(t). \tag{8}$$

In a PID controller, the control variable $\boldsymbol{u}(t)$ is composed of the proportional (P), integral (I), and derivative (D) terms and given by:

$$\boldsymbol{u}(t) = K_p \boldsymbol{e}(t) + K_i \int_0^t \boldsymbol{e}(\tau) d\tau + K_d \frac{d\boldsymbol{e}(t)}{dt}, \tag{9}$$

where $K_p, K_i, K_d \geq 0$ are the proportional, integral, and derivation gains, respectively. In PID control design, the P, I, and D play different roles: *Proportional term (P)* outputs a correction proportional to the current error $\boldsymbol{e}_t$, but alone leaves a steady-state offset; *Integral term (I)* accumulates past errors to remove residual bias, ensuring offsets are corrected even as proportional effects fade; and *Derivative term (D)* responds to the error's rate of change, damping rapid growth to improve stability and reduce overshoot.

**State-Feedback PID Controller.** A special case of the PID controller is obtained by choosing the measured output $\boldsymbol{y}(t)$ to be the state variable $\boldsymbol{x}(t)$ in Eqn. 7, yielding the following state-space model

$$\dot{\boldsymbol{x}}(t) = g(\boldsymbol{x}(t), \boldsymbol{u}(t), t), \boldsymbol{y}(t) = \boldsymbol{x}(t). \tag{10}$$

The control error then becomes the state tracking error, $\boldsymbol{e}(t) = \boldsymbol{x}_{sp}(t) - \boldsymbol{x}(t)$, and the system is controlled through feedback of the state (Åström & Murray, 2021).

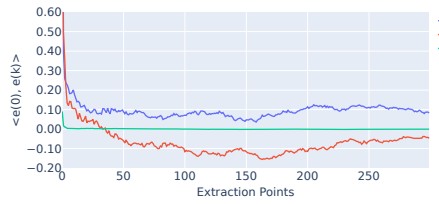 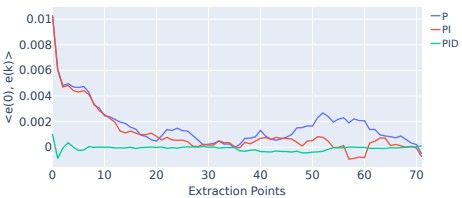

(a) Randomly Initialized LLama3      (b) Pretrained Qwen2.5-3B-Instruct

Figure 3: Scalar errors across time step of randomly initialized model after applying P, PI, and PID controller.

## 3   STEERING WITH A FEEDBACK CONTROLLER

In this section, we will formulate popular activation steering methods, such as ActAdd, DirAblate, and Mean-AcT, as a state-feedback P controller. Based on this new interpretation, we propose PID Steering, a novel steering method that uses a PID controller.

### 3.1   ACTIVATION STEERING AS A P CONTROLLER

We consider the state-feedback PID controller given in Eqn. 10 and the continuous steering vector $r(t)$ in which we replace the layer index $k$ by the time index $t$. Substituting the state tracking error $e(t)$ by the difference-in-means vector $r(t)$ and using the P controller whose system dynamics is governed by $g(x(t),u(t),t) = f(\rho_{\text{steer}}(x(t),u(t)),t) - x(t)$, we obtain

$$\dot{x}(t) = f(\rho_{\text{steer}}(x(t),K_p r(t)),t) - x(t). \tag{11}$$

We discretize Eqn. 11 using Euler method (Euler, 1768; Hairer et al., 1993) to obtain

$$x(k) - x(k-1) = f^{(k)}(\rho_{\text{steer}}(x(k-1),K_p r(k-1))) - x(k-1),$$

or equivalently,

$$x(k) = f^{(k)}(\rho_{\text{steer}}(x(k-1),K_p r(k-1))), \tag{12}$$

where $f^{(k)}(\cdot) = f(\cdot,k)$, a function depending on index $k$.

Comparing Eqn. 12 with Eqn. 1 and 2 shows that applying the steering vectors as in Section 2.2.1 is equivalent to implementing the P controller, where $f^{(k)}$ is the $k$-th layer in an LLM, $u(t) = K_p r(t)$ is the new steering vector. Thus, activation steering computes the expected state tracking error.

$$r(t) = \bar{e}(t) = \mathbb{E}_{q_{sp} \in \mathcal{D}_{\text{target}}^{(\text{train})}}[x_{sp}(t,q_{sp})] - \mathbb{E}_{q \in \mathcal{D}_{\text{source}}^{(\text{train})}}[x(t,q)]. \tag{13}$$

This expected state tracking error, i.e., the difference-in-means vector $r(t)$, can be computed non-sequentially or sequentially, as explained in Section 2.2.2. When $r(t)$ is computed non-sequentially and $\rho_{\text{steer}}(x(k),u(k)) = x(k) + \alpha u(k)$ or $x(k) - u(k)u(k)^\top x(k)$, we obtain ActAdd or DirAblate, respectively. When $r(t)$ is computed sequentially and $\rho_{\text{steer}}(x(k),u(k)) = x(k) + \alpha u(k)$, we attain Mean-AcT.

**Limitations of P Controller.** There is always a steady state error in P control. The error decreases with increasing gain, but the tendency towards oscillation also increases. Since activation steering methods, i.e., ActAdd, DirAblate, and Mean-Act, are P controllers, they share the same limitations. We informally state our theoretical guarantees that P-control activation steering methods cannot alleviate the steady state error in Proposition 1 below and provide detailed proofs in Appendix B.4.

**Proposition 1 (Steady-state error of P-control activation steering)** *P-control activation steering ensures input-to-state stability (ISS) for an appropriate range of $K_p$. However, there still exists a steady-state error due to the disturbance $w(k)$ to the state of the system. In the best case, when $w(k)$ converges to $w$, under a mild condition, the expected error, i.e., the difference-in-means, $r(k) = \bar{e}(k)$ eventually converges to a steady state $\bar{e}_{ss} \propto w$. Therefore, $\bar{e}_{ss} \neq 0$ if $w \neq 0$.*

We further provide empirical validation of Proposition 1 and illustrate the effect of integral and derivative terms (discussed in Section 4.2) in Fig. 3. To achieve this, we apply Sequential P-control activation steering (P Steering) on a randomly initialized model with 150 layers deep, and pretrained Qwen2.5-3B-Instruct. We plot the scalar signal $\langle \bar{e}(0), \bar{e}(k) \rangle$ which measures how much of the mean error at layer $k$ remains aligned with the initial mean error (see Appendix B.6 for further explanation). A nonzero plateau of this quantity indicates a persistent component of the initial error (i.e., a steady-state error), whereas convergence to values near zero means that this component has been eliminated.

In this light, the behaviour in Figure 3 matches the theoretical discussion: the P curve (blue) decays but clearly settles on a nonzero plateau, confirming that P-control is ISS but admits a steady-state error. The PI curve (red) crosses zero and, after the trasient, remains close to it, meaning that the integral action eliminates the steady-state offsets in $< e(0), e(k) >$; the large negative dip before convergence is the classical overshoot of PI control, which we discuss in the "Limitations of PI control" in Section 4.2.1 of our main text and analysis in Appendix B.6. The PID curve (green) also settles near zero but with substantially reduced overshoot, showing that derivative action damps the PI transient while preserving its steady-state advantage. We analyze the overshoot reduction of PID control in Theorem 2 in Section 4.2.2 of our main text and in Appendix B.7.2.

## 3.2 PROPORTIONAL–INTEGRAL–DERIVATIVE (PID) STEERING

### 3.2.1 OVERVIEW

To overcome the steady-state error inherent in P-control activation steering, we extend the method by adding integral (I) and derivative (D) terms to the steering vectors. PID Steering thus (i) reacts immediately to errors via the P term for greater responsiveness, (ii) removes steady-state offsets with the I term, ensuring convergence to the desired set point, and (iii) anticipates error trends through the D term, improving stability and reducing overshoot. Together, these properties yield the following advantages:

- **Generalization.** PID Steering extends P-control methods like ActAdd, DirAblate, and Mean-AcT by adding integral and derivative components.
- **Methodological Agnosticism.** Our PID framework can be applied across different activation steering techniques, including ActAdd, DirAblate, and Mean-AcT.
- **Stability.** We theoretical prove and empirical demonstrate that PID Steering reduces steady-state error and overshoot in P-controllers, improving existing steering methods.
- **Interpretability.** Derived from classical feedback control (Minorsky, 1922), the framework inherits the simplicity and interpretability that underpin the wide use of PID controllers.

### 3.2.2 COMPUTING THE STEERING DIRECTION USING A PID FEEDBACK CONTROLLER

Following Section 3.1, we consider the state-feedback PID controller in Eqn. 10, replacing the state tracking error $e(t)$ with the difference-in-means vector $r(t)$. With the PID controller governed by the system dynamics $g(\boldsymbol{x}(t), \boldsymbol{u}(t), t) = f(\rho_{\text{steer}}(\boldsymbol{x}(t), \boldsymbol{u}(t)), t) - \boldsymbol{x}(t)$, we obtain

$$\dot{\boldsymbol{x}}(t) = f(\rho_{\text{steer}}(\boldsymbol{x}(t), K_p \boldsymbol{r}(t) + K_i \int_0^t \boldsymbol{r}(\tau) d\tau + K_d \frac{d\boldsymbol{r}(t)}{dt}), t) - \boldsymbol{x}(t). \tag{14}$$

Eqn. 14 defines the continuous-time model of PID Steering, whose steering vector is given by:

$$\boldsymbol{u}(t) = K_p \boldsymbol{r}(t) + K_i \int_0^t \boldsymbol{r}(\tau) d\tau + K_d \frac{d\boldsymbol{r}(t)}{dt}. \tag{15}$$

In order to obtain the discrete-time formulation of PID Steering, we first discretize the sytem dynamics $\dot{\boldsymbol{x}}(t) = f(\rho_{\text{steer}}(\boldsymbol{x}(t), \boldsymbol{u}(t)), t) - \boldsymbol{x}(t)$ using Euler method (Euler, 1768; Hairer et al., 1993), same as in Section 3.1, and attain

$$\boldsymbol{x}(k) = f^{(k)}(\rho_{\text{steer}}(\boldsymbol{x}(k-1), \boldsymbol{u}(k-1))), \tag{16}$$

Next, we discretize $\boldsymbol{u}(t)$ given in Eqn. 15 to obtain $\boldsymbol{u}(k)$ using Lemma 1 below.

**Lemma 1 (Discretizing PID steering vector)** *Consider the continuous PID steering vector defined in Eqn. 15. The discrete-time PID steering vector is given by:*

$$\boldsymbol{u}(k) = K_p \boldsymbol{r}(k) + K_i \sum_{j=0}^{k-1} \boldsymbol{r}(j) + K_d (\boldsymbol{r}(k) - \boldsymbol{r}(k-1)). \tag{17}$$

Proof of Lemma 1 is in Appendix B.1. With Eqn. 16 and Lemma 1, we now define PID Steering.

**Definition 1 (PID Steering)** *Given a large language model whose layers are $\left\{ f^{(k)} \right\}_{k=1}^K$ and a steering function $\rho_{steer}$, PID Steering constructs the steering vectors as follows:*

$$\boldsymbol{u}(k) = K_p \boldsymbol{r}(k) + K_i \sum_{j=0}^{k-1} \boldsymbol{r}(j) + K_d (\boldsymbol{r}(k) - \boldsymbol{r}(k-1)), \tag{18}$$

*where for non-sequential mapping,*

$$\boldsymbol{r}(k) = \mathbb{E}_{\boldsymbol{q}_{sp} \in \mathcal{D}_{target}^{(train)}}[\boldsymbol{x}_{sp}(k, \boldsymbol{q}_{sp})] - \mathbb{E}_{\boldsymbol{q} \in \mathcal{D}_{source}^{(train)}}[\boldsymbol{x}(k, \boldsymbol{q})],$$

*and for sequential mapping,*

$$\tilde{\boldsymbol{x}}(k) = f^{(k)}\big(\rho_{steer}\big(\boldsymbol{x}(k-1), \boldsymbol{u}(k-1)\big)\big), \quad \boldsymbol{r}(k) = \mathbb{E}_{\boldsymbol{q}_{sp} \in \mathcal{D}_{target}^{(train)}}[\boldsymbol{x}_{sp}(k, \boldsymbol{q}_{sp})] - \mathbb{E}_{\boldsymbol{q} \in \mathcal{D}_{source}^{(train)}}[\tilde{\boldsymbol{x}}(k, \boldsymbol{q})].$$

## 4 THEORETICAL ANALYSIS OF PID STEERING

This section provides theoretical evidence for our claims: (i) adding integral action (PI) reduces steady-state error that remains under pure P-control (Proposition 3); and (ii) adding a derivative term (PID) preserves bias removal while mitigating oscillations/overshoot (Theorem 1 and 2). We denote $\boldsymbol{K}_p := K_p \boldsymbol{I}$, $\boldsymbol{K}_i := K_i \boldsymbol{I}$, and $\boldsymbol{K}_d := K_d \boldsymbol{I}$. Detailed proofs are provided in Appendix B.

### 4.1 DYNAMICS OF THE AVERAGE ERROR ACROSS LAYERS

To formalize the problem, we consider $N$ pairs of prompts/input tokens from two contrastive datasets, e.g. harmful and harmless, $\{(\boldsymbol{q}_i^+, \boldsymbol{q}_i^-)\}_{i=1}^N$ with corresponding activations $\boldsymbol{x}_i^{\pm}(k) \in \mathbb{R}^d$ at layer $k$. A steering input $\boldsymbol{u}(k)$ perturbs the undesired branch:

$$\boldsymbol{x}_i^-(k+1) = f_i^{(k)}\big(\boldsymbol{x}_i^-(k) + \boldsymbol{u}(k)\big). \tag{19}$$

Let $\bar{\boldsymbol{e}}(k) := \frac{1}{N} \sum_{i=1}^N (\boldsymbol{x}_i^+(k) - \boldsymbol{x}_i^-(k))$, the error dynamics of activation steering is then given by Proposition 2 below.

**Proposition 2 (Error dynamics of activation steering)** *The error dynamics $\bar{\boldsymbol{e}}(k)$ in activation steering is of the form:*

$$\bar{\boldsymbol{e}}(k+1) = \bar{\boldsymbol{A}}(k)\bar{\boldsymbol{e}}(k) - \bar{\boldsymbol{A}}(k)\boldsymbol{u}(k) + \boldsymbol{w}(k), \tag{20}$$

*where $\bar{\boldsymbol{A}}(k)$ is the mean local Jacobian of $f_i^{(k)}$ at $\boldsymbol{x}_i^+(k)$ and the disturbance term $\boldsymbol{w}(k)$ collects heterogeneity. See Appendix B.3 for detailed proof and explanations of the terms.*

Our control objective is to drive $\bar{\boldsymbol{e}}(k)$ to zero with input-to-state stability (ISS) for disturbed discrete system Eqn. 20 (Jiang et al., 1999; Edwards et al., 2000).

### 4.2 STABILITY OF THE ERROR DYNAMICS: ROLES AND CAVEATS OF PI AND PID CONTROL

In the following stability analysis, we consider the orthogonal decomposition for the disturbance $\boldsymbol{w}(k) = \boldsymbol{w}^{\|}(k) + \boldsymbol{w}^{\perp}(k)$, where $\boldsymbol{w}^{\|}(k) \in \text{Im}\bar{\boldsymbol{A}}(k)$ and $\boldsymbol{w}^{\perp}(k) \in (\text{Im}\bar{\boldsymbol{A}}(k))^{\perp}$.

#### 4.2.1 PI CONTROL

The following proposition provides a theoretical guarantee of PI Steering's steady-state error reduction.

**Proposition 3 (Stabilizing the PI loop reduces steady-state error)** *Let $\boldsymbol{M}_p(k) = \bar{\boldsymbol{A}}(k)(\boldsymbol{I} - \boldsymbol{K}_p)$, and denote $\|\boldsymbol{K}_i\| =: h$. Assume $\sup_k \|\bar{\boldsymbol{A}}(k)\| \leq M < \infty$ and $\sup_k \|\boldsymbol{M}_p(k)\| \leq q < 1$. If $q + Mh < 1$, then the PI closed-loop control is ISS. Furthermore, the integral part exactly cancels the matched disturbance component $\boldsymbol{w}^{\|}$. The remaining error is due only to the unmatched component $\boldsymbol{w}^{\perp}$, which cannot be compensated. Full proof and term explanations provided in Appendix B.5*

**Limitations of PI control.** Overshoot is common under PI: the closed loop oscillates about the setpoint before settling (Åström & Hägglund, 1995a, Ch. 3, §3.3, pp. 68-69), and large overshoot can arise with a high integral gain $\boldsymbol{K}_i$. In our steering setting, we explain this by scalarizing the dynamics along a reference direction . The scalarized integral state accumulates past error, pushing the trajectory beyond the setpoint; when the scalarized error changes sign, the integral discharges and the error subsequently approaches zero. See Fig. 7 for an illustration and Appendix B.6 for the formal derivation.

#### 4.2.2 PID CONTROL

The derivative action counteracts PI-induced oscillations near the setpoint by responding to decreases in the scalarized error, while preserving the integral term's bias-removal role, as shown in Theorems 1 and 2. For detailed proofs and explanations, see Appendix B.7.

**Theorem 1 (Stabilizing the PID loop preserves bias removal)** *Let $\boldsymbol{M}_p(k) = \bar{\boldsymbol{A}}(k)(\boldsymbol{I} - \boldsymbol{K}_p)$, and denote $\|\boldsymbol{K}_i\| =: h$, $\|\boldsymbol{K}_d\| =: \ell$. Assume $\sup_k \|\bar{\boldsymbol{A}}(k)\| \leq M < \infty$ and $\sup_k \|\boldsymbol{M}_p(k)\| \leq q < 1$. If $q + Mh < 1$ (stable PI loop), then there exists $\ell > 0$ such that the PID closed-loop control is ISS. Therefore, the integral part in PID design still cancels the matched disturbance component $\boldsymbol{w}^{\|}$.*

Table 1: Toxicity mitigation results for Gemma-2B and Llama-8B, averaged over 10 runs. Lower is better for toxicity and perplexity; higher is better for MMLU. Bold = best, underline = second-best within each model.[1]

| | | Seq. | CLS Tox. (%) ↓ | 0-shot Tox. (%) ↓ | QVQ (%) ↓ | PPL Wikipedia ↓ | PPL Mistral-7B ↓ | MMLU ↑ |
|---|---|---|---|---|---|---|---|---|
| **Gemma2-2B** | Original | – | $4.17_{\pm0.32}$ | $13.42_{\pm1.08}$ | $14.17_{\pm0.08}$ | 13.98 | 6.68 | 53.1 |
| | ActADD | | $3.96_{\pm0.24}$ | $13.43_{\pm1.42}$ | $14.17_{\pm0.08}$ | $14.69_{\pm0.22}$ | $\mathbf{6.67}_{\pm0.15}$ | $\mathbf{53.00}_{\pm0.51}$ |
| | CAA | | $1.20_{\pm0.25}$ | $5.35_{\pm0.50}$ | $5.88_{\pm0.36}$ | $14.60_{\pm0.20}$ | $6.85_{\pm0.22}$ | $51.70_{\pm0.48}$ |
| | AURA | | $2.12_{\pm0.27}$ | $9.04_{\pm0.66}$ | $9.72_{\pm0.27}$ | $\mathbf{14.18}_{\pm0.14}$ | $7.04_{\pm0.34}$ | $\mathbf{53.00}_{\pm0.30}$ |
| | ITI-C | | $0.74_{\pm0.18}$ | $5.36_{\pm0.91}$ | $6.10_{\pm0.13}$ | $14.90_{\pm0.29}$ | $7.44_{\pm0.19}$ | $\underline{52.6}_{\pm0.55}$ |
| | Mean-AcT | | $1.12_{\pm0.23}$ | $5.20_{\pm0.42}$ | $5.80_{\pm0.15}$ | $\underline{14.53}_{\pm0.21}$ | $\underline{6.81}_{\pm0.19}$ | $51.74_{\pm0.55}$ |
| | Linear-AcT | | $0.95_{\pm0.36}$ | $5.37_{\pm0.80}$ | $5.92_{\pm0.11}$ | $14.75_{\pm0.22}$ | $7.24_{\pm0.24}$ | $51.63_{\pm0.50}$ |
| | Mean-AcT | ✓ | $\underline{0.68}_{\pm0.21}$ | $\underline{3.23}_{\pm0.44}$ | $\underline{3.70}_{\pm0.14}$ | $14.92_{\pm0.25}$ | $6.97_{\pm0.74}$ | $51.80_{\pm0.55}$ |
| | Linear-AcT | ✓ | $1.00_{\pm0.27}$ | $4.13_{\pm0.89}$ | $4.64_{\pm0.04}$ | $14.98_{\pm0.22}$ | $7.13_{\pm0.70}$ | $51.47_{\pm0.50}$ |
| | PID-AcT (Ours) | ✓ | $\mathbf{0.51}_{\pm0.21}$ | $\mathbf{2.90}_{\pm0.55}$ | $\mathbf{3.40}_{\pm0.04}$ | $15.22_{\pm0.24}$ | $7.02_{\pm0.65}$ | $51.30_{\pm0.52}$ |
| **Llama3-8B** | Original | – | 5.80 | 15.00 | $15.81_{\pm0.09}$ | 9.06 | 5.68 | 65.30 |
| | ActADD | | $5.57_{\pm0.45}$ | $15.73_{\pm0.21}$ | $16.48_{\pm0.19}$ | $9.71_{\pm0.46}$ | $5.85_{\pm0.26}$ | $\mathbf{65.50}_{\pm0.34}$ |
| | CAA | | $1.82_{\pm0.36}$ | $6.70_{\pm0.58}$ | $7.40_{\pm0.06}$ | $9.40_{\pm0.25}$ | $5.50_{\pm0.30}$ | $64.30_{\pm0.37}$ |
| | AURA | | $1.90_{\pm0.61}$ | $8.12_{\pm0.85}$ | $8.80_{\pm0.17}$ | $9.52_{\pm0.32}$ | $6.05_{\pm0.30}$ | $\mathbf{65.50}_{\pm0.33}$ |
| | ITI-C | | $1.60_{\pm0.22}$ | $6.53_{\pm0.66}$ | $7.19_{\pm0.06}$ | $9.48_{\pm0.24}$ | $6.17_{\pm0.14}$ | $\underline{64.70}_{\pm0.44}$ |
| | Mean-AcT | | $1.78_{\pm0.33}$ | $6.56_{\pm0.54}$ | $7.30_{\pm0.25}$ | $\underline{9.36}_{\pm0.28}$ | $\mathbf{5.45}_{\pm0.34}$ | $64.35_{\pm0.39}$ |
| | Linear-AcT | | $1.87_{\pm0.39}$ | $6.55_{\pm0.21}$ | $7.30_{\pm0.15}$ | $\mathbf{9.35}_{\pm0.17}$ | $5.56_{\pm0.33}$ | $64.55_{\pm0.33}$ |
| | Mean-AcT | ✓ | $\underline{1.21}_{\pm0.41}$ | $5.09_{\pm0.64}$ | $\underline{5.73}_{\pm0.05}$ | $9.83_{\pm0.21}$ | $5.71_{\pm0.33}$ | $64.22_{\pm0.40}$ |
| | Linear-AcT | ✓ | $1.68_{\pm0.48}$ | $6.47_{\pm0.38}$ | $7.12_{\pm0.26}$ | $9.48_{\pm0.19}$ | $\underline{5.46}_{\pm0.44}$ | $64.49_{\pm0.38}$ |
| | PID-AcT (Ours) | ✓ | $\mathbf{0.72}_{\pm0.49}$ | $\mathbf{4.36}_{\pm0.81}$ | $\mathbf{4.90}_{\pm0.14}$ | $9.56_{\pm0.20}$ | $6.08_{\pm0.37}$ | $64.50_{\pm0.36}$ |

**Theorem 2 (PID reduces the first-overshoot amplitude)** *Let the first overshoot occur at index $k_0$ with amplitude $A_0$ (definition in Eqn. 54). Then, the first-overshoot amplitude under PID Steering, $A_0^{\mathrm{PID}}$, satisfies $A_0^{\mathrm{PID}} \leq A_0^{\mathrm{PI}}$, where $A_0^{\mathrm{PI}}$ denotes the corresponding amplitude under PI Steering.*

To support the theory, we present empirical evidence in Fig. 3. PI and PID controllers clearly improve over P-only control: PI removes steady-state error but causes large overshoot, while adding the derivative term mitigates overshoot and enables faster, cleaner convergence to zero.

## 5    CONTROLLING THE STEERING EFFECT

In this section, we demonstrate the applicability and effectiveness of PID-Steering by using it as a drop-in replacement for the steering vector computation step across multiple steering frameworks.

### 5.1    TOXICITY MITIGATION

We evaluate the effectiveness of PID Steering for toxic language mitigation in comparison to sequential steering methods, specifically Linear-AcT and Mean-AcT (Rodriguez et al., 2025), by closely following their experimental setup. We apply PID-Steering into Mean-AcT and call it PID-AcT. Results from other baselines, namely ActADD (Turner et al., 2024a), CAA (Rimsky et al., 2024), AURA (Suau et al., 2024), ITI-C (Li et al., 2023), are also reported.

**Experimental Setup.** Our evaluation is conducted on Gemma2-2B (Gemma Team et al., 2024) and Llama3-8B (Dubey et al., 2024), using 1,000 randomly sampled prompts from the RealToxicityPrompts dataset (Gehman et al., 2020). Toxicity is quantified with a ROBERTA-based classifier (Logacheva et al., 2022), following the methodology of Suau et al. (2024). We also assess toxicity in a zero-shot setting by employing Llama3-8B-Instruct as an LLM-as-a-judge (Zheng et al., 2023). Additionally, in Tab. 1, we report QVQ (%) metric measured by QVQ-72B-Preview (Qwen Team, 2024) as LLM-as-a-judge's.

To measure general utility of the intervened models, we report: (i) perplexity (PPL) on a fixed set of 20k Wikipedia sentences, (ii) PPL of model-generated outputs evaluated with Mistral-7B (Jiang et al., 2023), and (iii) 5-shot MMLU (Hendrycks et al., 2021) accuracy.

**Results. PID-AcT achieves the strongest toxicity reduction with minimal utility loss across both models.** As shown in Tab. 1, it lowers toxicity by up to **8.2**× on Gemma2-2B and **8.1**× on Llama3-8B under both classifier and LLM-judge evaluations. PID-AcT outperforms Mean-AcT and Linear-AcT within the sequential family (Seq., ✓) and surpasses strong activation-editing baselines (ActADD, AURA, ITI-C, CAA). Utility remains stable: perplexity is comparable to other baselines; MMLU accuracy aligns with AcT-based methods and slightly lower than non-AcT approaches, likely reflecting properties of the AcT framework rather than our method. Overall, AcT-style methods yield stronger toxicity mitigation, with PID-AcT consistently ranking highest.

## 5.2 JAILBREAKING LARGE LANGUAGE MODELS

We evaluate our method on ActAdd within the Angular Steering framework (Vu & Nguyen, 2025) on the jailbreaking task, which seeks to override a model's refusal behavior and elicit harmful outputs.

**Experimental Setup.** Following (Vu & Nguyen, 2025), we replace DIM with our method and baselines RePE (Zou et al., 2023a) and ITI (Li et al., 2023). Refusal directions are built from 80% of ADVBENCH (Zou et al., 2023b) and 512 harmless ALPACA (Taori et al., 2023) samples, with the remaining 20% for evaluation. General LM ability is tested on TINYBENCHMARKS (Maia Polo et al., 2024). We evaluate across Gemma2, LLaMA3, and Qwen2.5 models (3B–14B).

Table 2: Comparison of Original, DIM, ITI, RePE, and PID across models on ASR and general benchmarks. Bold = best, underline = second-best within each model (ASR column). Refer to Tab. 3 for results on all tested models.

| | Method | ASR↑ | tinyArc↑ | tinyGSM8k strict↑ | tinyMMLU↑ | tinyTruthQA↑ | tinyHellaSwag↑ | tinyWinoGrande↑ |
|---|---|---|---|---|---|---|---|---|
| **Qwen2.5-3B Instruct** | *Original* | – | 62.29 | 17.64 | 68.03 | 56.43 | 73.18 | 70.65 |
| | DIM | 74.03 | 61.95 | 14.80 | 66.11 | 54.95 | 72.40 | 69.85 |
| | ITI | 70.19 | 61.28 | 15.57 | 66.62 | 54.75 | 72.71 | 70.12 |
| | RePE | 68.44 | 61.05 | 14.60 | 65.70 | 54.30 | 72.03 | 69.40 |
| | PID (ours) | **76.07** | 61.20 | 16.01 | 67.29 | 54.10 | 72.59 | 69.72 |
| **Qwen2.5-14B Instruct** | *Original* | – | 73.96 | 90.12 | 74.60 | 64.50 | 82.70 | 73.77 |
| | DIM | 90.38 | 72.74 | 87.01 | 74.30 | 63.01 | 81.94 | 72.93 |
| | ITI | 33.65 | 73.15 | 89.27 | 74.55 | 64.03 | 82.24 | 73.31 |
| | RePE | 25.42 | 72.40 | 86.20 | 73.90 | 63.20 | 81.52 | 72.60 |
| | PID (ours) | **92.65** | 72.13 | 88.96 | 74.52 | 63.60 | 82.60 | 73.04 |
| **Llama3.1-8B Instruct** | *Original* | – | 65.33 | 63.21 | 62.02 | 54.39 | 82.51 | 65.56 |
| | DIM | 93.26 | 62.01 | 60.57 | 60.96 | 54.17 | 81.73 | 64.81 |
| | ITI | 79.80 | 64.26 | 61.85 | 61.37 | 54.33 | 82.01 | 65.21 |
| | RePE | 70.42 | 61.40 | 60.00 | 60.20 | 53.70 | 81.35 | 64.45 |
| | PID (ours) | **94.85** | 62.30 | 61.99 | 61.54 | 54.24 | 81.87 | 64.93 |
| **Gemma2-9B Instruct** | *Original* | – | 69.31 | 83.19 | 76.60 | 55.07 | 82.31 | 72.34 |
| | DIM | 77.88 | 68.21 | 80.14 | 72.29 | 51.86 | 81.45 | 71.51 |
| | ITI | 35.57 | 68.32 | 81.47 | 75.33 | 53.13 | 81.70 | 71.82 |
| | RePE | 28.64 | 67.50 | 79.20 | 71.10 | 51.10 | 81.20 | 71.15 |
| | PID (ours) | **79.50** | 67.91 | 79.24 | 74.89 | 52.49 | 81.59 | 71.42 |
| **Gemma2-27B Instruct** | *Original* | – | 73.45 | 86.91 | 76.11 | 61.36 | 83.24 | 75.47 |
| | DIM | 74.03 | 72.13 | 84.70 | 74.59 | 59.49 | 81.79 | 74.21 |
| | ITI | 37.36 | 72.84 | 86.38 | 75.51 | 60.85 | 82.83 | 75.11 |
| | RePE | 24.19 | 71.03 | 84.00 | 73.90 | 58.79 | 80.82 | 73.44 |
| | PID | **79.80** | 72.93 | 86.60 | 75.67 | 60.74 | 82.95 | 75.06 |

**Results. PID Steering consistently outperforms DIM and scales robustly across models and metrics** (see Tab. 2). On Qwen2.5-14B and LLaMA3.1-8B, PID achieves the largest ASR reductions of **92.7%** and **94.9%**, exceeding DIM by 1.5-2 points, while maintaining almost the same performance, with marginal cost, on TinyBenchmarks. Smaller models also see consistent gains: +2.0 ASR on Qwen2.5-3B and +1.3 on LLaMA3.2-3B. In contrast, ITI and RePE fail to scale, collapsing on larger models with ASR values of 33.7 and 25.4, respectively, on Qwen2.5-14B. A full version of Tab. 2 which also studies Qwen2.5-7B and Llama3.2-3B is provided in Appendix C.2.

## 5.3 IMAGE GENERATION STYLES CONTROL

We study activation steering in diffusion models using FLUX.1.Schnell's denoising transformer (Labs, 2024), built on T5-XXL encoders (Raffel et al., 2020) and requiring just 4 diffusion steps.

**Experimental Setup.** Following (Rodriguez et al., 2025), we intervene on all normalization layers after most residual blocks in FLUX. Style/concept expression is measured by a CLIP zero-shot classifier with two labels (A picture of a {style/concept} vs. A picture of something), and content preservation by CLIPScore (Hessel et al., 2021). Training uses 2,048 COCO Captions (Chen et al., 2015) prompts augmented with *cyberpunk*/*steampunk* modifiers from LLaMA-8B-Instruct (source = unmodified $p$, target = modified $q$). Evaluation samples 512 validation prompts to generate images across intervention strengths.

**Results.** In Fig. 4, increasing the intervention strength from 0 to 1 produces a smooth and coherent shift in style–e.g., neon tones for cyberpunk, metallic textures for steampunk–while preserving core content.

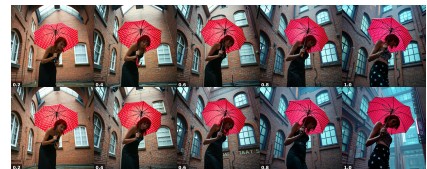
(a) Cyberpunk concept.

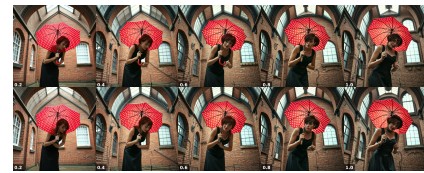
(b) Steampunk concept.

Figure 4: Qualitative results of activation steering in FLUX-Schnell across two style concepts with the prompt *"Lady bent over with red polka dot umbrella inside a brick building."*

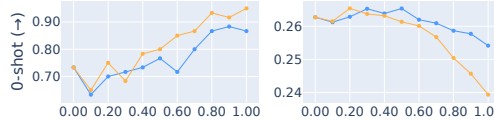
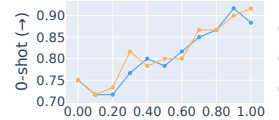
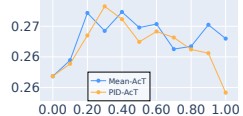

(a) Cyberpunk concept

(b) Steampunk concept

Figure 5: 0-shot and CLIPScore results for "cyberpunk" and "steampunk" concept.

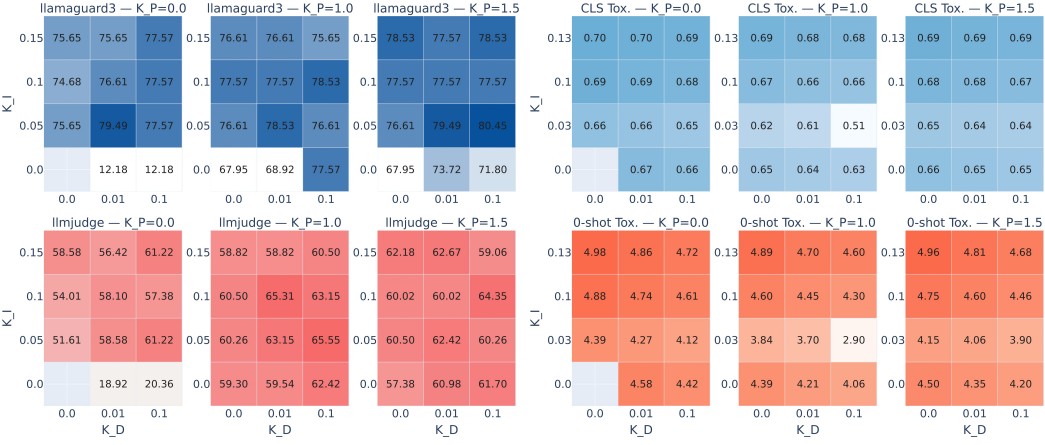

(a) Gemma-2-9b-it on Jailbreaking task.

(b) Gemma-2-2B on Toxicity mitigation task.

Figure 6: Ablation Study on $(K_p, K_i, K_d)$ parameters. We sweep through different values of $K_p, K_i, K_d$ and report (a) Llamaguard3 (ASR in Tab. 2) and Llmjudge (evaluated using QVQ-72B-Preview) metrics for Gemma-2-9b-it, and (b) CLS Tox. and 0-shot Tox. metrics for Gemma2-2B.

Moderate strengths yield strong yet faithful stylization, and even at high strengths, semantic alignment remains largely intact. Quantitatively (Figs. 5), style intensity rises monotonically, with PID-AcT outperforming Mean-AcT at mid strengths (0.4–0.8). CLIPScore shows the expected trade-off: both methods decline steadily, with PID-AcT only slightly lower and by a small margin.

## 5.4 ABLATION STUDY: CONTROLLER GENERALIZATION ACROSS ARCHITECTURES

Figs. 6a and 6b reveal a remarkably consistent PID response surface across different models and tasks. In the Gemma-2-9B-IT jailbreak setting (Fig. 6a), increasing $K_i$ substantially reduces harmfulness under both LLaMAguard3 and LLM-Judge, with the strongest gains concentrated in $K_i \in [0.05, 0.10]$. However, large $K_i$ introduces instability when $K_p$ is small, visible as degraded pockets at high integral gain; incorporating a small derivative term ($K_d \in [0.01, 0.05]$) reliably suppresses these oscillations and restores smooth, monotone improvements. A nearly identical pattern appears in the Gemma-2-2B toxicity setting (Fig. 6b), despite differences in scale and objective: toxicity decreases sharply as $K_i$ increases from 0 to 0.05 before saturating, while $K_d$ stabilizes high-$K_i$ regimes–particularly when $K_p = 0$. The zero-shot toxicity metric follows the same geometry, with $K_i$ driving consistent gains and $K_d$ preventing degradation at strong integral levels. Overall, across architectures, sizes, and evaluation metrics, $K_i$ primarily improves bias removal, $K_d$ damps instability, and $K_p$ modulates amplitude without altering the underlying structure. This cross-setting consistency supports our claim that PID behavior stems from architecture-agnostic error dynamics and generalizes without retraining.

## 6 CONCLUDING REMARKS

We introduced PID Steering, a control-theoretic approach to activation steering that models layer-wise representations as a dynamical system. This framework unifies prior methods, offers robustness

guarantees, and leverages PID dynamics for computing steering vectors. Across language and diffusion models, PID Steering achieves stronger and more stable performance than existing approaches in toxicity mitigation, jailbreak prevention, and style control, while preserving model utility. Our results highlight control theory as a principled foundation for developing reliable and generalizable steering methods. A limitation of our work is the use of "stability-first, one-gain-at-a-time" analytical strategy to find controller gains: it clarifies the role of each component but may miss optimal choices and can overlook broader feasible regions. To address this, numerical methods, for example, LMI-based computations, can be employed. We leave these for future work.

ACKNOWLEDGMENTS

This research / project is supported by the National Research Foundation Singapore under the AI Singapore Programme (AISG Award No: AISG2-TC-2023-012-SGIL). This research / project is supported by the Ministry of Education, Singapore, under the Academic Research Fund Tier 1 (FY2023) (A-8002040-00-00, A-8002039-00-00). This research / project is also supported by the NUS Presidential Young Professorship Award (A-0009807-01-00) and the NUS Artificial Intelligence Institute–Seed Funding (A-8003062-00-00).

NYP acknowledges support from the Application Driven Mathematics Program funded and organized by the Vingroup Innovation Fund and VinBigData.

We would like to thank FPT for their generous support in providing discounted GPU access on FPT AI Factory, which enabled the experiments in this work.

**Ethics Statement.** Given the nature of the work, we do not foresee any negative societal and ethical impacts of our work. However, risks remain: by tightening activation-level control, PID Steering may inadvertently ease the generation of nuanced harmful content (e.g., persuasive misinformation or biased narratives). Although it does not expand the baseline risk profile of LLMs, robust safeguards, transparency, accountability, and ongoing ethical review are required for responsible use.

**Reproducibility Statement.** Source codes for our experiments are provided in the supplementary materials of the paper. The details of our experimental settings and computational infrastructure are given in Section 5 and the Appendix. All datasets that we used in the paper are published, and they are easy to access in the Internet.

**LLM Usage Declaration.** We use large language models (LLMs) for grammar checking and correction.

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

# Supplement to "Activation Steering with a Feedback Controller"

**Table of Contents**

## A  RELATED WORKS

Recent works increasingly frame large language models (LLMs) as *dynamical systems*, where generation is a trajectory in latent space. This view shifts activation steering from heuristic nudging to principled *control*: rather than biasing outputs without guarantees, controllers enforce constraints on trajectories with formal assurances (Cheng & Amo Alonso, 2024). In our PID-steering framework, this distinction is key: we treat the model as a plant with hidden states evolving under controlled interventions.

**Controllability of LLMs.** Soatto et al. (2023) model decoder-only LLMs as discrete-time stochastic systems and prove that, under idealized assumptions (Euclidean embeddings, well-trained semantics), they are controllable in the space of meanings. This establishes that any semantic state is theoretically reachable, but probabilities may be negligible in practice, and emergent behaviors like chain-of-thought are not captured. Their results highlight both opportunities and risks: controllability opens adversarial attack surfaces but also suggests defensive controllers.

**Prompting as open-loop control.** Luo et al. (2023) recast multi-round prompt engineering as an optimal control problem, where each prompt is a control input maximizing task reward. This provides a unifying formalism for prompt strategies, akin to open-loop control. Yet, the framework remains conceptual: metrics in discrete language space are poorly defined, and no guarantees of stability or convergence are provided.

**Closed-loop activation control.** Cheng & Amo Alonso (2024) propose *Linear Semantic Control* (LiSeCo), which projects activations into safe subspaces at each decoding step via a closed-form controller. This yields lightweight, guaranteed control of simple attributes (e.g., toxicity, sentiment).

However, the linearity assumption only approximates LLM embeddings, guarantees are local rather than global, and long-horizon stability remains unaddressed.

**Dynamic representation editing.** Kong et al. (2024) introduce *RE-CONTROL*, which learns a value function on hidden states and applies gradient-based interventions at test time. This dynamic approach generalizes steering into a Bellman-optimal control problem, balancing alignment with fluency. Still, accuracy of the learned value function is critical, test-time optimization adds overhead, and local interventions may not guarantee global alignment.

Together, these works move activation steering from heuristics to control theory. Soatto et al. (2023) prove fundamental controllability (but under strong assumptions), Luo et al. unify prompt strategies as open-loop control (without guarantees), Cheng & Amo Alonso (2024) derive closed-form activation control (limited to linear approximations), and Kong et al. (2024) extend to dynamic optimal control (with overhead and approximation risks).

## B    THEORETICAL PROOFS

### B.1    DISCRETIZED PID CONTROLLER

Implementing a continuous-time controller on digital hardware, such as PID, requires discretizing its derivative and integral terms (Åström & Hägglund, 1995a, p.95)

**Lemma 1 (Discretizing PID steering vector)** *Consider the continuous PID steering vector defined in Eqn. 15. The discrete-time PID steering vector is given by:*

$$\boldsymbol{u}(k) = K_p \boldsymbol{r}(k) + K_i \sum_{j=0}^{k-1} \boldsymbol{r}(j) + K_d(\boldsymbol{r}(k) - \boldsymbol{r}(k-1)). \tag{17}$$

**Proof.** We follow the discretization procedure for PID controllers in (Åström & Hägglund, 1995a, Sec. 3.6, Ch. 3). For simplicity, the sampling period is normalized to $h = 1$.

*Proportional term in Eqn. 15.*
$$P(t) = K_p \boldsymbol{r}(t).$$
The discrete-time form is obtained by substituting sampled variables for their continuous counterparts:
$$P(k) = K_p \boldsymbol{r}(k). \tag{21}$$

*Integral term in Eqn. 15.*

$$I(t) = K_i \int_0^t \boldsymbol{r}(\tau)d\tau \qquad \Rightarrow \qquad \frac{dI}{dt} = K_i \boldsymbol{r}(t).$$

Using forward Euler with $h = 1$,
$$I(k+1) - I(k) = K_i \boldsymbol{r}(k).$$

Hence
$$I(k+1) = I(k) + K_i \boldsymbol{r}(k),$$

which is equivalent to

$$I(k) = I(0) + K_i \sum_{j=0}^{k-1} \boldsymbol{r}(j) = K_i \sum_{j=0}^{k-1} \boldsymbol{r}(j), \tag{22}$$

since $I(0) = 0$.

*Derivative term in Eqn. 15.*

$$D(t) = K_d \frac{d\boldsymbol{r}(t)}{dt}.$$
Approximating the derivative by the backward Euler difference with $h = 1$ gives
$$D(k) = K_d \big(\boldsymbol{r}(k) - \boldsymbol{r}(k-1)\big). \tag{23}$$

Combining equation 21, equation 22, and equation 23 yields

$$\boldsymbol{u}(k) = K_p \boldsymbol{r}(k) + K_i \sum_{j=0}^{k-1} \boldsymbol{r}(j) + K_d \big(\boldsymbol{r}(k) - \boldsymbol{r}(k-1)\big).$$

$\square$

## B.2 BACKGROUND ON INPUT-TO-STATE STABILITY & NOTATIONS

**Background on Input-to-state Stability (ISS)** In our proofs, the input-to-state stability (ISS) of a system can be established either through the definition of an ISS system in (Jiang et al., 1999, Def. 2.1) and (Edwards et al., 2000, Def. 2.1), or via the use of an ISS-Lyapunov function, as outlined in (Jiang et al., 1999, Def. 2.2, Prop. 2.3) and (Edwards et al., 2000, Theorem 1). We also rely on the definition of a Lyapunov function and the difference Lyapunov equation for linear discrete-time homogeneous dynamical systems in (Gajic & Qureshi, 2008, Ch. 1, p. 8). The existence of a solution to the Lyapunov equation, together with its bound, is stated in (Gajic & Qureshi, 2008, Ch. 4, p. 110).

For reference, we briefly note that input-to-state stability (ISS) extends the classical notion of Lyapunov by explicitly accounting for external inputs: the state remains bounded and eventually converges whenever the input is bounded. A Lyapunov function provides an energy-like certificate for stability, while the associated Lyapunov equation offers a constructive method for obtaining such functions in linear settings. These notions are central for analyzing stability and will be used throughout our proofs.

**Conventions and assumptions (used throughout).** Let $\|\cdot\|$ denote the Euclidean norm on $\mathbb{R}^d$; for a matrix $M \in \mathbb{R}^{d \times d}$ we also write $\|M\|$ for the operator norm induced by the Euclidean norm, i.e. $\|M\| := \sup_{\|x\|=1} \|Mx\|$ (the spectral norm) (Horn & Johnson, 2012, pp. 343–346). We assume (i) $\sup_k \|\bar{A}(k)\| < \infty$; (ii) $w(k)$ is bounded (for a signal $w$ we set $\|w\|_\infty := \sup_k \|w(k)\|$); (iii) the controller gains are static and time-invariant scalar multiples of the identity, i.e., $K_p := K_p I$, $K_i := K_i I$, and $K_d := K_d I$. We use the standard meaning of the classes $\mathcal{K}$ and $\mathcal{KL}$ as in (Jiang et al., 1999).

## B.3 DYNAMICS OF THE AVERAGE ERROR ACROSS LAYERS

To formalize the problem setup, we consider $N$ pairs of contrastive prompt/input tokens $\{(q_i^+, q_i^-)\}_{i=1}^N$, where $q_i^+$ carries the desired property and $q_i^-$ represents the opposite. For discrete time (layer) $k$, let $x_i^\pm(k) \in \mathbb{R}^d$ denote the corresponding activation vectors. The layer-to-layer evolution is

$$x_i(k+1) = f_i^{(k)}(x(k)), i = 1, ..., N, \tag{24}$$

with $f_i^{(k)} : \mathbb{R}^d \to \mathbb{R}^d$ differentiable on the operating region. A steering input $u(k)$ is applied on the undesired branch:

$$x_i^-(k+1) = f_i^{(k)}(x_i^-(k) + u(k)). \tag{25}$$

Defining $\bar{x}^\pm(k) := \frac{1}{N}\sum_{i=1}^N x_i^\pm(k)$, we track the per-pair and average errors as

$$e_i(k) := x_i^+(k) - x_i^-(k), \qquad \bar{e}(k) := \bar{x}^+(k) - \bar{x}^-(k), \qquad \tilde{e}_i(k) = e_i(k) - \bar{e}(k). \tag{26}$$

Furthermore, we define $A_i(k)$ as the Jacobian of $f_i^{(k)}$ at $x_i^+(k)$:

$$A_i(k) := J_{f_i^{(k)}}(x_i^+(k)), \qquad \bar{A}(t) := \frac{1}{N}\sum_{i=1}^N A_i(k), \qquad \tilde{A}_i(k) = A_i(k) - \bar{A}(k). \tag{27}$$

The dynamic of the average error $\bar{e}(k)$ is then given by Proposition 2.

**Proposition 2 (Error dynamics of activation steering)** *The error dynamics $\bar{e}(k)$ in activation steering is of the form:*
$$\bar{e}(k+1) = \bar{A}(k)\bar{e}(k) - \bar{A}(k)u(k) + w(k), \tag{20}$$
*where $\bar{A}(k)$ is the mean local Jacobian of $f_i^{(k)}$ at $x_i^+(k)$ and the disturbance term $w(k)$ collects heterogeneity. See Appendix B.3 for detailed proof and explanations of the terms.*

**Proof.** The evolution of the average error $\bar{e}(k)$ through layers can be described as follows:

$$\bar{e}(k+1) = \bar{x}^+(k+1) - \bar{x}^-(k+1) = \frac{1}{N}\sum_{i=1}^N \left[ f_i^{(k)}(x_i^+(k)) - f_i^{(k)}(x_i^-(k) + u(k)) \right]. \tag{28}$$

Linearizing $f_i^{(k)}$ around $x_i^+(k)$, we obtain

$$f_i^{(k)}(x_i^+(k) + \delta_i(k)) \approx f_i^{(k)}(x_i^+(k)) + J_{f_i^{(k)}}(x_i^+(k)) \cdot \delta_i(k), \tag{29}$$

where $J_{f_i^{(k)}}$ denotes the Jacobian of $f_i^{(k)}$.

Setting $\delta_i(k)=-e_i(k)+u(k)$ yields

$$f_i^{(k)}\big(x_i^+(k)+\delta_i(k)\big) \approx f_i^{(k)}\big(x_i^+(k)\big) + A_i(k)\big(-e_i(k)+u(k)\big),$$

Insert this into Eqn.28 we obtain

$$\bar{e}(k+1) = \frac{1}{N}\sum_{i=1}^{N} A_i(k)e_i(k) - \bar{A}(k)u(k). \tag{30}$$

Recall that

$$e_i(k) = \bar{e}(k)+\tilde{e}_{(i)}(k), \text{ then } \frac{1}{N}\sum_{i=1}^{N}\tilde{e}_{(i)}(k)=0,$$

and

$$A_{(i)}(t) = \bar{A}(k)+\tilde{A}_{(i)}(k), \text{ then } \frac{1}{N}\sum_{i=1}^{N}\tilde{A}_{(i)}(k)=0.$$

Therefore,

$$\begin{aligned}
\bar{e}(k+1) &= \frac{1}{N}\sum_{i=1}^{N} A_i(k)e_i(k) - \bar{A}(k)u(k) \\
&= \frac{1}{N}\sum_{i=1}^{N}\bar{A}(k)\bar{e}_i(k) - \bar{A}(k)u(k) + \frac{1}{N}\sum_{i=1}^{N}\tilde{e}_{(i)}(k)\tilde{A}_{(i)}(k) \\
&\quad + \underbrace{\bar{A}(k)\frac{1}{N}\sum_{i=1}^{N}\tilde{e}_{(i)}(k) + \bar{e}(k)\frac{1}{N}\sum_{i=1}^{N}\tilde{A}_{(i)}(k)}_{=0}
\end{aligned} \tag{31}$$

We then obtain the final state-space model for the dynamics of $\bar{e}(t)$ as

$$\bar{e}(k+1) = \bar{A}(k)\bar{e}(k) - \bar{A}(k)u(k) + w(k), \tag{32}$$

where

$$w(k) = \frac{1}{N}\sum_{i=1}^{N}\tilde{A}_i(k)\tilde{e}_i(k),$$

which acts as a time-dependent exogenous disturbance to the model $\qquad\square$

### B.4 PROPORTIONAL (P) CONTROL

Consider proportional control with

$$u(k) = K_p\bar{e}(k), \quad (K_i=K_d=0). \tag{33}$$

The dynamics Eqn. 20 then become

$$\bar{e}(k) = M_p(k)\bar{e}(k) + w(k), \tag{34}$$

where $M_p(k) = \bar{A}(k)(I - K_p)$.

With a suitable choice of $K_p$, the system can be made input-to-state stable (ISS); that is, there exist a $\mathcal{KL}$-function $\beta$ and a $\mathcal{K}$-function $\gamma$ such that, for all disturbance $w$ with bounded sup norm and all initial states $\bar{e}(0)$,

$$\|\bar{e}(k)\| \leq \beta(\|\bar{e}(0)\|,k) + \gamma(\|w\|_\infty), \quad k\in\mathbb{Z}_{\geq 0}, \tag{35}$$

see (Jiang et al., 1999, Def. 2.1).

In particular, the error decays from the initial condition and remains bounded under bounded disturbances.

**Proposition 1 (Steady-state error of P-control activation steering)** *P-control activation steering ensures input-to-state stability (ISS) for an appropriate range of $K_p$. However, there still exists a steady-state error due to the disturbance $w(k)$ to the state of the system. In the best case, when $w(k)$ converges to $w$, under a mild condition, the expected error, i.e., the difference-in-means, $r(k)=\bar{e}(k)$ eventually converges to a steady state $\bar{e}_{ss}\propto w$. Therefore, $\bar{e}_{ss}\neq 0$ if $w\neq 0$.*

**Proof.** Assume $\sup_k \|\bar{A}(k)\| \leq M < \infty$, $K_p = K_p I = pI$ with $p > 0$. Since $M_p(k) = \bar{A}(k)(I - K_p) = \bar{A}(k)(1-p)I$, by sub-multiplicative property of matrix norm we have

$$\|M_p(t)\| \leq \|\bar{A}(t)\|\|(1-p)I\| \leq M|1-p| =: q. \tag{36}$$

For $p \in \left(1 - \frac{1}{M}, 1 + \frac{1}{M}\right)$, we have $q < 1$.

Expanding recursively,

$$\bar{e}(k) = M_p(k-1)\cdots M_p(0)\bar{e}(0) + \sum_{j=0}^{k-1} M_p(k-1)\cdots M_p(j+1)w(j).$$

Hence,

$$\|\bar{e}(k)\| \leq q^k \|\bar{e}(0)\| + \sum_{j=0}^{k-1} q^{k-1-j}\|w(j)\| \leq q^k \|\bar{e}(0)\| + \frac{1-q^k}{1-q}\|w\|_\infty \leq q^k \|\bar{e}(0)\| + \frac{1}{1-q}\|w\|_\infty. \tag{37}$$

Since $q < 1$, we can set $\beta(s,k) = q^k s$, which is a $\mathcal{KL}$-function (decaying to zero as $k \to \infty$), and $\gamma(s) = \frac{1}{1-q}s$, which is a $\mathcal{K}$-function, satisfying Eqn. 35. Therefore, the system is ISS.

However, there exists a steady-state error due to the disturbance $w(k)$. In the best case, when $\bar{A}(k)$ converges to $\bar{A}$ and $w(k)$ converges to $w$, the error $\bar{e}(k)$ eventually converges to a steady state given by

$$\bar{e}_{ss} = (I - \bar{A}(1-pI))^{-1}w.$$

Therefore, $\bar{e}_{ss} \neq 0$ if $w \neq 0$. $\qquad\square$

**Remark 1 (Convergence rate versus $K_p$.)** *From Ineq. 37, smaller $q$ yields faster convergence. Because*

$$q(p) = M|1-p| = \begin{cases} M(1-p), & p \in \left(1 - \frac{1}{M}, 1\right), \\ M(p-1), & p \in \left[1, 1 + \frac{1}{M}\right), \end{cases}$$

*we have $\frac{d}{dp}q(p) = -M < 0$ for $p < 1$ and $\frac{d}{dp}q(p) = M > 0$ for $p > 1$. Therefore the contraction factor $q(p)$ is minimized at*

$$p^\star = 1 \implies q^\star = 0,$$

*and increases as $p$ moves away from $1$ within the admissible interval.*

### B.5 PROPORTIONAL-INTEGRAL (PI) CONTROL

To reduce the steady-state error, the proportional controller is extended with an integral action, resulting in a proportional-integral (PI) control law:

$$u(k) = K_p \bar{e}(k) + K_i s(k), \qquad s(k+1) = s(k) + \bar{e}(k), \qquad (K_d = 0). \tag{38}$$

The dynamics Eqn. 20 then become

$$\bar{e}(k+1) = \bar{A}(k)(I - K_p)\bar{e}(k) - \bar{A}(k)K_i s(k) + w(k). \tag{39}$$

We use the following orthogonal decomposition for $w(k)$:

$$w(k) = w^\|(k) + w^\perp(k),$$

where $w^\|(k) \in \mathrm{Im}\,\bar{A}(k)$ and $w^\perp(k) \in (\mathrm{Im}\,\bar{A}(k))^\perp$.

The impact of $w^\|(k)$ on the error can be eliminated by PI control, as discussed below. On the other hand, P-only control is not able to do so, because keeping $\bar{e}(k) = 0$ requires $u(k) = 0$, leaving no component in $u(k)$ that can compensate for $w^\|(k)$.

Since $w^\|(k) \in \mathrm{Im}\,\bar{A}(k)$, it can be expressed as

$$w^\|(k) = \bar{A}(k)K_i s^*(k). \iff s^*(k) = K_i^{-1}\bar{A}(k)^\dagger w^\|(k)$$

Let $\tilde{s}(k) = s(k) - s^*(k)$ and $d(k) = s^*(k+1) - s^*(k)$. Therefore,

$$\tilde{s}(k+1) = \tilde{s}(k) + \bar{e}(k) - d(k) \tag{40}$$

Insert $\boldsymbol{s}(k) = \boldsymbol{s}^*(k) + \tilde{\boldsymbol{s}}(k)$ and $\boldsymbol{w}(k) = \boldsymbol{w}^{\parallel}(k) + \boldsymbol{w}^{\perp}(k) = \bar{\boldsymbol{A}}(k)\boldsymbol{K}_i s^*(k) + \boldsymbol{w}^{\perp}(k)$ into Eqn. 39,

$$
\begin{aligned}
\bar{\boldsymbol{e}}(k+1) &= \bar{\boldsymbol{A}}(k)(I - \boldsymbol{K}_p)\bar{\boldsymbol{e}}(k) - \bar{\boldsymbol{A}}(k)\boldsymbol{K}_i s^*(k) - \bar{\boldsymbol{A}}(k)\boldsymbol{K}_i \tilde{\boldsymbol{s}}(k) + \bar{\boldsymbol{A}}(k)\boldsymbol{K}_i s^*(k) + \boldsymbol{w}^{\perp}(k) \\
&= \bar{\boldsymbol{A}}(k)(I - \boldsymbol{K}_p)\bar{\boldsymbol{e}}(k) - \bar{\boldsymbol{A}}(k)\boldsymbol{K}_i \tilde{\boldsymbol{s}}(k) + \boldsymbol{w}^{\perp}(k)
\end{aligned}
\tag{41}
$$

We introduce the lifted state $\tilde{\zeta}_{\mathrm{PI}}(k) = \begin{bmatrix} \bar{\boldsymbol{e}}(k) \\ \tilde{\boldsymbol{s}}(k) \end{bmatrix}$ with its dynamic derived from Eqn. 40-41 as follow

$$
\tilde{\zeta}_{\mathrm{PI}}(k+1) = \boldsymbol{M}_i(t)\tilde{\zeta}_{\mathrm{PI}}(k) + \tilde{\boldsymbol{w}}_{\mathrm{PI}}(k),
\tag{42}
$$

where

$$
\boldsymbol{M}_i(k) = \begin{bmatrix} \boldsymbol{M}_p(k) & -\boldsymbol{G}(k) \\ I & I \end{bmatrix},
$$

with $\boldsymbol{M}_p(k) = \bar{\boldsymbol{A}}(p)(I - \boldsymbol{K}_p), \boldsymbol{G}(k) = \bar{\boldsymbol{A}}(k)\boldsymbol{K}_i$ and

$$
\tilde{\boldsymbol{w}}_{\mathrm{PI}}(k) = \begin{bmatrix} \boldsymbol{w}^{\perp}(k) \\ -\boldsymbol{d}(k) \end{bmatrix},
$$

**Proposition 3 (Stabilizing the PI loop reduces steady-state error)** *Let $\boldsymbol{M}_p(k) = \bar{\boldsymbol{A}}(k)(I - \boldsymbol{K}_p)$, and denote $\|\boldsymbol{K}_i\| =: h$. Assume $\sup_k \|\bar{\boldsymbol{A}}(k)\| \le M < \infty$ and $\sup_k \|\boldsymbol{M}_p(k)\| \le q < 1$. If $q + Mh < 1$, then the PI closed-loop control is ISS. Furthermore, the integral part exactly cancels the matched disturbance component $\boldsymbol{w}^{\parallel}$. The remaining error is due only to the unmatched component $\boldsymbol{w}^{\perp}$, which cannot be compensated. Full proof and term explanations provided in Appendix B.5*

**Proof.** Using the sub-multiplicativity of the induced matrix norm and the triangle inequality, and noting that $\|\boldsymbol{M}_p(k)\| \le q, \|\boldsymbol{G}(k)\| = \|\bar{\boldsymbol{A}}(k)\boldsymbol{K}_i\| \le \|\bar{\boldsymbol{A}}(k)\|\|\boldsymbol{K}_i\| \le Mh$, we obtain

$$
\begin{aligned}
\|\bar{\boldsymbol{e}}(k+1)\| &\le \|\boldsymbol{M}_p(k)\|\|\bar{\boldsymbol{e}}(k)\| + \|\boldsymbol{G}(k)\|\|\tilde{\boldsymbol{s}}(k)\| + \|\boldsymbol{w}^{\perp}(k)\| \\
&\le q\|\bar{\boldsymbol{e}}(k)\| + Mh\|\tilde{\boldsymbol{s}}(k)\| + \|\boldsymbol{w}^{\perp}(k)\|,
\end{aligned}
\tag{43}
$$

$$
\|\tilde{\boldsymbol{s}}(k+1)\| \le \|\bar{\boldsymbol{e}}(k)\| + \|\tilde{\boldsymbol{s}}(k)\| + \|d(k)\|.
\tag{44}
$$

Introduce

$$
z(k) := \begin{bmatrix} \|\bar{\boldsymbol{e}}(k)\| \\ \|\tilde{\boldsymbol{s}}(k)\| \end{bmatrix}, \qquad H := \begin{bmatrix} q & Mh \\ 1 & 1 \end{bmatrix}, \qquad v(k) := \begin{bmatrix} \|\boldsymbol{w}^{\perp}(k)\| \\ \|\boldsymbol{d}(k)\| \end{bmatrix}.
$$

Then Eqn. 43- 44 give the comparison system

$$
z(k+1) \le Hz(k) + v(k).
\tag{45}
$$

Expanding Eqn. 45 recursively yields

$$
z(k) \le H^k z(0) + \sum_{i=0}^{k-1} H^{k-1-i} v(i).
\tag{46}
$$

Consider the characteristic equation of $H$:

$$
(\lambda - q)(\lambda - 1) - Mh = 0 \iff \lambda^2 - (q+1)\lambda + (q + Mh) = 0.
$$

Since $q + Mh < 1$, the maximal root $\lambda^\star$ satisfies $\lambda^\star < 1$, hence the spectral radius $\rho(H) < 1$.

Let $r := \rho(H) < 1$ be the spectral radius of $H$. By the Gelfand formula for induced (operator) norms,

$$
\lim_{k \to \infty} \|H^k\|^{1/k} = r \qquad \text{(Horn \& Johnson, 2012, p. 349)}.
$$

Fix any $\rho \in (r, 1)$. Then, by the definition of the limit, there exists $N \in \mathbb{N}$ such that

$$
\|H^k\|^{1/k} \le \rho \quad \text{for all } k \ge N \implies \|H^k\| \le \rho^k \quad \forall k \ge N.
$$

Define the constant

$$
C := \max\left\{ 1, \max_{0 \le k \le N} \|H^k\| \rho^{-k} \right\}.
$$

Then:

- If $k \geq N$, we have $\|H^k\|\rho^{-k} \leq 1 \leq C$, hence $\|H^k\| \leq C\rho^k$.
- If $0 \leq k \leq N$, we have $\|H^k\|\rho^{-k} \leq \max_{0 \leq k \leq N}\|H^k\|\rho^{-k} \leq C$, hence $\|H^k\| \leq C\rho^k$.

Therefore,
$$\|H^k\| \leq C\rho^k \qquad \text{for all } k \geq 0. \tag{47}$$

Applying Eqn. 47 to Eqn. 46 gives
$$\begin{aligned}
\|z(k)\| &\leq \|H^k\|\|z(0)\| + \sum_{i=0}^{k-1}\|H^{k-1-i}\|\|v(i)\| \\
&\leq C\rho^k\|z(0)\| + C\sum_{i=0}^{k-1}\rho^{k-1-i}\|v(i)\| \\
&\leq C\rho^k\|z(0)\| + \frac{C}{1-\rho}\|v\|_\infty,
\end{aligned} \tag{48}$$

where $\|v\|_\infty := \sup_{i \geq 0}\|v(i)\|$.

By construction,
$$\|\tilde{\zeta}_{PI}(k)\| = \left\|\begin{bmatrix}\bar{e}(k) \\ \tilde{s}(k)\end{bmatrix}\right\| = \left(\|\bar{e}(k)\|^2 + \|\tilde{s}(k)\|^2\right)^{1/2} = \|z(k)\|.$$

Combining this identity with Eqn. 48, the ISS estimate follows with
$$\beta(s,k) := C\rho^k s \in \mathcal{KL}, \qquad \gamma(s) := \frac{C}{1-\rho}s \in \mathcal{K},$$

which proves that the PI closed loop Eqn. 42 is ISS. $\qquad\square$

The integral part exactly cancels the matched disturbance component $w^\|$. The remaining error is due only to the unmatched component $w^\perp$, which cannot be compensated, and to the variation rate $d(k)$ when $\bar{A}(k)$ and $w(k)$ change over time. In the best scenario, if a steady state exists, i.e., $\bar{A}(k) \to \bar{A}$ and $w(k) \to w$ with $w \in \text{Im}\bar{A}$, then $w^\perp \equiv 0$, $d \equiv 0$, and thus $\bar{e}(k) \to 0$.

**Remark 2 (Convergence rate versus $K_i$)** *From proposition 3, the convergence rate of $\tilde{\zeta}_{PI}(t)$ depends on $\rho$: the smaller $\rho$, the faster the convergence. We also adopt the convention (as in the proof) that $\rho \in (r,1)$. Equivalently, we examined $r(h) = \rho(H)$ and proved that with $h = \frac{(1-q)^2}{4M}$ this quantity is minimized.*

**Proof** Consider the characteristic polynomial of $H$:
$$\lambda^2 - (q+1)\lambda + (q+Mh) = 0.$$
Its discriminant is
$$\Delta(h) = (q+1)^2 - 4(q+Mh) = (q-1)^2 - 4Mh.$$

If $\Delta(h) \geq 0$ (i.e., $0 \leq h \leq \frac{(1-q)^2}{4M}$), then
$$r(h) = \frac{q+1+\sqrt{\Delta(h)}}{2},$$
and $r(h)$ decreases as $h$ increases.

If $\Delta(h) < 0$ (i.e., $\frac{(1-q)^2}{4M} < h < \frac{1-q}{M}$), then
$$r(h) = \sqrt{q+Mh},$$
and $r(h)$ decreases as $h$ decreases.

Hence $r(h)$ can achieve its best (smallest) value at
$$h = \frac{(1-q)^2}{4M}, \tag{49}$$

for which the error converges to zero the fastest. $\qquad\square$

Nevertheless, as we discuss in the next section, in some situations such a large value of $h$ may become a practical obstacle for PI control.

### B.6 OVESHOOT MECHANISM

**Phenomenon.** A common issue in standard PI settings is *overshooting*: the closed loop oscillates around the setpoint before settling (see Åström & Hägglund (1995a, Ch. 3, §3.3)). In our terms, the integral part accumulates past error and can push the output beyond the setpoint; subsequent sign changes of the error gradually "discharge" the integral, producing a decaying oscillation. The big overshoot is undesirable when we prefer a more stable response near zero. Below we analyze the same mechanism for our PI steering setting.

Citing an observation from Vu & Nguyen (2025): in the *absence* of steering, the cosine similarity between error vectors at different layers is consistently *positive*, i.e.,

$$\cos\angle\big(\bar{\boldsymbol{e}}(i),\bar{\boldsymbol{e}}(j)\big)>0 \quad \text{for all layers } i,j,$$

so the layerwise errors share (approximately) the same direction. Consequently, with $\{\bar{\boldsymbol{x}}^+(k)\}$ serving as the trajectory setpoints and $\{\bar{\boldsymbol{x}}^-(k)\}$ the system output, an *overshoot event* occurs when the instantaneous error reverses its initial orientation, namely when

$$\langle\bar{\boldsymbol{e}}(k),\bar{\boldsymbol{e}}(0)\rangle<0.$$

We now introduce the definitions used below.

**Scalarization along a direction.** Let

$$v := \frac{\bar{\boldsymbol{e}}(0)}{\|\bar{\boldsymbol{e}}(0)\|} \qquad \text{and project onto } v: \qquad \boldsymbol{e}_v(k):=v^\top\bar{\boldsymbol{e}}(k), \quad \boldsymbol{s}_v(k):=v^\top\tilde{\boldsymbol{s}}(k).$$

From the PI loop dynamics Eqn. 42 we obtain the scalar PI pair

$$\boldsymbol{e}_v(k+1)=a(k)\boldsymbol{e}_v(k) - b(k)\boldsymbol{s}_v(k) + \boldsymbol{w}_v^\perp(k), \tag{50}$$

$$\boldsymbol{s}_v(k+1)=\boldsymbol{s}_v(k) + \boldsymbol{e}_v(k) - \boldsymbol{d}_v(k), \tag{51}$$

with $a(k) = v^\top\bar{\boldsymbol{A}}(k)(I - \boldsymbol{K}_p)v = v^\top\boldsymbol{M}_p(k)v$, $b(k) = v^\top\bar{\boldsymbol{A}}(k)\boldsymbol{K}_i v = v^\top\boldsymbol{G}(k)v$, and projected disturbances $\boldsymbol{w}_v^\perp(k) := v^\top\boldsymbol{w}^\perp(k)$, $\boldsymbol{d}_v(k) := v^\top\boldsymbol{d}(k)$. Empirically (and consistently with the angular-steering observation in our setup), we have $v^\top\bar{\boldsymbol{A}}(k)v \geq 0$ for all $k$; together with the gain $\boldsymbol{K}_i=hI$ with $h\geq 0$, this implies $b(k)\geq 0$.

Also, the assumption $q:=\sup_k\|\boldsymbol{M}_p(k)\|$ and $M:=\sup_k\|\bar{\boldsymbol{A}}(k)\|$ yeilds

$$a(k)\leq q<1, \qquad 0\leq b(k)\leq Mh, \tag{52}$$

Since the system Eqn. 42 is ISS, so is the system Eqn. 50-Eqn. 51. In other words, both $\boldsymbol{e}_v(k)$ and $\boldsymbol{s}_v(k)$ decay. Recall that

$$\boldsymbol{s}_v(k)=\sum_{i=0}^{k-1}\boldsymbol{e}_v(i).$$

Hence, $\boldsymbol{s}_v(k)$ can only decrease when $\boldsymbol{e}_v(k) < 0$, which is precisely the moment when overshoot occurs. These overshooting and decaying phenomena are observed in empirical simulation, see Fig. 7. Below, we define the overshoot in our setting.

**Definition 2 (Overshoot and its amplitude)** *We say an* overshoot *occurs from time $k_a$ to $k_a+m$ if $\boldsymbol{e}_v(k)<0 \,\forall k=k_a,k_a+1,...,k_a+m-1$ and $\boldsymbol{e}_v(k)\geq 0$ for $k=k_a-1,k_a+m$. Its amplitude is defined as*

$$A_a := \max_{kt_a\leq i\leq k_a+m-1}|\boldsymbol{e}_v(i)|. \tag{53}$$

In standard PID settings illustrated in Åström & Hägglund (1995a, Ch. 3, §3.3)), it is observed that the overshoot amplitude decays over time. This decay is also consistent with the ISS property of the closed loop: as both $\boldsymbol{e}_v(t)$ and $\boldsymbol{s}_v(t)$ are driven down, subsequent oscillations tend to diminish in magnitude. In our simulation (see Fig. 7), the first overshoot appear to be representative. Hence, while we are not yet able to provide a formal proof, the empirical evidence and ISS intuition justify the first overshoot which is typically the dominant one and serves as a representative indicator of oscillatory behavior.

This assumption is for Proposition 4. Suppose that $a(t) \geq 0$; equivalently, $p \in (1 - \frac{1}{M},1]$, which is the result of proposition 1. This assumption is expected to entail no loss of generality relative to the $|a(t)|\leq q<1$ assumption.

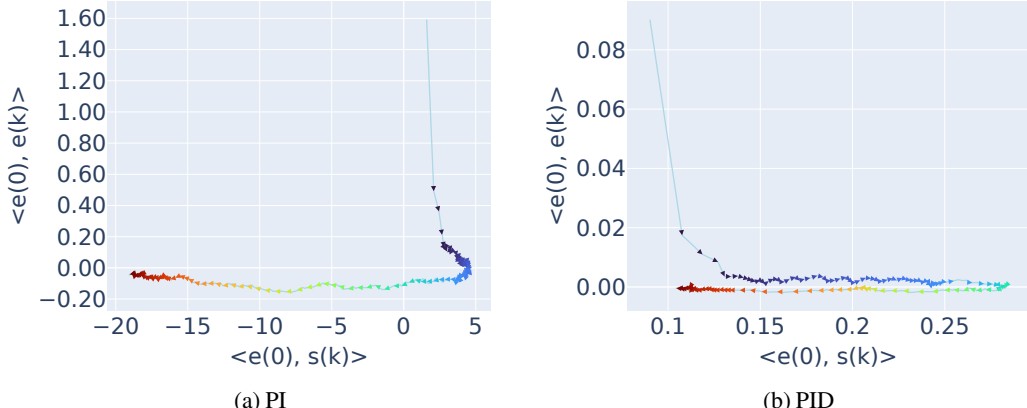

(a) PI                    (b) PID

Figure 7: Scalar errors across time step of randomly initialized model after applying PI and PID controller. Colors from blue to red denote the time (layer) dimension.

**Proposition 4 (Agressive PI gain leads to a large first overshoot)** *Let $k_0$ be the first sign-change time, i.e.,*

$$\boldsymbol{e_v}(j) \geq 0 \;\; \forall j = 0, 1, ..., k_0 - 1, \qquad \boldsymbol{e_v}(k_0) < 0,$$

*and let $k_1$ be the first time the trajectory returns to the nonnegative side,*

$$\boldsymbol{e_v}(k_1) \geq 0, \qquad \boldsymbol{e_v}(i) < 0 \;\; \forall i = k_0, k_0 + 1, ..., k_1 - 1.$$

*As in Eqn.53, the first overshoot amplitude is*

$$A_0 = \max_{k_0 \leq i \leq k_1 - 1} |\boldsymbol{e_v}(i)| = |\boldsymbol{e_v}(i_{max})|. \tag{54}$$

*Assume $\sup_k \|\bar{\boldsymbol{A}}(k)\| \leq M < \infty$ and $\sup_k \|\boldsymbol{M}_p(k)\| \leq q < 1$. Denote $\|\boldsymbol{K}_i\| =: h$. and given $q + Mh < 1$. Therefore,*

$$A_0 \leq Mh\Big(\frac{1}{1-q} + \frac{1}{(1-q)^2}\Big)\boldsymbol{e_v}(0) + \Big(\frac{Mh}{1-q}(k_0-1) + \frac{Mh}{1-q}\Big)\|\boldsymbol{d}\|_\infty + \frac{Mh(k_0-1)+1}{1-q}\|\boldsymbol{w}\|_\infty; \tag{55}$$

**Proof.** Before the first crossing ($j \leq k_0 - 1$) we have $\boldsymbol{s}_v(j) \geq 0$, hence from Eqn. 50

$$\boldsymbol{e}_v(j+1) = a(j)\boldsymbol{e}_v(j) - b(j)\boldsymbol{s}_v(j) + \boldsymbol{w}_v^\perp(j) \leq a(j)\boldsymbol{e}_v(j) + \boldsymbol{w}_v^\perp(j) \leq q\boldsymbol{e}_v(j) + \|\boldsymbol{w}\|_\infty,$$

so by induction $\boldsymbol{e}_v(j) \leq q^j \boldsymbol{e}_v(0) + \frac{1}{1-q}\|\boldsymbol{w}\|_\infty$. Summing Eqn. 51,

$$\boldsymbol{s}_v(k_0-1) = \sum_{i=0}^{k_0-2} \boldsymbol{e}_v(i) - \sum_{i=0}^{k_0-2} \boldsymbol{d}_v(i) \leq \frac{\boldsymbol{e}_v(0)}{1-q} + (k_0-1)\|\boldsymbol{d}\|_\infty + \frac{k_0-1}{1-q}\|\boldsymbol{w}\|_\infty. \tag{56}$$

Since $a(k) \geq 0$ and $\boldsymbol{e}_v(k_0-1) \geq 0$, at the crossing step,

$$\begin{aligned}
|\boldsymbol{e}_v(k_0)| = -\boldsymbol{e}_v(k_0) &\leq b(k_0-1)\boldsymbol{s}_v(k_0-1) + \|\boldsymbol{w}\|_\infty \\
&\leq Mh\Big(\frac{\boldsymbol{e}_v(0)}{1-q} + (k_0-1)\|\boldsymbol{d}\|_\infty + \frac{k_0-1}{1-q}W_\infty\Big) + \|\boldsymbol{w}\|_\infty. \\
&= \frac{Mh}{1-q}\boldsymbol{e}_v(0) + Mh(k_0-1)\|\boldsymbol{d}\|_\infty + (Mh\frac{k_0-1}{1-q}+1)\|\boldsymbol{w}\|_\infty
\end{aligned} \tag{57}$$

Assume that $\boldsymbol{d}_v(k)$ is small enough s.t during the overshoot time, $\boldsymbol{s}_v$ is nonincreasing (since $\boldsymbol{e}_v < 0$ a.e. on $[k_0, k_1-1]$), so $\boldsymbol{s}_v(i) \leq \boldsymbol{s}_v(k_0-1)$ for $i \in [k_0, k_1-1]$. Using Eqn. 50 again and unrolling $m$ steps from $k_0$,

$$\begin{aligned}
|\boldsymbol{e}_v(k_0+m)| &\leq q^m |\boldsymbol{e}_v(k_0)| + \sum_{k=0}^{m-1} q^k \big(Mh\boldsymbol{s}_v(k_0-1) + \|\boldsymbol{w}\|_\infty\big) \\
&\leq |\boldsymbol{e}_v(k_0)| + \frac{Mh\boldsymbol{s}_v(k_0-1) + \|\boldsymbol{w}\|_\infty}{1-q}.
\end{aligned} \tag{58}$$

Taking the maximum over $m \in \{0, 1, ..., k_1 - k_0\}$ and substituting Ineq. 56 and Ineq. 57 into Ineq. 58 yields

$$A_0 \leq Mh\Big(\frac{1}{1-q} + \frac{1}{(1-q)^2}\Big)e_v(0) + \Big(\frac{Mh}{1-q}(k_0-1) + \frac{Mh}{1-q}\Big)\|d\|_\infty + \frac{Mh(k_0-1)+1}{1-q}\|w\|_\infty \tag{59}$$

$$\square$$

Consequently, the right-hand side of Ineq.59 is *monotone increasing in $h$* (via the factor $Mh$) and increases as $q$ decreases (through the factors $\frac{1}{1-q}$). In particular, more aggressive PI leads to a larger first-overshoot amplitude.

**Remark 3 ("Fast-PI" specialization.)** *With the tuning used in our analysis in remark 2, $h = \frac{(1-q)^2}{4M}$ (the value that minimizes the comparison-system rate), so $Mh = \frac{(1-q)^2}{4}$. Plugging into equation 59 gives*

$$A_0 \leq \Big(\frac{1-q}{4} + \frac{1}{4}\Big)e_v(0) + \Big(\frac{1-q}{4}(k_0-1) + \frac{1}{4}\Big)\|d\|_\infty + \frac{k_0}{1-q}\|w\|_\infty.$$

*In particular, in the disturbance-free case ($\|w\|_\infty = \|d\|_\infty = 0$) we obtain*

$$A_0 \leq \Big(\frac{1-q}{4} + \frac{1}{4}\Big)e_v(0),$$

*so stronger proportional action (smaller $q$) comes with a larger first-overshoot envelope, even though the closed-loop settles faster.*

## B.7 PID CONTROL

### B.7.1 STABILITY OF PID CLOSED LOOP

We consider the PID update

$$u(k) = K_p\bar{e}(k) + K_i s(k) + K_d\big(\bar{e}(k) - \bar{e}(k-1)\big),$$

and define the auxiliary matrices

$$M_p(k) := \bar{A}(k)\big(I - K_p\big), \qquad G(k) := \bar{A}(k)K_i, \qquad H(k) := \bar{A}(k)K_d,$$

together with the error increment

$$\Delta\bar{e}(k) := \bar{e}(k) - \bar{e}(k-1), \qquad \Delta\bar{e}(-1) = 0$$

Using the plant relation, we obtain

$$\Delta\bar{e}(k+1) = \big(M_p(k) - I\big)\bar{e}(k) - G(k)\tilde{s}(k) - H(k)\Delta\bar{e}(k) + w^\perp(k), \tag{60}$$

Introduce the lifted state from the auxiliary PI state in Eqn. 42

$$\tilde{\zeta}_{\mathrm{PID}}(k) := \begin{bmatrix} \bar{e}(k) \\ \tilde{s}(k) \\ \Delta\bar{e}(k) \end{bmatrix},$$

Then the closed-loop evolution reads

$$\tilde{\zeta}_{\mathrm{PID}}(k+1) = M_d(k)\tilde{\zeta}_{\mathrm{PID}}(k) + \tilde{w}_{\mathrm{PID}}(k), \tag{61}$$

where

$$M_d(k) := \begin{bmatrix} M_p(k) & -G(k) & -H(k) \\ I & I & 0 \\ M_p(k)-I & -G(k) & -H(k) \end{bmatrix}, \qquad \tilde{w}_{\mathrm{PID}}(k) := \begin{bmatrix} w^\perp(k) \\ -d(k) \\ w^\perp(k) \end{bmatrix}.$$

**Theorem 1 (Stabilizing the PID loop preserves bias removal)** *Let $M_p(k) = \bar{A}(k)(I - K_p)$, and denote $\|K_i\| =: h$, $\|K_d\| =: \ell$. Assume $\sup_k\|\bar{A}(k)\| \leq M < \infty$ and $\sup_k\|M_p(k)\| \leq q < 1$. If $q + Mh < 1$ (stable PI loop), then there exists $\ell > 0$ such that the PID closed-loop control is ISS. Therefore, the integral part in PID design still cancels the matched disturbance component $w^\|$.*

**Proof.** We establish the ISS for system 61 using the method of ISS-Lyapunov function, see (Jiang et al., 1999, Def. 2.2, Prop. 2.3). It then suffices to construct a candidate ISS-Lyapunov function $\boldsymbol{V}_{\mathrm{PID}}(k)$ satisfying that there exist class $\mathcal{K}_\infty$ functions $\alpha_1, \alpha_2, \alpha_3$ and a class $\mathcal{K}$ function $\sigma$ such that

$$\alpha_1(\|\tilde{\zeta}_{\mathrm{PID}}(k)\|) \leq \boldsymbol{V}_{\mathrm{PID}}(\tilde{\zeta}_{\mathrm{PID}}(k)) \leq \alpha_2(\|\tilde{\zeta}_{\mathrm{PID}}(k)\|), \tag{62}$$

and

$$\boldsymbol{V}(\tilde{\zeta}_{\mathrm{PID}}(k+1)) - \boldsymbol{V}(\tilde{\zeta}_{\mathrm{PID}}(k)) \leq -\alpha_3(\|\tilde{\zeta}_{\mathrm{PID}}(k)\|) + \sigma(\|\boldsymbol{w}\|). \tag{63}$$

Step 1: Candidtate $\boldsymbol{V}_{PID}(k)$ and PI-closed loop baseline

Define

$$\boldsymbol{V}_{\mathrm{PID}}(k) := \boldsymbol{V}_{\mathrm{PI}}(\tilde{\zeta}_{\mathrm{PI}}(k), k) + r\|\Delta\bar{e}(k)\|^2, \qquad r > 0, \tag{64}$$

where $\boldsymbol{V}_{\mathrm{PI}}(\tilde{\zeta}_{\mathrm{PI}}, k) = \tilde{\zeta}_{\mathrm{PI}}^\top \boldsymbol{P}(k)\tilde{\zeta}_{\mathrm{PI}}$ with $\boldsymbol{P}(k) = \boldsymbol{P}(k)^\top \succ 0$ and there exists some $\mu_{\mathrm{PI}} > 0$ such that

$$\boldsymbol{M}_i(k)^\top \boldsymbol{P}(k+1)\boldsymbol{M}_i(k) - \boldsymbol{P}(k) \leq -\mu_{\mathrm{PI}}I, \qquad \forall k, \tag{65}$$

Regarding the existence of such $\boldsymbol{V}_{\mathrm{PI}}$, recall the homogeneous PI-loop $\tilde{\zeta}_{\mathrm{PI}}(k+1) = \boldsymbol{M}_i(k)\tilde{\zeta}_{\mathrm{PI}}(k)$ with $\boldsymbol{M}_i(k) = \begin{bmatrix} \boldsymbol{M}_p(k) & -\boldsymbol{G}(k) \\ I & I \end{bmatrix}$, being asymptotically stable. Suppose there is $\boldsymbol{Q}(k) = \boldsymbol{Q}(k)^\top \succeq 0$ bounded so that the pair $(\boldsymbol{M}_i(k), \sqrt{\boldsymbol{Q}(k)})$ is observable for all $k$, hence the difference Lyapunov equation

$$\boldsymbol{M}_i^\top(k)\boldsymbol{P}(k+1)\boldsymbol{M}_i(k) - \boldsymbol{P}(k) = -\boldsymbol{Q}(k)$$

admits a unique positive definite solution $\boldsymbol{P}(k) = \boldsymbol{P}^\top(k) \succ 0$ for all $k$, and a uniform bound $\|\boldsymbol{P}\|_\infty := \sup_k \|\boldsymbol{P}(k)\| < \infty$ (see (Gajic & Qureshi, 2008, Ch. 4, p. 110))

Step 2: Condition (i) as in Eqn. 62

We write $\boldsymbol{V}_{\mathrm{PID}}$ as a quadratic form

$$\boldsymbol{V}_{\mathrm{PID}}(k) = \tilde{\zeta}_{\mathrm{PID}}(k)^\top \boldsymbol{P}_*(k)\tilde{\zeta}_{\mathrm{PID}}(k), \qquad \boldsymbol{P}_*(k) := \begin{bmatrix} \boldsymbol{P}(k) & 0 \\ 0 & rI \end{bmatrix}. \tag{66}$$

Clearly $\boldsymbol{P}_*(k) = \boldsymbol{P}_*(k)^\top$ and $\boldsymbol{P}_*(k) \succ 0$ because $\boldsymbol{P}(k) \succ 0$ and $rI \succ 0$; hence $\boldsymbol{P}_*(k)$ is symmetric positive definite for all $k$.

By the spectral theorem, there exists an orthogonal matrix $U_*(k)$ and a diagonal $\Lambda_*(k) = \mathrm{diag}(\lambda_1(k), \ldots, \lambda_{n_*}(k))$ with positive entries such that $\boldsymbol{P}_*(k) = U_*(k)\Lambda_*(k)U_*(k)^\top$. Moreover, because $\boldsymbol{P}_*(k)$ is block diagonal, its eigenvalues are precisely the union of the eigenvalues of $\boldsymbol{P}(k)$ and the repeated eigenvalue $r$. Using the uniform bounds already established for $\boldsymbol{P}(k)$ (there exists $\underline{\lambda} > 0$ with $\lambda_{\min}(\boldsymbol{P}(k)) \geq \underline{\lambda}$ and $\lambda_{\max}(\boldsymbol{P}(k)) \leq \|\boldsymbol{P}\|_\infty := \sup_k \|\boldsymbol{P}(k)\| < \infty$), we obtain the $k$-independent bounds

$$\lambda_{\min}(\boldsymbol{P}_*(k)) \geq \underline{\lambda}_* := \min\{\underline{\lambda}, r\} > 0, \qquad \lambda_{\max}(\boldsymbol{P}_*(k)) \leq \overline{\lambda}_* := \max\{\|\boldsymbol{P}\|_\infty, r\} < \infty.$$

For every vector $z$ and every symmetric positive definite $M$, $\lambda_{\min}(M)\|z\|^2 \leq z^\top M z \leq \lambda_{\max}(M)\|z\|^2$. Applying this to $M = \boldsymbol{P}_*(k)$ and $z = \tilde{\zeta}_{\mathrm{PID}}(k)$ in Eqn. 66 yields

$$\underline{\lambda}_*\|\tilde{\zeta}_{\mathrm{PID}}(k)\|^2 \leq \boldsymbol{V}_{\mathrm{PID}}(k) \leq \overline{\lambda}_*\|\tilde{\zeta}_{\mathrm{PID}}(k)\|^2.$$

Therefore, choosing the class-$\mathcal{K}_\infty$ functions

$$\alpha_1(\boldsymbol{s}) := \underline{\lambda}_*\boldsymbol{s}^2, \qquad \alpha_2(\boldsymbol{s}) := \overline{\lambda}_*\boldsymbol{s}^2,$$

we obtain the desired bound

$$\alpha_1\left(\|\tilde{\zeta}_{\mathrm{PID}}(k)\|\right) \leq \boldsymbol{V}_{\mathrm{PID}}(k) \leq \alpha_2\left(\|\tilde{\zeta}_{\mathrm{PID}}(k)\|\right), \tag{67}$$

which establishes condition (i) in 62.

Step 3: Condition (ii) as in Eqn. 63.

From Eqn. 64,

$$\Delta\boldsymbol{V}_{\mathrm{PID}}(k) = \underbrace{\Delta\boldsymbol{V}_{\mathrm{PI}}\left(\tilde{\zeta}_{\mathrm{PI}}(k)\right)\Big|_{\mathrm{PID}}}_{\text{PI part under PID update}} + r\left(\|\Delta\bar{e}(k+1)\|^2 - \|\Delta\bar{e}(k)\|^2\right). \tag{68}$$

*Bounding the PI part under the PID update.*

Under PI rule ($\boldsymbol{K}_d = 0$),

$$\tilde{\zeta}_{\mathrm{PI}}(k+1)\big|_{\mathrm{PI}} = \boldsymbol{M}_i(k)\tilde{\zeta}_{\mathrm{PI}}(k) + \tilde{\boldsymbol{w}}_{\mathrm{PI}}(k), \qquad \boldsymbol{M}_i(k) := \begin{bmatrix} \boldsymbol{M}_p(k) & -\boldsymbol{G}(k) \\ I & I \end{bmatrix}$$

then

$$\Delta \boldsymbol{V}_{\mathrm{PI}}\big(\tilde{\zeta}_{\mathrm{PI}}(k)\big)\big|_{\mathrm{PI}} = \tilde{\zeta}_{\mathrm{PI}}(k)^\top \big(\boldsymbol{M}_i(k)^\top \boldsymbol{P}(k+1)\boldsymbol{M}_i(k) - \boldsymbol{P}(k)\big)\tilde{\zeta}_{\mathrm{PI}}(k)$$
$$+ 2\tilde{\zeta}_{\mathrm{PI}}(k)^\top \boldsymbol{M}_i(k)^\top \boldsymbol{P}(k+1)\tilde{\boldsymbol{w}}_{\mathrm{PI}}(k) + \tilde{\boldsymbol{w}}_{\mathrm{PI}}(k)^\top \boldsymbol{P}(k+1)\tilde{\boldsymbol{w}}_{\mathrm{PI}}(k)$$

By Eqn. 65, Cauchy-Schwarz inequality and Young's inequality,

$$\Delta \boldsymbol{V}_{\mathrm{PI}}\big(\tilde{\zeta}_{\mathrm{PI}}(k)\big)\big|_{\mathrm{PI}} \leq -\mu_{\mathrm{PI}}\|\tilde{\zeta}_{\mathrm{PI}}(k)\|^2 + 2\|\boldsymbol{M}_i\|_\infty\|\boldsymbol{P}\|_\infty\|\tilde{\zeta}_{\mathrm{PI}}(k)\|\|\tilde{\boldsymbol{w}}_{\mathrm{PI}}(k)\| + \|\boldsymbol{P}\|_\infty\|\tilde{\boldsymbol{w}}_{\mathrm{PI}}(k)\|^2$$

$$\leq -(\mu_{\mathrm{PI}} - \varepsilon_1)\|\tilde{\zeta}_{\mathrm{PI}}(k)\|^2 + \Big(\frac{\|\boldsymbol{M}_i\|_\infty^2\|\boldsymbol{P}\|_\infty^2}{\varepsilon_1} + \|\boldsymbol{P}\|_\infty\Big)\|\tilde{\boldsymbol{w}}_{\mathrm{PI}}(k)\|^2$$

$$= -\mu_{\mathrm{PI}}^*\|\tilde{\zeta}_{\mathrm{PI}}(k)\|^2 + C_1\|\tilde{\boldsymbol{w}}_{\mathrm{PI}}(k)\|^2,$$

for any $\varepsilon_1 > 0$.

Under PID rule ($\boldsymbol{K}_d \neq 0$),

$$\tilde{\zeta}_{\mathrm{PI}}(k+1)\big|_{\mathrm{PID}} = \boldsymbol{M}_i(k)\tilde{\zeta}_{\mathrm{PI}}(k) - \delta(k) + \boldsymbol{w}_{\mathrm{PI}}(k),$$

with the "perturbation"

$$\delta(k) := \begin{bmatrix} H(k)\Delta\bar{\boldsymbol{e}}(k) \\ 0 \end{bmatrix}$$

Hence,

$$\Delta \boldsymbol{V}_{\mathrm{PI}}\big(\tilde{\zeta}_{\mathrm{PI}}(k)\big)\big|_{\mathrm{PID}} = \boldsymbol{V}_{\mathrm{PI}}\big(\tilde{\zeta}_{\mathrm{PI}}(k+1)\big|_{\mathrm{PI}}\big) - \boldsymbol{V}_{\mathrm{PI}}\big(\tilde{\zeta}_{\mathrm{PI}}(k)\big)$$
$$= \boldsymbol{V}_{\mathrm{PI}}\big(\tilde{\zeta}_{\mathrm{PI}}(k+1)\big|_{\mathrm{PI}}\big) - \boldsymbol{V}_{\mathrm{PI}}\big(\tilde{\zeta}_{\mathrm{PI}}(k)\big) + 2\big(\tilde{\zeta}_{\mathrm{PI}}(k+1)\big|_{\mathrm{PI}}\big)^\top \boldsymbol{P}\delta(k) + \delta(k)^\top \boldsymbol{P}\delta(k)$$
$$\tag{69}$$

Bounding each term in Eqn. 69

- $\boldsymbol{V}_{\mathrm{PI}}\big(\tilde{\zeta}_{\mathrm{PI}}(k+1)\big|_{\mathrm{PI}}\big) - \boldsymbol{V}_{\mathrm{PI}}\big(\tilde{\zeta}_{\mathrm{PI}}(k)\big) \leq -\mu_{\mathrm{PI}}^*\|\tilde{\zeta}_{\mathrm{PI}}(k)\|^2 + C_1\|\tilde{\boldsymbol{w}}_{\mathrm{PI}}(k)\|$
- Applying Young's inequality for inner product, there exists $\varepsilon > 0$ s.t

$$2\big(\tilde{\zeta}_{\mathrm{PI}}(k+1)\big|_{\mathrm{PI}}\big)^\top \boldsymbol{P}\delta(k) \leq \varepsilon\big\|\tilde{\zeta}_{\mathrm{PI}}(k+1)\big|_{\mathrm{PI}}\big\|^2 + \tfrac{1}{\varepsilon}\delta(k)^\top \boldsymbol{P}\delta(k) \tag{70}$$

$$\leq \varepsilon\|\boldsymbol{P}\|\big\|\tilde{\zeta}_{\mathrm{PI}}(k+1)\big|_{\mathrm{PI}}\big\|^2 + \frac{\|\boldsymbol{P}\|}{\varepsilon}M^2\ell^2\|\Delta\bar{\boldsymbol{e}}(k)\|^2 \tag{71}$$

Since $\big\|\tilde{\zeta}_{\mathrm{PI}}(k+1)\big|_{\mathrm{PI}}\big\|^2 \leq 2\|\boldsymbol{M}_i\|_\infty^2\|\tilde{\zeta}_{\mathrm{PI}}(k)\|^2 + 2\|\tilde{\boldsymbol{w}}_{\mathrm{PI}}(k)\|^2$

$$\Rightarrow 2\big(\tilde{\zeta}_{\mathrm{PI}}(k+1)\big|_{\mathrm{PI}}\big)^\top \boldsymbol{P}\delta(k) \leq 2\varepsilon\|\boldsymbol{P}\|\|\boldsymbol{M}_i\|_\infty^2\|\tilde{\zeta}_{\mathrm{PI}}(k)\|^2 + 2\varepsilon\|\boldsymbol{P}\|\|\tilde{\boldsymbol{w}}_{\mathrm{PI}}(k)\|^2$$
$$+ \frac{\|\boldsymbol{P}\|}{\varepsilon}M^2\ell^2\|\Delta\bar{\boldsymbol{e}}(k)\|^2$$
$$= 2\varepsilon\|\boldsymbol{P}\|\|\boldsymbol{M}_i\|_\infty^2\|\tilde{\zeta}_{\mathrm{PI}}(k)\|^2 + \frac{\|\boldsymbol{P}\|}{\varepsilon}M^2\ell^2\|\Delta\bar{\boldsymbol{e}}(k)\|^2$$
$$+ C_2\|\tilde{\boldsymbol{w}}_{\mathrm{PI}}(k)\|^2,$$

where $C_2 = 2\varepsilon\|\boldsymbol{P}\|$
- $\delta(k)^\top \boldsymbol{P}\delta(k) \leq \|\boldsymbol{P}\|\|\delta(k)\|^2 \leq \|\boldsymbol{P}\|M^2\ell^2\|\Delta\bar{\boldsymbol{e}}(k)\|^2$

Therefore,

$$\Delta \boldsymbol{V}_{\mathrm{PI}}\big(\tilde{\zeta}_{\mathrm{PI}}(k)\big)\big|_{\mathrm{PID}} \leq -\mu_{\mathrm{PI}}^*\|\tilde{\zeta}_{\mathrm{PI}}(k)\|^2 + C_1\|\tilde{\boldsymbol{w}}_{\mathrm{PI}}(k)\|^2$$
$$+ 2\varepsilon\|\boldsymbol{P}\|\|\boldsymbol{M}_i\|_\infty^2\|\tilde{\zeta}_{\mathrm{PI}}(k)\|^2 + \frac{\|\boldsymbol{P}\|}{\varepsilon}M^2\ell^2\|\Delta\bar{\boldsymbol{e}}(k)\|^2 + C_2\|\tilde{\boldsymbol{w}}_{\mathrm{PI}}(k)\|^2$$
$$+ \|\boldsymbol{P}\|M^2\ell^2\|\Delta\bar{\boldsymbol{e}}(k)\|^2 \tag{72}$$
$$= -(\mu_{\mathrm{PI}}^* - 2\varepsilon\|\boldsymbol{P}\|\|\boldsymbol{M}_i\|_\infty^2)\|\tilde{\zeta}_{\mathrm{PI}}(k)\|^2 + \|\boldsymbol{P}\|M^2\ell^2\Big(\frac{1}{\varepsilon} + 1\Big)\|\Delta\bar{\boldsymbol{e}}(k)\|^2$$
$$+ C_3\|\tilde{\boldsymbol{w}}_{\mathrm{PI}}(k)\|^2,$$

where $C_3 = C_1 + C_2$.

*Bounding the increment term.*

From Eqn. 60 and applying the inequality $(x+y+z)^2 \le 3(x^2+y^2+z^2)$,

$$\|\Delta\bar{e}(k+1)\|^2 \le 3\Big(\|M_p(k)-I\|_\infty^2 + Mh\Big)\|\tilde{\zeta}_{\mathrm{PI}}(k)\|^2 + 3M^2\ell^2\|\Delta\bar{e}(k)\|^2 + 3\|\tilde{w}_{\mathrm{PID}}(k)\|^2,$$

and so

$$r\big(\|\Delta\bar{e}(k+1)\|^2 - \|\Delta\bar{e}(k)\|^2\big) \le 3r\Big(\|M_p(k)-I\|_\infty^2 + Mh\Big)\|\tilde{\zeta}_{\mathrm{PI}}(k)\|^2 \\ - r\big(1-3M^2\ell^2\big)\|\Delta\bar{e}(k)\|^2 + 3r\|\tilde{w}_{\mathrm{PID}}(k)\|^2. \tag{73}$$

*Combination*

Combining Eqn. 68, Ineq. 72, and Ineq. 73,

$$\Delta V_{\mathrm{PID}}(k) \le -\Big(\mu_{\mathrm{PI}}^* - 2\varepsilon\|P\|_\infty\|M_i\|_\infty^2 - 3r\big(\|M_p(k)-I\|_\infty^2 + Mh\big)\Big)\|\tilde{\zeta}_{\mathrm{PI}}(k)\|^2 \\ - \Big(r(1-3M^2\ell^2) - \|P\|_\infty M^2\ell^2\big(\tfrac{1}{\varepsilon}+1\big)\Big)\|\Delta\bar{e}(k)\|^2 + C\|\tilde{w}_{\mathrm{PI}}(k)\|^2,$$

where $C = C_3 + r$. Define

$$S(r,\varepsilon) := \mu_{\mathrm{PI}}^* - 2\varepsilon\|P\|_\infty\|M_i\|_\infty^2 - 3r\big(\|M_p(k)-I\|_\infty^2 + Mh\big), \tag{74}$$

$$T(r,\varepsilon,\ell) := r(1-3M^2\ell^2) - \|P\|_\infty M^2\ell^2\big(\tfrac{1}{\varepsilon}+1\big). \tag{75}$$

ISS of the PID loop follows if $S(r,\varepsilon) > 0$ and $T(r,\varepsilon,\ell) > 0$.

*Feasible choices.*

We are free to choose any $\varepsilon > 0$ and $r > 0$ such that $S(r,\varepsilon) > 0$. One convenient selection is

$$\varepsilon = \frac{\mu_{\mathrm{PI}}^*}{8\|P\|_\infty\|M_i\|_\infty^2}, \qquad r = \frac{\mu_{\mathrm{PI}}^*}{8\big(\|M_p(k)-I\|_\infty^2 + Mh\big)} \quad \Rightarrow \quad S(r,\varepsilon) = \tfrac{3}{8}\mu_{\mathrm{PI}}^* > 0.$$

With $\varepsilon, r$ fixed as above, pick $\ell > 0$ small enough to satisfy $T(r,\varepsilon,\ell) > 0$, namely

$$\ell^2 < \frac{r}{\big(\|P\|_\infty\big(\tfrac{1}{\varepsilon}+1\big) + 3r\big)M^2},$$

Under these choices, $\Delta V_{\mathrm{PID}}(k) \le -\alpha_3\|\zeta_{\mathrm{PI}}(k)\|^2 - \alpha_4\|\Delta\bar{e}(k)\|^2 + \beta\|w_{\mathrm{PI}}(k)\|^2$ for some $\alpha_3, \alpha_4, \beta > 0$, which satisfies condition (ii) as in Eqn. 63 and proves ISS of the PID closed loop. $\qquad\square$

### B.7.2 OVERSHOOTING UNDER PID LAW OF CONTROL

Developing from Sec. B.6, we introduce scalar PID recursion along $v$:

$$e_v(k+1) = a(k)e_v(k) - b(k)s_v(k) - c(k)\Delta e_v(k) + w_v^\perp(k), \qquad s_v(k+1) = s_v(k) + e_v(k) - d_v(k), \tag{76}$$

where

$$a(k) := v^\top M_p(k)v, \qquad b(k) := v^\top G(k)v, \qquad c(k) := v^\top H(k)v, \\ \Delta e_v(k) = v^\top \Delta\bar{e}(k), \qquad w_v^\perp(k) = v^\top w^\perp(k).$$

By construction $a(k) \le q < 1$, $b(k) \le Mh$, $c(k) \le M\ell$ with $M := \sup_k \|\bar{A}(k)\|$, $h := \|K_i\|$ and $\ell := \|K_d\|$.

We now impose an additional requirement on the derivative gain $K_d$ so that, without the effect of noise, the PID update secures the monotonic decrease of $e_v(k)$ before the first negative peak of scalar error $e_v(k)$.

**Remark 4 (Pre-overshoot monotonic decrease of scalar errors)** *Assume the setting of Proposition 4 and further suppose the scalar error trajectory before the first largest overshoot under PID law is* smooth *in the sense that there exists $R \ge 1$ such that*

$$\frac{e_v(k-1)}{e_v(k)} \le R \qquad \text{for all } k = 1,2,\ldots,i_{max}-1, \tag{77}$$

where $\boldsymbol{A}_0 := \max_{k_0 \leq i \leq k_1} |\boldsymbol{e}_v(i)| = |\boldsymbol{e}_v(i_{max})|$ *from Eqn.54.*

*Assume that $\boldsymbol{w}_v^\perp \equiv 0$ and $\boldsymbol{d}_v \equiv 0$.*

*If, in addition, the derivative gain satisfies*

$$l = \|\boldsymbol{K}_d\| \leq \frac{1-q}{(R-1)M}, \tag{78}$$

*then under PID law*

$$\boldsymbol{e}_v(k+1) \leq \boldsymbol{e}_v(k) \qquad \text{for all } k = 0,1,...,i_{max}-1.$$

**Proof.** Before $k_0$, we have $\boldsymbol{e}_v(k) > 0$, so $\boldsymbol{s}_v(k) \geq 0$ (since $\boldsymbol{s}_v$ accumulates $\boldsymbol{e}_v$ and $\boldsymbol{s}_v(0) = 0$). For $k_0 \leq k \leq i_{max}-1$, $\boldsymbol{s}_v(k) \geq 0$ proved in remark 5

Hence

$$\begin{aligned}
\boldsymbol{e}_v(k+1) &= a(k)\boldsymbol{e}_v(k) - b(k)\boldsymbol{s}_v(k) - c(k)\big(\boldsymbol{e}_v(k) - \boldsymbol{e}_v(k-1)\big) \\
&\leq a(k)\boldsymbol{e}_v(k) + c(k)\big(\boldsymbol{e}_v(k-1) - \boldsymbol{e}_v(k)\big) \\
&\leq \big[a(k) + c(k)(R-1)\big]\boldsymbol{e}_v(k) \leq \big[q + (R-1)M\ell\big]\boldsymbol{e}_v(k) \leq \boldsymbol{e}_v(k),
\end{aligned}$$

where the last inequality is exactly Eqn. 78. $\qquad\square$

*Note for $R$:* In practice, one may estimate a conservative $R$ from PI-law traces and use a small safety factor

*Adding Disturbance:* With bounded disturbances, the scalar update reads

$$\boldsymbol{e}_v(k+1) \leq \big[q + (R-1)c_{\max}\big]\boldsymbol{e}_v(k) + |\boldsymbol{w}_v^\perp(k)|,$$

so the same one-step monotonicity conclusion holds whenever

$$|\boldsymbol{w}_v^\perp(k)| \leq \Big(1 - \big[q + (R-1)c_{\max}\big]\Big)\boldsymbol{e}_v(k) \quad \text{for all pre-first-largest-overshooting steps.}$$

If this smallness condition on disturbances fails at some step, one-step monotonicity may be lost, but the ISS bounds proved earlier still guarantee geometric decay up to a disturbance-dependent radius.

**Remark 5 (before the first negative peak, the integral state is positive)** *Assume the setting of Remark. 4. Hence,*

$$\boldsymbol{s}_v(k) > 0 \qquad \text{for all } k = k_0,...,i_{\max}-1.$$

**Proof.** We argue by contradiction. Suppose there exists the first $\tau \in [k_0, i_{\max}-1]$ such that $\boldsymbol{s}_v(\tau) \leq 0$. Then $\boldsymbol{s}_v(\tau-1) > 0$, and since we are on the first negative lobe, $\boldsymbol{e}_v(\tau) < 0$. Compute the one-step change of $\boldsymbol{e}_v$:

$$\begin{aligned}
\boldsymbol{e}_v(\tau+1) - \boldsymbol{e}_v(\tau) &= (a(\tau)-1)\boldsymbol{e}_v(\tau) - b(\tau)\boldsymbol{s}_v(\tau) \\
&= (1-a(\tau))|\boldsymbol{e}_v(\tau)| + b(\tau)(-\boldsymbol{s}_v(\tau)) > 0,
\end{aligned}$$

because $a(\tau) \leq q < 1$, $\boldsymbol{e}_v(\tau) < 0$ and $\boldsymbol{s}_v(\tau) \leq 0$. Hence $\boldsymbol{e}_v(\tau+1) > \boldsymbol{e}_v(\tau)$. By the same reasoning, as long as both $\boldsymbol{e}_v(k) < 0$ and $\boldsymbol{s}_v(k) \leq 0$ hold, we have

$$\boldsymbol{e}_v(k+1) - \boldsymbol{e}_v(k) \geq (1-q)|\boldsymbol{e}_v(k)| + b_{\min}(-\boldsymbol{s}_v(k)) > 0,$$

where $b_{\min} := \inf_k b(k) > 0$. Meanwhile $\boldsymbol{s}_v(k+1) = \boldsymbol{s}_v(k) + \boldsymbol{e}_v(k) \leq \boldsymbol{s}_v(k)$ on that interval, so $\boldsymbol{s}_v(k)$ is non-increasing; equivalently $-\boldsymbol{s}_v(k)$ is non-decreasing. If $\boldsymbol{e}_v$ stayed negative forever, then $\sum_{k=0}^N \boldsymbol{e}_v(\tau+k) \to -\infty$, so $-\boldsymbol{s}_v(k)$ would grow without bound and the increments $\boldsymbol{e}_v(k+1) - \boldsymbol{e}_v(k)$ would eventually be arbitrarily large, forcing $\boldsymbol{e}_v$ to cross 0 in finite time. This contradicts the choice of $i_{\max}$ as the first negative peak. Therefore such $\tau$ cannot exist and $\boldsymbol{s}_v(k) > 0$ for all $k = k_0,...,i_{\max}-1$. $\square$

**Theorem 2 (PID reduces the first-overshoot amplitude)** *Let the first overshoot occur at index $k_0$ with amplitude $A_0$ (definition in Eqn. 54). Then, the first-overshoot amplitude under PID Steering, $A_0^{\text{PID}}$, satisfies $A_0^{\text{PID}} \leq A_0^{\text{PI}}$, where $A_0^{\text{PI}}$ denotes the corresponding amplitude under PI Steering.*

**Proof.** Under the PI law we have

$$A_0^{\text{PI}} = -a(i_{max}-1)\boldsymbol{e}_v(i_{max}-1) + b(i_{max}-1)\boldsymbol{s}_v(i_{max}-1) - \boldsymbol{w}_v^\perp(i_{max}-1),$$

while under the PID law

$$A_0^{\mathrm{PID}} = -a(i_{max}-1)\boldsymbol{e}_v(i_{max}-1) + b(i_{max}-1)\boldsymbol{s}_v(i_{max}-1) + c(i_{max}-1)\Delta\boldsymbol{e}_v(i_{max}-1) \quad (79)$$
$$-\boldsymbol{w}_v^{\perp}(i_{max}-1), \quad (80)$$

Due to the monotone decrease before this first largest overshooting condition stated in the previous part and the fact that $c(k) > 0$, we have $c(k_0-1)\Delta\boldsymbol{e}_v(k_0-1) < 0$. Therefore,

$$A_0^{PID} \le b(i_{max}-1)\boldsymbol{s}_v(i_{max}-1) - a(i_{max}-1)\boldsymbol{e}_v(i_{max}-1) - \boldsymbol{w}_v^{\perp}(i_{max}-1) = A_0^{PI}. \quad (81)$$

$\square$

Remark 4 is neccessary because monotonic decrease of $e_v(t)$ before the first peak is both a key technical property for proving Theorem 2 and a desirable feature of PID control itself. Indeed, as noted by Åström & Hägglund (1995a, p.70), poorly tuned derivative gains may produce non-monotonicity, in which case reducing only the first overshoot does not translate into improved overall behavior.

### B.8 BEYOND LOCALLY LINEARIZED ACTIVATION DYNAMICS

In the main text, we analysed the averaged error dynamics under P/PI/PID steering by applying a first-order Taylor expansion of the activation map around the desired "plus" trajectory $x_i^+(k)$, leading to a linear time-varying (LTV) model with an additive disturbance term. In this appendix, we make explicit how the higher-order nonlinear terms enter the dynamics and discuss their impact on the closed-loop behaviour. We also justify why the perturbations

$$\delta_i(k) = -\boldsymbol{e}_i(k) + \boldsymbol{u}(k)$$

remain controlled in norm under the proposed controllers, even under strong steering.

#### B.8.1 NONLINEAR REMAINDER IN THE AVERAGED ERROR DYNAMICS

Recall that the minus activations after steering are given by

$$x_i^-(k) + \boldsymbol{u}(k) = x_i^+(k) + \delta_i(k), \qquad \delta_i(k) = -\boldsymbol{e}_i(k) + \boldsymbol{u}(k),$$

where $\boldsymbol{e}_i(k) = x_i^+(k) - x_i^-(k)$ denotes the layer-wise error for pair $i$, and $\boldsymbol{u}(k)$ is the steering control shared across pairs. We consider the exact Taylor expansion of the activation map $f_i^{(k)}$ around the desired activation $x_i^+(k)$:

$$f_i^{(k)}\big(x_i^+(k) + \delta_i(k)\big) = f_i^{(k)}\big(x_i^+(k)\big) + J_{f_i^{(k)}}\big(x_i^+(k)\big)\delta_i(k) + O\big(\|\delta_i(k)\|^2\big), \quad (82)$$

where $J_{f_i^{(k)}}(x)$ denotes the Jacobian of $f_i^{(k)}$ at $x$ and the $O(\cdot)$ term collects the higher-order Taylor remainder.

Averaging over $i$ and expressing everything in terms of the error variables leads to the following averaged error dynamics:

$$\bar{\boldsymbol{e}}(k+1) = \bar{\boldsymbol{A}}(k)\bar{\boldsymbol{e}}(k) - \bar{\boldsymbol{A}}(k)\boldsymbol{u}(k) + \boldsymbol{w}(k), \quad (83)$$

where

$$\bar{\boldsymbol{e}}(k) = \frac{1}{N}\sum_{i=1}^{N}\boldsymbol{e}_i(k), \qquad \tilde{\boldsymbol{e}}_i(k) = \boldsymbol{e}_i(k) - \bar{\boldsymbol{e}}(k), \quad (84)$$

$$\bar{\boldsymbol{A}}(k) = \frac{1}{N}\sum_{i=1}^{N}A_i(k) = \frac{1}{N}\sum_{i=1}^{N}J_{f_i^{(k)}}\big(x_i^+(k)\big), \quad (85)$$

$$\tilde{A}_i(k) = A_i(k) - \bar{\boldsymbol{A}}(k). \quad (86)$$

The disturbance term $\boldsymbol{w}(k)$ now has two contributions:

$$\boldsymbol{w}(k) = \frac{1}{N}\sum_{i=1}^{N}\tilde{A}_i(k)\tilde{\boldsymbol{e}}_i(k) + \frac{1}{N}\sum_{i=1}^{N}O\big(\|-\boldsymbol{e}_i(k) + \boldsymbol{u}(k)\|^2\big). \quad (87)$$

The first part is the "heterogeneity" term that already appears in the linearized analysis (capturing pair-to-pair variation in Jacobians and errors), while the second part arises purely from the higher-order Taylor remainder in Eqn. 82.

In other words, when we do not truncate the Taylor series after the linear term, the closed-loop dynamics still take the form

$$\bar{e}(k+1) = \bar{A}(k)\bar{e}(k) - \bar{A}(k)u(k) + w(k),$$

but $w(k)$ is augmented by the additional nonlinear contribution $\frac{1}{N}\sum_i O(\|-e_i(k) + u(k)\|^2)$.

From the perspective of input-to-state stability, this is a benign modification: all ISS results in the main text only require that the disturbance $w(k)$ is uniformly bounded. The higher-order terms in Eqn. 87 simply enter $w(k)$, and as long as $\|-e_i(k) + u(k)\|$ remains bounded and the activation maps $f_i^{(k)}$ have bounded higher derivatives along the trajectories of interest, these terms are also uniformly bounded across layers. In that case, all ISS statements and Lyapunov-based bounds derived in the main text continue to hold; the only effect is that the (conservative) upper bound on $\|\bar{e}(k)\|$ scales with the enlarged disturbance norm $\|w\|_\infty$.

### B.8.2 MAGNITUDE OF $\delta_i(k)$ UNDER P/PI/PID STEERING

A natural concern is whether $\|\delta_i(k)\| = \|-e_i(k) + u(k)\|$ may become large under strong steering, potentially making the higher-order terms in Eqn. 87 significant. We show here that, under our gain conditions, $\delta_i(k)$ primarily measures a residual mismatch that remains controlled in norm.

By definition,

$$\delta_i(k) = -e_i(k) + u(k) = -\left(x_i^+(k) - x_i^-(k)\right) + u(k), \tag{88}$$

so that

$$x_i^-(k) + u(k) = x_i^+(k) + \delta_i(k). \tag{89}$$

Thus, $\delta_i(k)$ measures the residual mismatch between the desired activation $x_i^+(k)$ and the steered activation $x_i^-(k) + u(k)$ at the same layer. When the controller functions as intended, the steered trajectory $x_i^-(k) + u(k)$ stays close to $x_i^+(k)$, and hence $\|\delta_i(k)\|$ remains small in norm. We make this more precise by examining $\delta_i(k)$ for the three controller classes.

Recall that we decompose the per-pair error into its mean and heterogeneous components,

$$e_i(k) = \bar{e}(k) + \tilde{e}_i(k), \qquad \bar{e}(k) = \frac{1}{N}\sum_{i=1}^{N} e_i(k), \qquad \tilde{e}_i(k) = e_i(k) - \bar{e}(k).$$

**(i) P control.** For P control we have

$$u(k) = K_p \bar{e}(k),$$

and therefore

$$\delta_i(k) = -e_i(k) + u(k) = -\bar{e}(k) - \tilde{e}_i(k) + K_p \bar{e}(k) \tag{90}$$
$$= (K_p - 1)\bar{e}(k) - \tilde{e}_i(k). \tag{91}$$

Our P-gain stability condition requires

$$1 - \frac{1}{M} < K_p < 1 + \frac{1}{M},$$

so $K_p - 1$ is necessarily small in norm. This cancels most of the contribution of $\bar{e}(k)$ in $\delta_i(k)$: in particular, the component of $\delta_i(k)$ along $\bar{e}(k)$ is scaled by $(K_p - 1)$, not by $K_p$. As a result, $\|\delta_i(k)\|$ is dominated by the heterogeneity term $\|\tilde{e}_i(k)\|$ rather than by the overall steering magnitude $\|\bar{e}(k)\|$.

**(ii) PI control.** For PI control the steering law is

$$u(k) = K_p \bar{e}(k) + K_i s(k), \qquad s(k) = \sum_{j=0}^{k-1} \bar{e}(j),$$

so

$$\delta_i(k) = -e_i(k) + u(k) \tag{92}$$
$$= -\bar{e}(k) - \tilde{e}_i(k) + K_p \bar{e}(k) + K_i s(k) \tag{93}$$
$$= (K_p - 1)\bar{e}(k) + K_i s(k) - \tilde{e}_i(k). \tag{94}$$

All contributions to $\delta_i(k)$ remain linear in $\bar{e}(\cdot)$ and its history through $s(k)$. The gain conditions derived in the main text ensure closed-loop stability and imply that

$$\bar{e}(k), \quad s(k)$$

remain uniformly bounded. Consequently, $\|u(k)\|$ remains in a comparable bounded range, and so does $\|\delta_i(k)\|$; the PI controller does not cause $\delta_i(k)$ to blow up, even under strong steering.

**(iii) PID control.** For PID control we add a derivative-like term,

$$\boldsymbol{u}(k) = \boldsymbol{K}_p\bar{\boldsymbol{e}}(k) + \boldsymbol{K}_i\boldsymbol{s}(k) + \boldsymbol{K}_d\Delta\bar{\boldsymbol{e}}(k), \qquad \Delta\bar{\boldsymbol{e}}(k) = \bar{\boldsymbol{e}}(k) - \bar{\boldsymbol{e}}(k-1),$$

which yields

$$\boldsymbol{\delta}_i(k) = -\boldsymbol{e}_i(k) + \boldsymbol{u}(k) \tag{95}$$

$$= -\bar{\boldsymbol{e}}(k) - \tilde{\boldsymbol{e}}_i(k) + \boldsymbol{K}_p\bar{\boldsymbol{e}}(k) + \boldsymbol{K}_i\boldsymbol{s}(k) + \boldsymbol{K}_d\Delta\bar{\boldsymbol{e}}(k) \tag{96}$$

$$= (\boldsymbol{K}_p - 1)\bar{\boldsymbol{e}}(k) + \boldsymbol{K}_i\boldsymbol{s}(k) + \boldsymbol{K}_d\Delta\bar{\boldsymbol{e}}(k) - \tilde{\boldsymbol{e}}_i(k). \tag{97}$$

Again, all terms are linear in $\bar{\boldsymbol{e}}(\cdot)$, $s(\cdot)$, and $\Delta\bar{\boldsymbol{e}}(\cdot)$. Under the PID gain conditions, the closed-loop system is stable and the signals

$$\bar{\boldsymbol{e}}(k), \quad \boldsymbol{s}(k), \quad \Delta\bar{\boldsymbol{e}}(k)$$

remain bounded in norm, implying bounded $\|\boldsymbol{u}(k)\|$ and consequently bounded $\|\boldsymbol{\delta}_i(k)\|$.

In summary, even when the raw error $\|\boldsymbol{e}_i(k)\|$ may become large under strong steering, the controller is designed to cancel the dominant component of $\bar{\boldsymbol{e}}(k)$ in $\boldsymbol{u}(k)$ and to keep the integral and derivative terms bounded. The residual mismatch $\boldsymbol{\delta}_i(k)$ is primarily governed by heterogeneity and bounded controller memory, rather than by the overall steering magnitude. As long as $\|\boldsymbol{\delta}_i(k)\|$ stays bounded, the higher-order nonlinear contributions in Eqn. 87 remain bounded as well, so the ISS guarantees derived from the locally linearized model continue to apply, with the disturbance norm $\|\boldsymbol{w}\|_\infty$ capturing both heterogeneity and curvature effects.

## C ADDITIONAL EXPERIMENTAL RESULTS

### C.1 QUALITATIVE EXAMPLES OF CONCEPT STEERING

Fig. 8 and 9 show that varying the intervention strength $\alpha \in [0,1]$ produces a smooth and controllable progression of stylistic traits in the generated images. At low strengths ($\alpha \approx 0.2$), subtle cues emerge, such as faint neon accents for the *cyberpunk* style or mild metallic shading for *steampunk*, while the overall image remains close to the original prompt. At moderate strengths ($\alpha \approx 0.5$), stylistic features become more salient: cyberpunk generations exhibit vivid neon lighting and futuristic cityscapes, whereas steampunk outputs show prominent brass textures, gears, and industrial motifs. Importantly, in this regime, the central semantic content of the prompt (i.e., objects, entities, and spatial composition) is preserved with high fidelity. At high intervention strengths ($\alpha \geq 0.8$), stylistic traits dominate the visual appearance, often saturating the scene with strong color palettes or dense textures, yet semantic alignment to the original prompt remains largely intact, indicating that the steering primarily affects style without eroding core content.

### C.2 JAILBREAKING LARGE LANGUAGE MODELS

Tab. 3 reports a comprehensive comparison of attack success rate (ASR) and general benchmark performance across multiple instruction-tuned models under different defense methods. Overall, PID consistently achieves the highest ASR among defenses, while maintaining comparable performance on downstream benchmarks.

We further compare our PID steering with recent steering-vector generation methods on a better safety-aligned variant of Gemma-9B-IT, namely Gemma2-9B-Instruct-With-Deeper-Safety-Alignment (Qi et al., 2025). Deeper safety alignment here trains the model to sustain refusal behavior beyond the first few tokens: instead of only shaping the opener, it conditions on partially harmful or misleading prefixes and optimizes the model to "recover" back to safe behavior later in the sequence. Practically, this extends safety pressure across positions so refusals remain stable under mild coercion, prefilling, or decoding variance, while seeking to preserve general-task utility. Under this strong safety-aligned regime, PID-based steering still achieves a non-trivial attack success rate and outperforms other state-of-the-art activation steering methods considered in our study.

### C.3 EMPIRICAL EVIDENCE FOR THE STABILITY INTERVAL

Proposition 4 and Remark 3 (Appendix B.6) show that the PI gains that maximise the asymptotic convergence rate of the linearised error dynamics (the "fastest convergence" choice of $K_p$ and $K_i$) also induce a large overshoot in the correlation trajectory $\langle\bar{e}(0), \bar{e}(k)\rangle$. Empirically, this manifests as poorer steering performance: aggressive integral action speeds up convergence but increases oscillation and overshoot, which is consistent with classical PI tuning principles (see, e.g., (Åström & Hägglund, 1995a)).

Theorem 2 further shows that the derivative term $K_d$ provides damping: for fixed $K_p$ and $K_i$, increasing $K_d$ reduces the overshoot of the error trajectory. Although we do not establish a formal

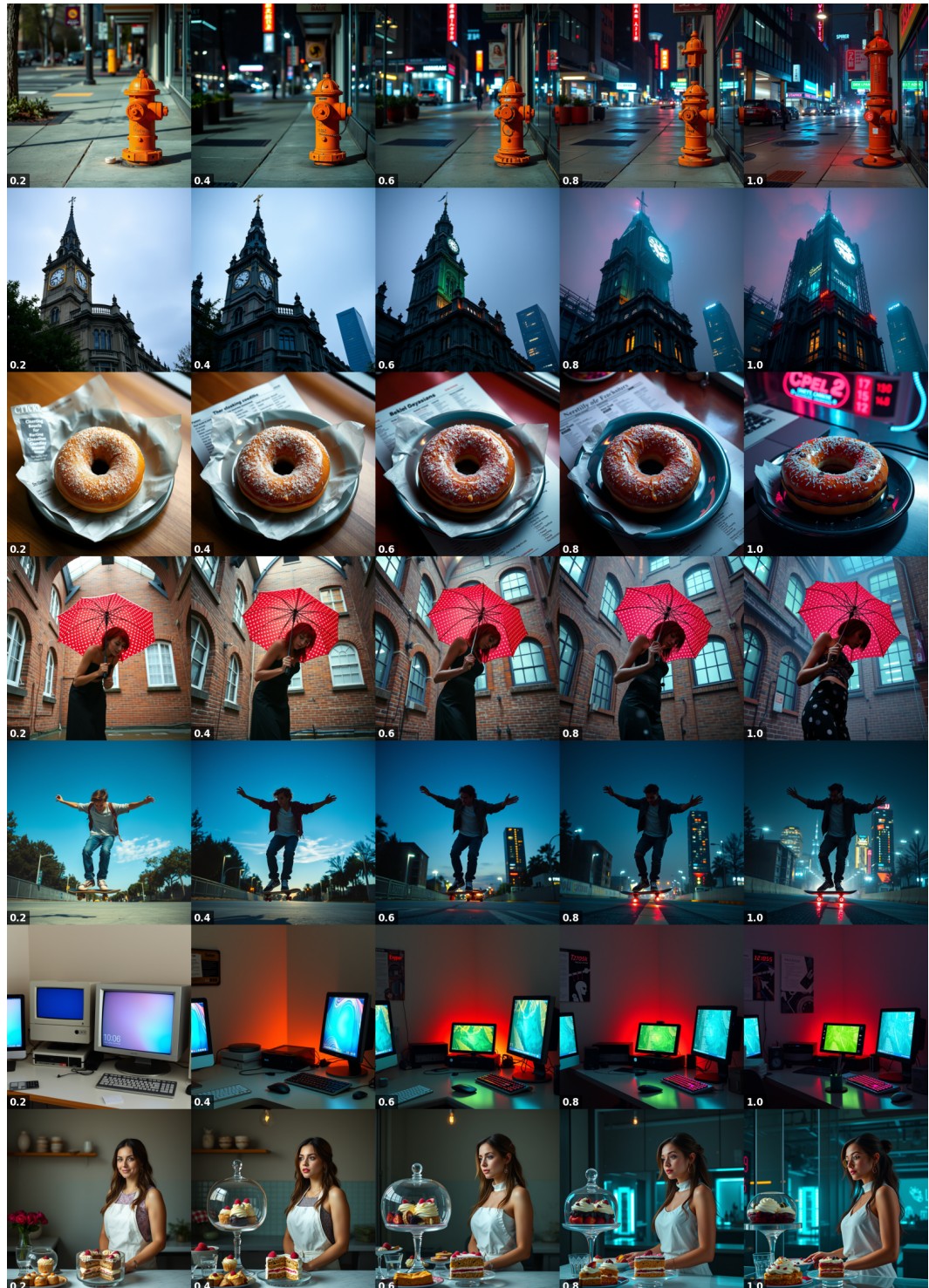

Figure 8: concept *cyberpunk*.

optimality theorem, it is natural to expect that the theoretically fastest PI gains, when combined with a suitably chosen derivative term, can outperform more conservative $K_i$ values that lie strictly inside the stability interval (under the same $K_p = 1$). Intuitively, because transformer depth is finite and we do not know at which layer the error will effectively settle, it is desirable to drive the error down quickly while preventing excessive overshoot, with $K_d$ acting as the compensating damping term.

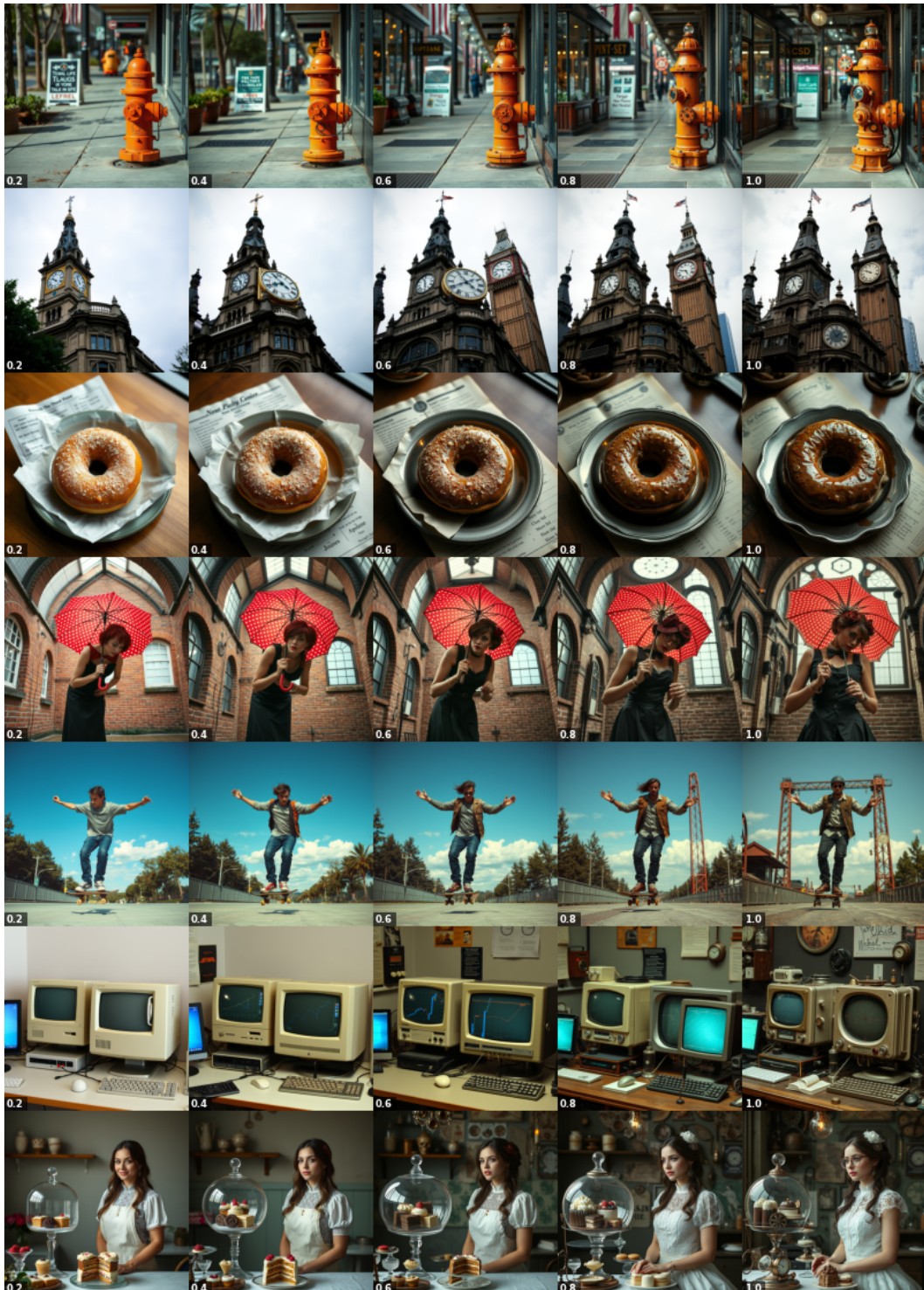

Figure 9: Concept *steampunk*

Our empirical results corroborate this interpretation. The stability intervals for $K_i$ at $K_p = 1$ are $(-0.23, 0.23)$ for Gemma-2-9B-it and $(-0.1355, 0.1355)$ for Gemma-2-2B. In Figs. 6a and 6b, we sweep across multiple $K_i$ values in these ranges. For Gemma-2-9B-it, the $K_i$ that yields the fastest theoretical convergence rate is 0.056. When $K_d$ increases from 0.0 to 0.01, the Llamaguard3 score rises from 76.61 to 78.53. This pattern suggests that with $K_d = 0$, the aggressive integral term produces noticeable overshoot that harms performance, whereas adding a small derivative term introduces sufficient damping to recover, and in some cases improve, steering performance.

Table 3: Full comparison of Original, DIM, ITI, RePE, and PID across models on ASR and general benchmarks on all tested models. Bold = best, underline = second-best within each model (ASR column).

| | Method | ASR↑ | tinyArc↑ | tinyGSM8k strict↑ | tinyMMLU↑ | tinyTruthQA↑ | tinyHellaSwag↑ | tinyWinoGrande↑ |
|---|---|---|---|---|---|---|---|---|
| **Qwen2.5-3B Instruct** | *Original* | – | 62.29 | 17.64 | 68.03 | 56.43 | 73.18 | 70.65 |
| | DIM | 74.03 | 61.95 | 14.80 | 66.11 | 54.95 | 72.40 | 69.85 |
| | ITI | 70.19 | 61.28 | 15.57 | 66.62 | 54.75 | 72.71 | 70.12 |
| | RePE | 68.44 | 61.05 | 14.60 | 65.70 | 54.30 | 72.03 | 69.40 |
| | PID | **76.07** | 61.20 | 16.01 | 67.29 | 54.10 | 72.59 | 69.72 |
| **Qwen2.5-7B Instruct** | *Original* | – | 68.36 | 81.68 | 72.57 | 56.41 | 78.87 | 75.19 |
| | DIM | 96.15 | 65.15 | 80.81 | 71.19 | 55.22 | 78.14 | 74.42 |
| | ITI | 84.61 | 65.76 | 79.48 | 71.23 | 55.63 | 78.36 | 74.75 |
| | RePE | 80.32 | 65.00 | 78.90 | 70.60 | 55.00 | 77.73 | 74.15 |
| | PID | **96.46** | 66.61 | 80.78 | 71.22 | 55.52 | 78.28 | 74.58 |
| **Qwen2.5-14B Instruct** | *Original* | – | 73.96 | 90.12 | 74.60 | 64.50 | 82.70 | 73.77 |
| | DIM | 90.38 | 72.74 | 87.01 | 74.30 | 63.01 | 81.94 | 72.93 |
| | ITI | 33.65 | 73.15 | 89.27 | 74.55 | 64.03 | 82.240 | 73.31 |
| | RePE | 25.42 | 72.40 | 86.20 | 73.90 | 63.20 | 81.52 | 72.60 |
| | PID | **92.65** | 72.13 | 88.96 | 74.52 | 63.60 | 82.60 | 73.04 |
| **Llama3.2-3B Instruct** | *Original* | – | 55.86 | 59.40 | 63.48 | 50.19 | 75.91 | 58.63 |
| | DIM | 88.46 | 54.24 | 58.63 | 61.68 | 49.78 | 75.10 | 57.94 |
| | ITI | 76.92 | 53.67 | 57.77 | 61.85 | 49.95 | 75.22 | 58.16 |
| | RePE | 70.15 | 53.40 | 57.00 | 61.10 | 49.50 | 74.75 | 57.53 |
| | PID | **89.76** | 53.93 | 57.26 | 62.01 | 50.19 | 75.07 | 57.83 |
| **Llama3.1-8B Instruct** | *Original* | – | 65.33 | 63.21 | 62.02 | 54.39 | 82.51 | 65.56 |
| | DIM | 93.26 | 62.01 | 60.57 | 60.96 | 54.17 | 81.73 | 64.81 |
| | ITI | 79.80 | 64.26 | 61.85 | 61.37 | 54.33 | 82.01 | 65.21 |
| | RePE | 70.42 | 61.40 | 60.00 | 60.20 | 53.70 | 81.35 | 64.45 |
| | PID | **94.85** | 62.30 | 61.99 | 61.54 | 54.24 | 81.87 | 64.93 |
| **Gemma2-9B Instruct** | *Original* | – | 69.31 | 83.19 | 76.60 | 55.07 | 82.31 | 72.34 |
| | DIM | 77.88 | 68.21 | 80.14 | 72.29 | 51.86 | 81.45 | 71.51 |
| | ITI | 35.57 | 68.32 | 81.47 | 75.33 | 53.13 | 81.70 | 71.82 |
| | RePE | 28.64 | 67.50 | 79.20 | 71.10 | 51.10 | 81.20 | 71.15 |
| | PID | **79.50** | 67.91 | 79.24 | 74.89 | 52.49 | 81.59 | 71.42 |
| **Gemma2-27B Instruct** | *Original* | – | 73.45 | 86.91 | 76.11 | 61.36 | 83.24 | 75.47 |
| | DIM | 74.03 | 72.13 | 84.70 | 74.59 | 59.49 | 81.79 | 74.21 |
| | ITI | 37.36 | 72.84 | 86.38 | 75.51 | 60.85 | 82.83 | 75.11 |
| | RePE | 24.19 | 71.03 | 84.00 | 73.90 | 58.79 | 80.82 | 73.44 |
| | PID | **79.80** | 72.93 | 86.60 | 75.67 | 60.74 | 82.95 | 75.06 |
| **Gemma2-9B-IT-Deeper-Align** | *Original* | – | 69.07 | 82.93 | 76.18 | 54.82 | 82.27 | 72.11 |
| | DIM | 11.91 | 67.88 | 79.76 | 72.12 | 51.83 | 81.33 | 71.47 |
| | ITI | 23.12 | 68.19 | 81.41 | 75.22 | 53.05 | 81.58 | 71.79 |
| | RePE | 11.23 | 67.37 | 78.96 | 70.92 | 51.06 | 81.08 | 71.09 |
| | PID | **34.75** | 67.83 | 79.11 | 74.77 | 52.44 | 81.46 | 71.33 |

