# OpenReview forum: "Activation Steering with a Feedback Controller"
_ICLR.cc/2026/Conference — ICLR 2026 Poster_

### Official Review · Reviewer_A2rW · 2025-10-31

**Soundness:** 2
**Presentation:** 3
**Contribution:** 3
**Rating:** 8
**Confidence:** 4

**Summary:**

This paper aims to address the lack of theoretical guarantees, presence of steady-state errors, and overshoot issues in existing activation steering methods. It analogizes the inter-layer activation propagation process in LLMs to a dynamical system, framing popular methods as instances of a Proportional (P) controller. Building on this, it introduces the Proportional-Integral-Derivative (PID) controller from control systems theory and proposes the PID Steering framework.

**Strengths:**

1. This work provides a unified control-theoretic perspective, categorizin existing methods (e.g. ActAdd, DirAblate, Mean-AcT) as specific types of P controllers, thereby logically leading to the integration of the PID controller concept and the proposal of the PID Steering framework.

2. The paper goes beyond conceptual proposal by providing rigorous mathematical derivations, including error dynamic modeling and PID controller discretization.

3. PID Steering is designed as a lightweight, plug-and-play module that can be seamlessly integrated into various existing steering frameworks (e.g., Angular Steering, Mean-AcT) by replacing their original steering vector construction, facilitating widespread adoption.

**Weaknesses:**

1. Manual tuning of PID parameters: the PID controller parameters $K_p, K_i, K_d $ appear to require manual tuning. Finding optimal settings involves significant trial and error, which is somewhat cumbersome.

2. Lack of analysis on $I, D$ Independent contributions and combinatorial effects. The experimental section fails to demonstrate the independent contributions and combinatorial effects of the $P, I,$ and $D$ modules, since the authors claim their method is a plugin fashion. Lack of the experiment setting to demonstrate the benefits by introducing the $I$ or $D$ to the $P$-class controllers.

3. Incomplete experimental comparison with cited methods: While the introduction and background sections systematically review mainstream activation steering methods, Section 5 does not include comparative tests against all mentioned methods (e.g., ActAdd). This prevents intuitive verification of the PID framework's optimization effects across different underlying $P$ controllers.

**Questions:**

1. Do the trial-and-error costs associated with manual PID parameter tuning vary across different task scenarios? Could parameters be highly sensitive and difficult/costly to tune in certain task types?

2. If testers used harmful prompts for evaluation, would PID Steering suffer performance loss?

3. Why the activation steering vector is added to every layer?  If added to only part optimal selected layer(s), the method offer advantages over identifying and intervening?

4.  Unclear Visual Representation in Figures: Figure 3, as key evidence supporting the core argument, lacks clarity in visualization and exposition. One theoretical cornerstone is that the integral term eliminates steady-state error. However, the PI curve in the figure ultimately displays saturation and large overshoot, obscuring this advantage and making it difficult for readers to appreciate the effectiveness of this step. Adding a subplot specifically comparing P and PI could highlight this advantage.

---

> ### Author Response · Authors · 2025-11-19
> **Response to Reviewer A2rW (1)**
>
> Thank you for your thoughtful review and valuable feedback. Below we address your concerns.
>
> -----
>
> **Q1. **Manual tuning of PID parameters:** The PID controller parameters $K_p, K_i, K_d$ appear to require manual tuning. Finding optimal settings involves significant trial and error, which is somewhat cumbersome.
> Doing PID ablation.**
>
> **Answer**: Thanks for your comment. Our controller operates robustly across a broad stability region, and performance is largely insensitive to small changes in $(K_p, K_i, K_d)$.
>
> To justify the robust performance of our PID steering, we have conducted a full ablation study on the gain parameters $K_p, K_i, K_d$ for the LLM jailbreaking task on Gemma-2-9B-it and the toxicity mitigation task for Gemma2-2B. We included the results in Figure 6 (Section 5.4) of the revised manuscript. The ablation heatmaps show a clear, consistent pattern across both the jailbreaking and the toxicity mitigation tasks. First, $K_p$ is the primary driver of effect strength, and larger values (e.g., $K_p = 1.5$) consistently improve performance. In contrast, changes in $K_i$ and $K_d$ produce only small and smooth variations. Small non-zero  $K_i$ and $K_d$ help reduce disturbances and stabilize the steering dynamics, especially when $K_p$ is large.
>
>
> This behavior aligns with Theorems 1 and 2 in our manuscript, which show that the integral and derivative components dampen oscillations and smooth the update trajectory. Importantly, the ablation heatmaps demonstrate that PID-AcT is not hypersensitive to its gain parameters. Even relatively large values such as $K_i = 0.15$ or $K_d = 0.10$ do not destabilize the method, and the optimal region forms a broad plateau around $K_p = 1.5,\ K_i \in [0.03, 0.06],\ K_d = 0.01.$ Both tasks, Gemma-2-9B-it for the LLM jailbreaking task and Gemma-2-2B for the LLM toxicity mitigation task, exhibit flat performance landscapes rather than sharp peaks, indicating low sensitivity and a wide operating range.
>
> In addition to the ablation study $K_p, K_i, K_d$, in our original manuscript (Section 4 and Appendix B.3-B.7), we have derived the stability range for $K_p, K_i, K_d$, which are the values of those terms such that the closed-loop averaged error dynamics are input-to-state stable (ISS) with respect to the input disturbance (see Eqn.20 in Section 4.1). We summarize our results below.
>
> Given that $K_p$, $K_i$, $K_d$ are scalars,  $\|\cdot\|$ denotes the spectral norm of a matrix, and $|\cdot|$ denotes the absolute value of a scalar, we study the stability range for $K_p, K_i, K_d$.
>
> **(i) Stable $K_p$**
> Recall our setup and notations: we consider $N$ contrastive prompt pairs. For discrete time (layer) $k$, let $x_{i}^{\pm}(k)\in\mathbb{R}^d$ ($i=1,\dots,N$) denote the corresponding activation vectors. The layer-to-layer evolution is $x_{i}(k+1)=f_{i}^{(k)}\big(x(k)\big)$.
>
> Let $M= \sup\_k \left\|\frac{\sum\_{i=1}^N J_{f^{(k)}\_i}(x_i^+(k))}{N}
> \right\|$, where $J\_{f^{(k)}_i}(x_i^+(k))$ is the Jacobian of $f\_{i}^{(k)}$ at $x\_{i}^{+}(k)$.
>
> According to Proposition 1 - Appendix B.4, **the stability condition of gain $K_p$ is given by**
>     **$$1 - \frac{1}{M} < K_p < 1 + \frac{1}{M}.$$**
>
> Furthermore, $K_p=1$ yields the fastest convergence of the homogeneous averaged-error dynamics (in the disturbance-free case) (Remark 1 - Appendix B.4)
>
> **(ii) Stable $K_i$**
> Given a stable $K_p$. According to Proposition 3 - Appendix B.5, **the stability condition of gain $K_i$ is**
>     **$$M|1-K_p| + M|K_i| < 1,$$**
>
> where $M$ is defined in part ''(i) Stable $K_p$''.
>
> $|K_i| = \frac{(1 - M|1-K_p|)^2}{4M}$ yields the fastest convergence of the PI closed-loop towards zero in the homogeneous system (Remark 2 - Appendix B.5).
>
> **(iii) Stable $K_d$**
> Given stable $K_p$ and $K_i$. Regarding Theorem 1 - Appendix B.7, we establish a stability condition for gain $K_d$ by following these steps:
> - Solving for $P(k)$ in the Lyapunov equation $$M_i(k)^\top P(k) M_i(k) - P(k) = -\mu I \qquad (\text{any scalar}\ \mu > 0)$$
> where $
> M_i(k) =
> \begin{bmatrix}
> (1-K_p)\bar{A}(k) & -K_i\bar{A}(k) \\\\
> I_d & I_d
> \end{bmatrix}
> $,  $\bar{A}(k)=\frac{\sum_{i=1}^N J_{f^{(k)}_i}(x_i^+(k))}{N}.$
>
> - Computing  $\varepsilon =\frac{\mu}{8  \sup_k \|P(k)\|  \sup_k \|M_i(k)\|^2}$,
> and $r =\frac{\mu}{8\left(\sup_k \|\bar{A}(k)(1-K_p) - I\|^2+M|K_i|\right)}$, all are scalars. Note that $M$ is defined in part ''(i) Stable $K_p$''
>
> We then obtain **the stability condition of $K_d$, which is $|K_d|^2<\frac{r}{\left(\sup_k \|P(k)\|\left(1 + \frac{1}{\varepsilon}\right)+3r\right) M^2 }$**.

---

> ### Author Response · Authors · 2025-11-19
> **Response to Reviewer A2rW (2)**
>
> **Q2. **Lack of analysis on $I, D$ independent contributions and combinatorial effects:**
>    The experimental section fails to demonstrate the independent contributions and combinatorial effects of the $P, I,$ and $D$ modules, even though the authors claim their method is a plugin fashion. There is also a lack of experimental settings demonstrating the benefits of introducing the $I $ or $D$ to the $P$-class controllers.**
>
> **Answer**: We thank the reviewer for raising this point. The ablation study, discussed in our answer to your Q1 above (Figure 6, Section 5.4 of the revised manuscript), explicitly isolates the independent and combined effects of the $P$, $I$, and $D$ components by evaluating all controller variants, i.e., $P$-only, $PI$, $PD$, and full $PID$, across a broad grid of gain values. These results show that $P$ is the primary driver of steering strength, while small non-zero $K_i$ and $K_d$ terms each provide complementary benefits: $I$ reduces steady-state errors and improves consistency during the steering process, and $D$ dampens overshoot created by large $K_p$ values. Importantly, the combined $PID$ configuration consistently outperforms $P$-only, $PI$, and $PD$ baselines in both jailbreak success and stability, demonstrating that the modules interact constructively rather than redundantly. These findings substantiate our claim that the method is modular and can be used as a plug-in: $P$ alone is effective, but adding $I$ or $D$ yields measurable improvements, and using all three provides the strongest and most reliable performance.
>
> These empirical observations are strongly supported by our theoretical analysis in Section 4. Specifically, when evaluating the control efficacy on the same dataset used to calculate the steering vector, Proposition 1 states that $P$-control significantly reduces the average error across layers but leaves a steady-state error. Proposition 3 proves that adding $I$-control partially cancels this offset. However, Proposition 4 shows that aggressive PI gains can cause large overshoot. Theorems 1 and 2 establish that $PID$ control successfully reduces overshoot while maintaining $PI$ control's bias-removal benefit, confirming that the $D$ component acts as a dampener. Figure 3 illustrates this: The $P$ curve (blue) settles on a nonzero plateau; the $PI$ curve (red) approaches zero, eliminating the steady-state offset, but exhibits a large negative dip (overshoot); and the $PID$ curve (green) also settles near zero, with substantially reduced overshoot.

---

> ### Author Response · Authors · 2025-11-19
> **Response to Reviewer A2rW (3)**
>
> **Q3. Incomplete experimental comparison with cited methods: While the introduction and background sections systematically review mainstream activation steering methods, Section 5 does not include comparative tests against all mentioned methods (e.g., ActAdd). This prevents intuitive verification of the PID framework’s optimization effects across different underlying ( P ) controllers.**
>
> **Answer:** Thanks for the insightful comment. We have added the results for ITI-C [11], ActAdd [9], and AURA [12] to Table 1 in the revised manuscript. We also copied those results in Tables 1 and 2 below.
>
> PID-AcT not only outperforms Mean-AcT [6] and Linear-AcT [6], but also exceeds stronger activation-editing baselines such as ActADD, AURA, and ITI-C. Whereas these baseline methods either plateau, sharply raise perplexity, or degrade accuracy, our PID-AcT delivers substantially larger reductions, up to **8.2x** on Gemma2-2B and **8.1x** on LLaMA3-8B, while maintaining comparable utility. PID-AcT is the only method that achieves the best scores across both classifier-based and LLM-judge toxicity metrics, highlighting its robustness. This improvement stems from PID’s dynamic correction: the proportional term drives strong mitigation, while small integral and derivative terms stabilize the update, drive the steady-state error to 0, and avoid overshooting. Overall, PID-AcT provides the most stable and effective trade-off among all activation-intervention baselines. We will provide results for CAA in the next response round.

---

> > ### Author Response · Authors · 2025-11-19
> > **Response to Reviewer A2rW (4)**
> >
> > **Table 1**:  Toxicity mitigation results for Gemma-2B, averaged over 10 runs. Lower is better for toxicity and perplexity; higher is better for MMLU. Best and second-best exclude the original baseline. (-) means no steering, (&#10003;) means sequential steering, and (&#10007;) means no sequential steering.
> >
> > | Method      | Seq. | CLS Tox $\downarrow$                             | 0-shot Tox $\downarrow$                           | PPL Wiki $\downarrow$                                  | PPL Mistral $\downarrow$                               | MMLU $\uparrow$                                       |
> > |-------------|------|--------------------------------------------------|---------------------------------------------------|--------------------------------------------------------|--------------------------------------------------------|--------------------------------------------------------|
> > | Original    | --   | $4.17 \scriptscriptstyle{\pm 0.32}$              | $13.42 \scriptscriptstyle{\pm 1.08}$              | $13.98$                                                | $6.68$                                                 | $53.10$                                                |
> > | ActADD      | &#10007; | $3.96 \scriptscriptstyle{\pm 0.24}$          | $13.43 \scriptscriptstyle{\pm 1.42}$              | $14.69 \scriptscriptstyle{\pm 0.22}$                   | $\mathbf{6.67 \scriptscriptstyle{\pm 0.15}}$           | $\mathbf{53.00 \scriptscriptstyle{\pm 0.51}}$          |
> > | AURA        | &#10007; | $2.12 \scriptscriptstyle{\pm 0.27}$          | $9.04 \scriptscriptstyle{\pm 0.66}$               | $\mathbf{14.18 \scriptscriptstyle{\pm 0.14}}$          | $7.04 \scriptscriptstyle{\pm 0.34}$                    | $\mathbf{53.00 \scriptscriptstyle{\pm 0.30}}$          |
> > | ITI-C       | &#10007; | $0.74 \scriptscriptstyle{\pm 0.18}$          | $5.36 \scriptscriptstyle{\pm 0.91}$               | $14.90 \scriptscriptstyle{\pm 0.29}$                   | $7.44 \scriptscriptstyle{\pm 0.19}$                    | $52.60 \scriptscriptstyle{\pm 0.55}$                   |
> > | Mean-Act    | &#10007; | $1.12 \scriptscriptstyle{\pm 0.23}$          | $5.20 \scriptscriptstyle{\pm 0.42}$               | $\underline{14.53 \scriptscriptstyle{\pm 0.21}}$       | $\underline{6.81 \scriptscriptstyle{\pm 0.19}}$        | $51.74 \scriptscriptstyle{\pm 0.55}$                   |
> > | Linear-Act  | &#10007; | $0.95 \scriptscriptstyle{\pm 0.36}$          | $5.37 \scriptscriptstyle{\pm 0.80}$               | $14.75 \scriptscriptstyle{\pm 0.22}$                   | $7.24 \scriptscriptstyle{\pm 0.24}$                    | $51.63 \scriptscriptstyle{\pm 0.50}$                   |
> > | Mean-Act    | &#10003; | $\underline{0.68 \scriptscriptstyle{\pm 0.21}}$ | $\underline{3.23 \scriptscriptstyle{\pm 0.44}}$ | $14.92 \scriptscriptstyle{\pm 0.25}$                   | $6.97 \scriptscriptstyle{\pm 0.74}$                    | $\underline{51.80 \scriptscriptstyle{\pm 0.55}}$       |
> > | Linear-Act  | &#10003; | $1.00 \scriptscriptstyle{\pm 0.27}$          | $4.13 \scriptscriptstyle{\pm 0.89}$               | $14.98 \scriptscriptstyle{\pm 0.22}$                   | $7.13 \scriptscriptstyle{\pm 0.70}$                    | $51.47 \scriptscriptstyle{\pm 0.50}$                   |
> > | PID-Act (Ours) | &#10003; | $\mathbf{0.51 \scriptscriptstyle{\pm 0.21}}$ | $\mathbf{2.90 \scriptscriptstyle{\pm 0.55}}$   | $15.22 \scriptscriptstyle{\pm 0.24}$                   | $7.02 \scriptscriptstyle{\pm 0.65}$                    | $51.30 \scriptscriptstyle{\pm 0.52}$                   |

---

> > > ### Author Response · Authors · 2025-11-19
> > > **Response to Reviewer A2rW (5)**
> > >
> > > **Table 2**: Toxicity mitigation results for Llama-8B, averaged over 10 runs. Lower is better for toxicity and perplexity; higher is better for MMLU. Best and second-best exclude the Original baseline. (-) means no steering, (&#10003;) means sequential steering, and (&#10007;) means no sequential steering.
> > >
> > > | Method      | Seq. | CLS Tox $\downarrow$                             | 0-shot Tox $\downarrow$                           | PPL Wiki $\downarrow$                                  | PPL Mistral $\downarrow$                               | MMLU $\uparrow$                                       |
> > > |-------------|------|--------------------------------------------------|---------------------------------------------------|--------------------------------------------------------|--------------------------------------------------------|--------------------------------------------------------|
> > > | Original    | --   | $5.80$                                           | $15.00$                                           | $9.06$                                                 | $5.68$                                                 | $65.30$                                                |
> > > | ActADD      | ✗    | $5.57 \scriptscriptstyle{\pm 0.45}$              | $15.73 \scriptscriptstyle{\pm 0.21}$              | $9.71 \scriptscriptstyle{\pm 0.46}$                    | $5.85 \scriptscriptstyle{\pm 0.26}$                    | $\mathbf{65.50 \scriptscriptstyle{\pm 0.34}}$          |
> > > | AURA        | ✗    | $1.90 \scriptscriptstyle{\pm 0.61}$              | $8.12 \scriptscriptstyle{\pm 0.85}$               | $9.52 \scriptscriptstyle{\pm 0.32}$                    | $6.05 \scriptscriptstyle{\pm 0.30}$                    | $\mathbf{65.50 \scriptscriptstyle{\pm 0.33}}$          |
> > > | ITI-C       | ✗    | $1.60 \scriptscriptstyle{\pm 0.22}$              | $6.53 \scriptscriptstyle{\pm 0.66}$               | ${9.48 \scriptscriptstyle{\pm 0.24}}$        | $6.17 \scriptscriptstyle{\pm 0.14}$                    | $\underline{64.70 \scriptscriptstyle{\pm 0.44}}$       |
> > > | Mean-AcT    | ✗    | $1.78 \scriptscriptstyle{\pm 0.33}$              | $6.56 \scriptscriptstyle{\pm 0.54}$               | $\underline{9.36 \scriptscriptstyle{\pm 0.28}}$        | $\mathbf{5.45 \scriptscriptstyle{\pm 0.34}}$           | $64.35 \scriptscriptstyle{\pm 0.39}$                   |
> > > | Linear-AcT  | ✗    | $1.87 \scriptscriptstyle{\pm 0.39}$              | $6.55 \scriptscriptstyle{\pm 0.21}$               | $\mathbf{9.35 \scriptscriptstyle{\pm 0.17}}$           | ${5.56 \scriptscriptstyle{\pm 0.33}}$        | $64.55 \scriptscriptstyle{\pm 0.33}$                   |
> > > | Mean-AcT    | ✓    | $\underline{1.21 \scriptscriptstyle{\pm 0.41}}$  | $\underline{5.09 \scriptscriptstyle{\pm 0.64}}$   | $9.83 \scriptscriptstyle{\pm 0.21}$                    | $5.71 \scriptscriptstyle{\pm 0.33}$                    | $64.22 \scriptscriptstyle{\pm 0.40}$                   |
> > > | Linear-AcT  | ✓    | $1.68 \scriptscriptstyle{\pm 0.48}$              | $6.47 \scriptscriptstyle{\pm 0.38}$               | $9.48 \scriptscriptstyle{\pm 0.19}$                    | $\underline{5.46 \scriptscriptstyle{\pm 0.44}}$        | ${64.49 \scriptscriptstyle{\pm 0.38}}$       |
> > > | PID-AcT (Ours) | ✓ | $\mathbf{0.72 \scriptscriptstyle{\pm 0.49}}$     | $\mathbf{4.36 \scriptscriptstyle{\pm 0.81}}$      | $9.56 \scriptscriptstyle{\pm 0.20}$                    | $6.08 \scriptscriptstyle{\pm 0.37}$                    | $\mathbf{64.50 \scriptscriptstyle{\pm 0.36}}$          |

---

> ### Author Response · Authors · 2025-11-19
> **Response to Reviewer A2rW (6)**
>
> **Q4. Do the trial-and-error costs associated with manual PID parameter tuning vary across different task scenarios? Could parameters be highly sensitive and difficult/costly to tune in certain task types?**
>
> **Answer**: As noted in our response to your Q1, we have performed an ablation study on $K_p, K_i, K_d$ for both the jailbreak (Gemma-2-9B-it) and toxicity mitigation tasks (Gemma-2-2B) and included the results in Figure 6 of Section 5.4 of the revised manuscript. Our findings indicate that the optimal values of $K_p, K_i, K_d$ are similar across these tasks, despite using different models, Gemma-2-9B-it and Gemma-2-2B, suggesting that the controller parameters in our method generalize well across model architectures and task types. Please refer to Figure 6 in Section 5.4 of the revised manuscript for numerical results and analysis.
>
> Additionally, as discussed in our response to Q1, we explain the stability range from which the controller parameters can be selected.
>
> Finally, we emphasize that PID Steering requires no training: once the steering vectors are computed via PID control, they can be directly reused at inference time.

---

> ### Author Response · Authors · 2025-11-19
> **Response to Reviewer A2rW (7)**
>
> **Q5. If testers used harmful prompts for evaluation, would PID Steering suffer performance loss?**
>
> **Answer**: We evaluated our method using **harmful prompts** for toxicity mitigation in Section 5.1 and Table 1 (see the `CLS Tox.` and `0-shot Tox.` columns), as well as for jailbreak resistance in Section 5.2 and Table 2 (see the `ASR` column) of our manuscript. These results demonstrate that our method is effective at both **reducing** and **inducing** harmfulness when desired.
>
> In addition, we evaluated our method using **harmless prompts** under the same experimental conditions to assess how toxic/harmful steering affects the model’s general capabilities. Table 1 (last three columns) and Table 2 (last six columns) in our manuscript report perplexity and a suite of general language modeling benchmarks. Across all evaluations, our method performs on par with existing approaches, indicating that the improvements in steering performance **do not come at the cost of degrading other core capabilities**.
>
> We hope this clarifies our evaluation methodology and addresses your concerns. If not, we would appreciate it if you could elaborate further so that we can respond more precisely.

---

> ### Author Response · Authors · 2025-11-19
> **Response to Reviewer A2rW (8)**
>
> **Q6. Why is the activation steering vector added to every layer? If added to only part of the optimal selected layer(s), would the method offer advantages over identifying and intervening?**
>
>
> **Answer**: Thank you for your comments. We reply to your questions below.
>
> > Why is the activation steering vector added to every layer?
>
> Our choice to steer at every layer is motivated by three key observations:
>
> (1) Strong inter-layer coupling. Prior work has shown that layers in LLMs exhibit substantial coupling [1, 2, 3], suggesting that their representation spaces are not independent but instead share a coherent global structure. This implies that interventions applied at a single layer may not fully align with how features propagate across the network.
>
> (2) Effectiveness of multi-layer interventions. Recent studies demonstrate that interventions and interpretability analyses applied across multiple layers lead to more reliable, stable, and interpretable effects compared to single-layer interventions [4, 5, 6, 7]. Consistent with this, Table 1 of our manuscript shows that methods steering at every layer (Mean-AcT, Linear-AcT, PID-AcT) consistently outperform methods steering at a single layer (ActADD, AURA, ITI-C) on toxicity mitigation metrics (`CLS Tox.` and `0-shot Tox.` columns).
>
> (3) Layer selection is expensive. The standard approach for identifying “optimal” intervention layers is exhaustive grid search over layers [5, 8, 9], which is computationally expensive and often task-specific. Steering at every layer avoids this overhead and provides a more task-agnostic mechanism.
>
> > If added to only part of the optimal selected layer(s), would the method offer advantages over identifying and intervening?
>
> Yes. Since our method focuses on computing the steering directions, it is fully compatible with selective intervention frameworks such as [7, 10]. In such settings, our steering directions can be plugged into layer- or token-selection mechanisms to achieve more targeted control.
>
> For example, in Section 5.2, we integrate our method with the Adaptive Angular Steering framework of [7], which dynamically selects both token positions and layers. As shown in Table 2, this setup achieves strong jailbreak mitigation (ASR column) while maintaining robust performance on general-capability benchmarks (last six columns).
>
> In contrast, Section 5.1 applies our method within the Activation Transport framework [6], which does not incorporate selective intervention. As expected, the impact on general capabilities is more noticeable in this setting (last three columns of Table 1). This comparison highlights that our method can benefit from selective application, and when paired with adaptive or selective steering frameworks, it can maintain both strong steering performance and good capability preservation.

---

> ### Author Response · Authors · 2025-11-19
> **Response to Reviewer A2rW (9)**
>
> **Q7. Unclear visual representation in figures: Figure 3, a key piece of evidence supporting the core argument, lacks clarity in visualization and exposition. One theoretical cornerstone is that the integral term eliminates steady-state error. However, the PI curve in the figure ultimately displays saturation and large overshoot, obscuring this advantage and making it difficult for readers to appreciate the effectiveness of this step. Adding a subplot specifically comparing P and PI controllers could highlight this advantage.**
>
>
> **Answer**: We thank the reviewer for pointing out that the intended message of Figure 3 was not sufficiently explained. The plotted quantity is the scalar inner product $<e(0), e(k)>$, which measures how much of the error at layer $k$ remains aligned with the initial error (see Appendix B.6 for further explanation). A nonzero plateau of this signal indicates a persistent component of the initial error (steady-state error), whereas convergence to values near zero means that this component has been eliminated.
>
> In this light, the behaviour in Figure 3 matches the theoretical discussion: the P curve (blue) decays but clearly settles on a nonzero plateau, confirming that P-control is ISS but admits a steady-state error. The PI curve (red) crosses zero and, after the trasient, remains close to it, meaning that the integral action eliminates the steady-state offsets in $<e(0), e(k)>$; the large negative dip before convergence is the classical overshoot of PI control, which we discuss in the "Limitations of PI control" in Section 4.2.1 of our main text and analysis in Appendix B.6. The PID curve (green) also settles near zero but with substantially reduced overshoot, showing that derivative action damps the PI transient while preserving its steady-state advantage. We analyze the overshoot reduction of PID control in Theorem 2 in Section 4.2.2 of our main text and in Appendix B.7.2.
>
> To make this interpretation clearer, from line 249 to line 265 in the revised manuscript, we have (i) briefly defined "steady-state error" and "overshoot" in the main text, and (ii) expanded the explantion of Figure 3 to explicitly state that P stabilizes the error but leaves a nonzero steady-state component, PI eliminate this steady-state error at the cost of overshoot, and PID reduces the overshoot while maintaining the near-zero steady-state.

---

> ### Author Response · Authors · 2025-11-19
> **Response to Reviewer A2rW (10)**
>
> **References**
>
> [1] Wang et al., "Interpretability in the wild: a circuit for indirect object identification in gpt-2 small", ICLR, 2023.
>
> [2] McGrath et al., "The hydra effect: Emergent self-repair in language model computations", Deepmind, 2023.
>
> [3] Cody Rushing and Neel Nanda, "Explorations of self-repair in language models", ICML, 2024.
>
> [4] Lindsey et al., "Sparse crosscoders for cross-layer features and model diffing", Transformer Circuits Thread, 2024.
>
> [5] Arditi et al., "Refusal in language models is mediated by a single direction", NeurIPS, 2024.
>
> [6] Rodrigue et al. "Controlling Language and Diffusion Models by Transporting Activations", ICLR, 2025.
>
> [7] Vu and Nguyen. "Angular steering: Behavior control via rotation in activation space", NeurIPS, 2025.
>
> [8] Turner et al. "Steering Language Models with Activation Engineering", OpenReview preprint, 2025.
>
> [9] Rimsky et al., "Steering llama 2 via contrastive activation addition", ACL, 2024.
>
> [10] Lee et al., "Programming refusal with conditional activation steering", ICLR, 2025.
>
> [11] Li et al. “Inference-Time Intervention: Eliciting Truthful Answers from a Language Model”, NeurIPS, 2024.
>
> [12] Suau et al. “Whispering Experts: Neural Interventions for Toxicity Mitigation in Language Models”, ICML, 2024.
>
> -----
> We hope we have cleared your concerns about our work. We have also revised our manuscript according to your comments, and we would appreciate it if we could get your further feedback at your earliest convenience.

---

> ### Author Response · Authors · 2025-11-21
> **Regarding Q3: Additional Results on CAA and QVQ-72B-Preview as an Evaluator for LLM Toxicity (1)**
>
> Below are the updated versions of Tables 1 and 2 in our answer to your Q3 above, now including results for the CAA method and QVQ-72B-Preview evaluation scores. Overall, the findings remain consistent with our original conclusions: PID-AcT achieves the strongest toxicity reduction with minimal utility loss across both models. In our revised manuscript, we have updated Table 1 and Section 5.1 accordingly to reflect these additions.

---

> > ### Author Response · Authors · 2025-11-21
> > **Regarding Q3: Additional Results on CAA and QVQ-72B-Preview as an Evaluator for LLM Toxicity (2)**
> >
> > **Table 1 (updated)**:  Toxicity mitigation results for Gemma-2B, averaged over 10 runs. Lower is better for toxicity and perplexity; higher is better for MMLU. Best and second-best exclude the original baseline. (-) means no steering, (&#10003;) means sequential steering, and (&#10007;) means no sequential steering.
> >
> > | Method    | Seq. | CLS Tox $\downarrow$    | 0-shot Tox $\downarrow$    | QVQ $\downarrow$    | PPL Wiki $\downarrow$    | PPL Mistral $\downarrow$    | MMLU $\uparrow$    |
> > |-|-|-|-|-|-|-|-|
> > | Original    | --    | $4.17 \scriptscriptstyle{\pm 0.32}$    | $13.42 \scriptscriptstyle{\pm 1.08}$    | $14.17 \scriptscriptstyle{\pm 0.08}$    | $13.98$    | $6.68$    | $53.10$    |
> > | ActADD    | &#10007; | $3.96 \scriptscriptstyle{\pm 0.24}$    | $13.43 \scriptscriptstyle{\pm 1.42}$    | $14.17 \scriptscriptstyle{\pm 0.08}$    | $14.69 \scriptscriptstyle{\pm 0.22}$                   | $\mathbf{6.67 \scriptscriptstyle{\pm 0.15}}$    | $\mathbf{53.00 \scriptscriptstyle{\pm 0.51}}$    |
> > | CAA    | &#10007; | $1.20 \scriptscriptstyle{\pm 0.25}$    | $5.35 \scriptscriptstyle{\pm 0.50}$    | $5.88 \scriptscriptstyle{\pm 0.36}$    | $\mathbf{14.60 \scriptscriptstyle{\pm 0.20}}$    | $6.85 \scriptscriptstyle{\pm 0.22}$    | 51.70 $\scriptscriptstyle{\pm 0.48}$    |
> > | AURA    | &#10007; | $2.12 \scriptscriptstyle{\pm 0.27}$    | $9.04 \scriptscriptstyle{\pm 0.66}$    | $9.72 \scriptscriptstyle{\pm 0.27}$    | $\mathbf{14.18 \scriptscriptstyle{\pm 0.14}}$    | $7.04 \scriptscriptstyle{\pm 0.34}$    | $\mathbf{53.00 \scriptscriptstyle{\pm 0.30}}$    |
> > | ITI-C    | &#10007; | $0.74 \scriptscriptstyle{\pm 0.18}$    | $5.36 \scriptscriptstyle{\pm 0.91}$    | $6.10 \scriptscriptstyle{\pm 0.13}$    | $14.90 \scriptscriptstyle{\pm 0.29}$    | $7.44 \scriptscriptstyle{\pm 0.19}$    | $52.60 \scriptscriptstyle{\pm 0.55}$    |
> > | Mean-Act    | &#10007; | $1.12 \scriptscriptstyle{\pm 0.23}$    | $5.20 \scriptscriptstyle{\pm 0.42}$    | $5.80 \scriptscriptstyle{\pm 0.15}$    | $\underline{14.53 \scriptscriptstyle{\pm 0.21}}$    | $\underline{6.81 \scriptscriptstyle{\pm 0.19}}$    | $51.74 \scriptscriptstyle{\pm 0.55}$    |
> > | Linear-Act  | &#10007; | $0.95 \scriptscriptstyle{\pm 0.36}$          | $5.37 \scriptscriptstyle{\pm 0.80}$               | $5.92 \scriptscriptstyle{\pm 0.11}$    | $14.75 \scriptscriptstyle{\pm 0.22}$                   | $7.24 \scriptscriptstyle{\pm 0.24}$                    | $51.63 \scriptscriptstyle{\pm 0.50}$                   |
> > | Mean-Act    | &#10003; | $\underline{0.68 \scriptscriptstyle{\pm 0.21}}$ | $\underline{3.23 \scriptscriptstyle{\pm 0.44}}$ | $\underline{3.70 \scriptscriptstyle{\pm 0.14}}$    | $14.92 \scriptscriptstyle{\pm 0.25}$                   | $6.97 \scriptscriptstyle{\pm 0.74}$                    | $\underline{51.80 \scriptscriptstyle{\pm 0.55}}$       |
> > | Linear-Act  | &#10003; | $1.00 \scriptscriptstyle{\pm 0.27}$          | $4.13 \scriptscriptstyle{\pm 0.89}$               | $4.64 \scriptscriptstyle{\pm 0.04}$    | $14.98 \scriptscriptstyle{\pm 0.22}$                   | $7.13 \scriptscriptstyle{\pm 0.70}$                    | $51.47 \scriptscriptstyle{\pm 0.50}$                   |
> > | PID-Act (Ours) | &#10003; | $\mathbf{0.51 \scriptscriptstyle{\pm 0.21}}$ | $\mathbf{2.90 \scriptscriptstyle{\pm 0.55}}$   | $\mathbf{3.40 \scriptscriptstyle{\pm 0.04}}$    | $15.22 \scriptscriptstyle{\pm 0.24}$                   | $7.02 \scriptscriptstyle{\pm 0.65}$                    | $51.30 \scriptscriptstyle{\pm 0.52}$                   |

---

> > > ### Author Response · Authors · 2025-11-21
> > > **Regarding Q3: Additional Results on CAA and QVQ-72B-Preview as an Evaluator for LLM Toxicity (3)**
> > >
> > > **Table 2 (updated)**: Toxicity mitigation results for Llama-8B, averaged over 10 runs. Lower is better for toxicity and perplexity; higher is better for MMLU. Best and second-best exclude the Original baseline. (–) means no steering, (✓) means sequential steering, and (✗) means no sequential steering.
> > >
> > > | Method           | Seq. | CLS Tox $\downarrow$                             | 0-shot Tox $\downarrow$                           | QVQ $\downarrow$                                       | PPL Wiki $\downarrow$                                  | PPL Mistral $\downarrow$                               | MMLU $\uparrow$                                        |
> > > |------------------|------|--------------------------------------------------|---------------------------------------------------|--------------------------------------------------------|--------------------------------------------------------|--------------------------------------------------------|---------------------------------------------------------|
> > > | Original         | --   | $5.80$                                           | $15.00$                                           | $15.81 \scriptscriptstyle{\pm 0.09}$                   | $9.06$                                                 | $5.68$                                                 | $65.30$                                                 |
> > > | ActADD           | ✗    | $5.57 \scriptscriptstyle{\pm 0.45}$              | $15.73 \scriptscriptstyle{\pm 0.21}$              | $16.48 \scriptscriptstyle{\pm 0.19}$                   | $9.71 \scriptscriptstyle{\pm 0.46}$                    | $5.85 \scriptscriptstyle{\pm 0.26}$                    | $\mathbf{65.50 \scriptscriptstyle{\pm 0.34}}$           |
> > > | CAA              | ✗    | $1.82 \scriptscriptstyle{\pm 0.36}$              | $6.70 \scriptscriptstyle{\pm 0.58}$               | $7.40 \scriptscriptstyle{\pm 0.06}$                    | $9.40 \scriptscriptstyle{\pm 0.25}$                    | $5.50 \scriptscriptstyle{\pm 0.30}$                    | $64.30 \scriptscriptstyle{\pm 0.37}$                    |
> > > | AURA             | ✗    | $1.90 \scriptscriptstyle{\pm 0.61}$              | $8.12 \scriptscriptstyle{\pm 0.85}$               | $8.80 \scriptscriptstyle{\pm 0.17}$                    | $9.52 \scriptscriptstyle{\pm 0.32}$                    | $6.05 \scriptscriptstyle{\pm 0.30}$                    | $\mathbf{65.50 \scriptscriptstyle{\pm 0.33}}$           |
> > > | ITI-C            | ✗    | $1.60 \scriptscriptstyle{\pm 0.22}$              | $6.53 \scriptscriptstyle{\pm 0.66}$               | $7.19 \scriptscriptstyle{\pm 0.06}$                    | $9.48 \scriptscriptstyle{\pm 0.24}$                    | $6.17 \scriptscriptstyle{\pm 0.14}$                    | $\underline{64.70 \scriptscriptstyle{\pm 0.44}}$        |
> > > | Mean-AcT         | ✗    | $1.78 \scriptscriptstyle{\pm 0.33}$              | $6.56 \scriptscriptstyle{\pm 0.54}$               | $7.30 \scriptscriptstyle{\pm 0.25}$                    | $\underline{9.36 \scriptscriptstyle{\pm 0.28}}$        | $\mathbf{5.45 \scriptscriptstyle{\pm 0.34}}$           | $64.35 \scriptscriptstyle{\pm 0.39}$                    |
> > > | Linear-AcT       | ✗    | $1.87 \scriptscriptstyle{\pm 0.39}$              | $6.55 \scriptscriptstyle{\pm 0.21}$               | $7.30 \scriptscriptstyle{\pm 0.15}$                    | $\mathbf{9.35 \scriptscriptstyle{\pm 0.17}}$           | $5.56 \scriptscriptstyle{\pm 0.33}$                    | $64.55 \scriptscriptstyle{\pm 0.33}$                    |
> > > | Mean-AcT         | ✓    | $\underline{1.21 \scriptscriptstyle{\pm 0.41}}$  | $\underline{5.09 \scriptscriptstyle{\pm 0.64}}$   | $\underline{5.73 \scriptscriptstyle{\pm 0.05}}$        | $9.83 \scriptscriptstyle{\pm 0.21}$                    | $5.71 \scriptscriptstyle{\pm 0.33}$                    | $64.22 \scriptscriptstyle{\pm 0.40}$                    |
> > > | Linear-AcT       | ✓    | $1.68 \scriptscriptstyle{\pm 0.48}$              | $6.47 \scriptscriptstyle{\pm 0.38}$               | $7.12 \scriptscriptstyle{\pm 0.26}$                    | $9.48 \scriptscriptstyle{\pm 0.19}$                    | $\underline{5.46 \scriptscriptstyle{\pm 0.44}}$        | $64.49 \scriptscriptstyle{\pm 0.38}$                    |
> > > | PID-AcT (Ours)   | ✓    | $\mathbf{0.72 \scriptscriptstyle{\pm 0.49}}$     | $\mathbf{4.36 \scriptscriptstyle{\pm 0.81}}$      | $\mathbf{4.90 \scriptscriptstyle{\pm 0.14}}$           | $9.56 \scriptscriptstyle{\pm 0.20}$                    | $6.08 \scriptscriptstyle{\pm 0.37}$                    | $64.50 \scriptscriptstyle{\pm 0.36}$                    |

---

> ### Author Response · Authors · 2025-11-22
> **Regarding Your Q1 and Q2: More Discussions and Empirical Evidence for the Stability Interval**
>
> Proposition 4 and Remark 3 (Appendix B.6) show that the PI gains that maximise the asymptotic convergence rate of the linearised error dynamics (the ``fastest convergence'' choice of $K_p$ and $K_i$) also induce a large overshoot in the correlation trajectory $\langle \bar e(0), \bar e(k)\rangle$. Empirically, this manifests as poorer steering performance: aggressive integral action speeds up convergence but increases oscillation and overshoot, which is consistent with classical PI tuning principles (see especially Ch. 3.3, p. 68 of Åström et al., "PID Controllers: Theory, Design, and Tuning", 2nd ed., ISA, 1995).
>
> Theorem 2 and Appendix B.7.2 further show that the derivative term $K_d$ provides damping: for fixed $K_p$ and $K_i$, increasing $K_d$ reduces the overshoot of the error trajectory. Although we do not establish a formal optimality theorem, it is natural to expect that the theoretically fastest PI gains, when combined with a suitably chosen derivative term, can outperform more conservative $K_i$ values that lie strictly inside the stability interval (under the same $K_p$, e.g., $K_p = 1$. Intuitively, because transformer depth is finite and we do not know at which layer the error will effectively settle, it is desirable to drive the error down quickly while preventing excessive overshoot, with $K_d$ acting as the compensating damping term.
>
> Our empirical results corroborate this interpretation. The stability intervals for $K_i$ at $K_p=1$ are $(-0.23\,0.23)$ for Gemma-2-9B-it and $(-0.1355\,0.1355)$ for Gemma-2-2B. In Figures 6(a) and 6(b) in our revised manuscript, we sweep across multiple $K_i$ values in these ranges. For example, for Gemma-2-9B-it using Llamaguard3 evaluation metrics (the upper-left panel of Figure 6), the $K_i$ that yields the fastest theoretical convergence rate is $0.056$. Holding $K_p=1$ and $K_i =0.056$, increasing $K_d$ from $0.0$ to $0.01$ raises the Llamaguard3 score from $76.61$ to $78.53$ (note that these additional results are not shown in Figure 6). This pattern suggests that with $K_d=0$, the aggressive integral term produces noticeable overshoot that harms performance, whereas adding a small derivative term introduces sufficient damping to recover, and in some cases improve, steering performance. The same pattern appears across the remaining three settings in Figure 6: the LLM-Judge metric for Gemma-2-9B-it and the CLS Toxicity and zero-shot Toxicity metrics for Gemma-2-2B. We have included these results in Appendix C.3 of our revised manuscript.
>
> **References**
>
> [13] Åström et al. "PID Controllers: Theory, Design, and Tuning", 1995.
>
> ---
> We hope our responses have addressed your concerns, and we would greatly appreciate any further feedback. We are happy to engage in follow-up discussions or clarify any additional points.

---

> > ### Author Response · Authors · 2025-11-27
> > **Additional Experimental Results: Further Comparison with State-of-the-Art Activation Steering Methods on Gemma2-9B-Instruct-With-Deeper-Safety-Alignment for the Jailbreaking Task**
> >
> > Dear Reviewer A2rW,
> >
> > We would like to thank the reviewer again for your thoughtful reviews and feedback.
> >
> > In Table 3 below, we further compare our PID steering with recent steering-vector generation methods on a better safety-aligned variant of Gemma-9B-IT, namely Gemma2-9B-Instruct-With-Deeper-Safety-Alignment [14]. Deeper safety alignment here trains the model to sustain refusal behavior beyond the first few tokens: instead of only shaping the opener, it conditions on partially harmful or misleading prefixes and optimizes the model to "recover" back to safe behavior later in the sequence. Practically, this extends safety pressure across positions so refusals remain stable under mild coercion, prefilling, or decoding variance, while preserving general-task utility. Under this strong safety-aligned regime, PID-based steering still achieves a non-trivial attack success rate and outperforms other state-of-the-art activation steering methods considered in our study.
> >
> > We have added these new results into Table 3 in Appendix C.2 of our revision.
> >
> > **Table 3**. Jailbreaking results on Gemma2-9B-Instruct-With-Deeper-Safety-Alignment.
> >
> > | Method     | ASR $\uparrow$        | tinyArc $\uparrow$  | tinyGSM8k strict $\uparrow$  | tinyMMLU $\uparrow$  | tinyTruthQA $\uparrow$  | tinyHellaSwag $\uparrow$  | tinyWinoGrande $\uparrow$  |
> > | ---------- | ------------ | --------- | ------------------ | ---------- | ------------- | --------------- | ---------------- |
> > | Original   | 1.22         | 69.07     | 82.93              | 76.18      | 54.82         | 82.27           | 72.11            |
> > | DIM        | 11.91         | 67.88     | 79.76              | 72.12      | 51.83         | 81.33           | 71.47            |
> > | ITI        | $\underline{23.12}$ | 68.19     | 81.41              | 75.22      | 53.05         | 81.58           | 71.79            |
> > | RePE       | 11.23        | 67.37     | 78.96              | 70.92      | 51.06         | 81.08           | 71.09            |
> > | PID (ours) | **34.75**    | 67.83     | 79.11              | 74.77      | 52.44         | 81.46           | 71.33            |
> >
> > We would be grateful if you could let us know whether our responses have satisfactorily addressed your concerns or if any questions remain regarding our submission or rebuttal.
> >
> > We would be happy to engage in any follow-up discussion or address any additional comments by the reviewer.
> >
> > If you feel our responses have fully addressed the concerns raised in your review, we would appreciate it if you could consider increasing your score to better reflect your current assessment. Thank you again for your time and thoughtful feedback.
> >
> >
> > **References**
> >
> > [14] Xiangyu Qi et al. "Safety alignment should be made more than just a few tokens deep". ICLR, 2025.

---

> ### Author Response · Authors · 2025-11-27
> **Reminder for Reviewer A2rW's Feedback**
>
> Dear Reviewer A2rW,
>
> Thank you again for the thoughtful feedback you provided earlier. Your comments have helped us strengthen the clarity and presentation of our work.
>
> We would like to gently remind you that we submitted our main rebuttal on November 18 (AoE) and subsequently added further experimental results on November 22 (AoE). As the discussion phase will close soon, specifically at 11:59 pm AoE on December 2, we want to ensure there is sufficient time to address any additional questions you may have. After this deadline, reviewer replies will no longer be possible, and we will not be able to respond after 11:59 pm AoE on December 3.
>
> We would appreciate it if you could let us know if our responses have addressed your concerns and whether you still have any other questions about our rebuttal, while the discussion is still open. We would be happy to do any follow-up discussion or address any additional comments.
>
> Thank you again for your time and thoughtful comments!
>
> Sincerely,
>
> Authors

---

### Official Review · Reviewer_hCNg · 2025-11-02

**Soundness:** 3
**Presentation:** 3
**Contribution:** 2
**Rating:** 6
**Confidence:** 4

**Summary:**

This paper introduces a control-theoretic framework for activation steering in LLMs, named PID Steering. The motivation here is that traditional activation steering methods, such as Activation Addition, Directional Ablation, and Mean Activation Transport,modify internal model activations to control behaviors like toxicity or refusal, but they lack formal guarantees and often rely on empirical heuristics.

Based on this the authors introduce PID for alignment, removing residual bias and mitigating overshoot when aligning.

Results demonstrate how PID control improves convergence and stability, reducing steady-state errors. Empirical experiments across multiple models (Gemma2, LLaMA3, Qwen2.5) and tasks (toxicity mitigation, jailbreak prevention, and image style control) show that PID Steering outperforms prior activation steering methods, achieving more consistent and interpretable control without harming overall model performance.

**Strengths:**

- The paper proposes an interesting methods of control-theoretic interpretation of activation steering.

- The proposed PID Steering introduces components that correct long-term drift and prevent oscillations making it robust.

- The approach is tested across diverse tasks (toxicity, jailbreaks, style transfer) and modalities (text and images), showing robustness and generality.

**Weaknesses:**

- Selecting appropriate PID gains (Kp, Ki, Kd) is nontrivial, and may result in suboptimal solutions.

- While results are broadly positive, the paper lacks detailed ablation studies on scenarios where PID might underperform or destabilize.

- Claims that PID Steering is lightweight are mentioned but not benchmarked in terms of latency or inference-time overhead.

- It is unclear to me how controller parameters generalize across model architectures or domains without retraining.

- The much smaller models such as Gemma is also used in this paper. Although, larger models (24B or 70B) are not tested, I was wondering if the authors could already make a size comparison between the models and present the observed differences?

**Questions:**

Please see the weaknesses.

---

> ### Author Response · Authors · 2025-11-19
> **Response to Reviewer hCNg (1)**
>
> Thank you for your thoughtful review and valuable feedback. Below we address your concerns.
>
> -----
> **Q1. Selecting appropriate PID gains (Kp, Ki, Kd) is nontrivial, and may result in suboptimal solutions.**
>
> **Answer:** Following the reviewer's suggestion, we have conducted a full ablation study on the gain parameters $K_p, K_i, K_d$ for LLM jailbreaking task on Gemma-2-9B-it and toxicity mitigation task for Gemma2-2B. We included the results in Figure 6 (Section 5.4) of the revised manuscript. The ablation heatmaps show a clear and consistent pattern across both the jailbreaking and toxicity mitigation tasks. First, $K_p$ is the primary driver of effect strength, and larger values (e.g., $K_p = 1.5$) consistently improve performance. In contrast, changes in $K_i$ and $K_d$ produce only small and smooth variations. Small non-zero  $K_i$ and $K_d$ help reduce disturbances and stabilize the steering dynamics, especially when $K_p$ is large.
>
> This behavior aligns with Theorem 1 and 2 in our manuscript, which show that the integral and derivative components dampen oscillations and smooth the update trajectory. Importantly, the ablation heatmaps demonstrate that PID-AcT is not hypersensitive to its gain parameters. Even relatively large values such as $K_i = 0.15$ or $K_d = 0.10$ do not destabilize the method, and the optimal region forms a broad plateau around $K_p = 1.5,\ K_i \in [0.03, 0.06],\ K_d = 0.01.$ Both tasks, Gemma-2-9B-it for LLM jailbreaking task and Gemma-2-2B for LLM toxicity mitigation task, exhibit flat performance landscapes rather than sharp peaks, indicating low sensitivity and a wide operating range.
>
> In addition to the ablation study $K_p, K_i, K_d$, in our original manuscript (Section 4 and Appendix B.3-B.7), we have derived the stability range for $K_p, K_i, K_d$, which are the values of those terms such that the closed-loop averaged error dynamics are input-to-state stable (ISS) with respect to the input disturbance (see Eqn.20 in Section 4.1). We summarize our results below.
>
> Given that $K_p$, $K_i$, $K_d$ are scalars,  $\|\cdot\|$ denotes the spectral norm of a matrix, and $|\cdot|$ denotes the absolute value of a scalar, we study the stability range for $K_p, K_i, K_d$.
>
> **(i) Stable $K_p$**
> Recall our set up and notations: we consider $N$ constrastive prompt pairs. For discrete time (layer) $k$, let $x_{i}^{\pm}(k)\in\mathbb{R}^d$ ($i=1,\dots,N$) denote the corresponding activation vectors. The layer-to-layer evolution is $x_{i}(k+1)=f_{i}^{(k)}\big(x(k)\big)$.
>
> Let $M= \sup\_k \left\|\frac{\sum\_{i=1}^N J\_{f^{(k)}_i}(x_i^+(k))}{N} \right\|$, where $J\_{f^{(k)}\_i}(x\_i^+(k))$ is the Jacobian of $f\_{i}^{(k)}$ at $x\_{i}^{+}(k)$.
>
> According to Proposition 1 - Appendix B.4, **the stability condition of gain $K_p$ is given by**
>     **$$1 - \frac{1}{M} < K_p < 1 + \frac{1}{M}.$$**
>
> Furthermore, $K_p=1$ yields fastest convergence of the homogeneous averaged-error dynamics (in the disturbance-free case) (Remark 1 - Appendix B.4)
>
> **(ii) Stable $K_i$**
> Given a stable $K_p$. According to Proposition 3 - Appendix B.5, **the stability condition of gain $K_i$ is**
>     **$$M|1-K_p| + M|K_i| < 1,$$**
>
> where $M$ is defined in part ''(i) Stable $K_p$''.
>
> $|K_i| = \frac{(1 - M|1-K_p|)^2}{4M}$ yields fastest convergence of the PI closed-loop towards zero in the homogeneous system (Remark 2 - Appendix B.5).
>
> **(iii) Stable $K_d$**
> Given stable $K_p$ and $K_i$. Regarding Theorem 1 - Appendix B.7, we establish stability condition for gain $K_d$ by following these steps:
> - Solving for $P(k)$ in the Lyapunov equation $$M_i(k)^\top P(k) M_i(k) - P(k) = -\mu I \qquad (\text{any scalar}\ \mu > 0)$$
> where $
> M_i(k) =
> \begin{bmatrix}
> (1-K_p)\bar{A}(k) & -K_i\bar{A}(k) \\\\
> I_d & I_d
> \end{bmatrix}
> $,  $\bar{A}(k)=\frac{\sum_{i=1}^N J_{f^{(k)}_i}(x_i^+(k))}{N}.$
>
> - Computing  $\varepsilon =\frac{\mu}{8  \sup_k \|P(k)\| \sup_k \|M_i(k)\|^2}$,
> and $r =\frac{\mu}{8\left(\sup_k \|\bar{A}(k)(1-K_p) - I\|^2+M|K_i|\right)}$, all are scalars. Note that $M$ is defined in part ''(i) Stable $K_p$''
>
> We then obtain **the stability condition of $K_d$, which is $|K_d|^2<\frac{r}{\left(\sup_k \|P(k)\|\left(1 + \frac{1}{\varepsilon}\right)+3r\right) M^2 }$**.

---

> ### Author Response · Authors · 2025-11-19
> **Response to Reviewer hCNg (2)**
>
> **Q2. While results are broadly positive, the paper lacks detailed ablation studies on scenarios where PID might underperform or destabilize.**
>
> **Answer**: We agree that understanding potential failure modes is important, and our ablation study, discussed in our answer to your Q1 above, directly addresses this concern by exploring a wide grid of proportional, integral, and derivative values. These results show that PID is generally stable, with underperformance occurring only in extreme settings, specifically when the integral or derivative terms are set high (e.g., $K_i, K_d \ge 0.1$). In these cases, the controller does not diverge or destabilize generation; rather, performance simply regresses toward that of a proportional-only baseline. This behavior aligns with control-theoretic intuition: $K_p$ drives the main effect, while small non-zero $K_i$ and $D$ primarily refine and stabilize the update. The recommended configuration ($K_p=1.5, K_i=0.01, K_d=0.01$) lies well within the empirically validated stability region. Overall, our ablations indicate that PID Steering is not fragile and maintains reliable performance across a wide range of settings, with clear guidance for avoiding the few suboptimal configurations.

---

> ### Author Response · Authors · 2025-11-19
> **Response to Reviewer hCNg (3)**
>
> **Q3. Claims that PID Steering is lightweight are mentioned but not benchmarked in terms of latency or inference-time overhead.**
>
> **Answer**:  Our method introduces no additional latency or inference-time overhead. It is used solely to compute the steering vectors/planes, which can be employed as a drop-in replacement for those used in prior work (e.g., [1, 2, 3]). This computation is performed once, offline, entirely separate from the inference pipeline. During inference, methods such as Directional Ablation [1], Angular Steering [2], and Activation Transport [3] simply utilize the pre-computed steering vectors/planes produced by our approach, thereby incurring no extra cost beyond what these frameworks already require.

---

> ### Author Response · Authors · 2025-11-19
> **Response to Reviewer hCNg (4)**
>
> **Q4. It is unclear to me how controller parameters generalize across model architectures or domains without retraining.**
>
> **Answer**: As noted in our response to your Q1, we have performed an ablation study on $K_p, K_i, K_d$ for both the jailbreak (Gemma-2-9B-it) and toxicity mitigation tasks (Gemma-2-2B) and included the results in Figure 6 of Section 5.4 of the revised manuscript. Our findings indicate that the optimal values of $K_p, K_i, K_d$ are similar across these tasks, despite using different models, Gemma-2-9B-it and gemma-2-B, suggesting that the controller parameters in our method generalize well across model architectures and task types.
>
> Additionally, as discussed in our response to Q1, we explain the stability range from which the controller parameters can be selected.
>
> Finally, we emphasize that PID Steering requires no training: once the steering vectors are computed via PID control, they can be directly reused at inference time.

---

> ### Author Response · Authors · 2025-11-19
> **Response to Reviewer hCNg (5)**
>
> **Q5. The much smaller models such as Gemma is also used in this paper. Although, larger models (24B or 70B) are not tested, I was wondering if the authors could already make a size comparison between the models and present the observed differences?**
>
> **Answer**: Thank you for the insightful comment. Following your suggestion, we conducted additional experiments on a larger model, Gemma2-27B Instruct, and report the results in Table 2 in Section 5.2 of the revised manuscript. For convenience, we also reproduce the results in **Table 1** below.
>
> **Table 1**. Gemma2-27B Instruct Results on Jailbreaking task.
>
>
> | Method      | ASR $\uparrow$   | tinyArc $\uparrow$ | tinyGSM8k strict $\uparrow$ | tinyMMLU $\uparrow$ | tinyTruthQA $\uparrow$ | tinyHellaSwag $\uparrow$ | tinyWinoGrande $\uparrow$ |
> |-------------|---------|-----------|---------------------|------------|----------------|------------------|-------------------|
> | Original    | --      | 73.45     | 86.91               | 76.11      | 61.36          | 83.24            | 75.47             |
> | DIM         | 74.03   | 72.13     | 84.70               | 74.59      | 59.49          | 81.79            | 74.21             |
> | ITI         | 37.36   | 72.84     | 86.38               | 75.51      | 60.85          | 82.83            | 75.11             |
> | RePE        | 24.19   | 71.03     | 84.00               | 73.90      | 58.79          | 80.82            | 73.44             |
> | PID         | **79.80** | 72.93   | 86.60               | 75.67      | 60.74          | 82.95            | 75.06             |
>
>
> Overall, we observe two notable trends as model size increases. First, weaker steering methods such as ITI and RePE exhibit substantial degradation in ASR performance, with ITI [4] dropping to 37.36 and RePE [5] to 24.19 on Gemma2-27B. In contrast, PID continues to scale effectively, achieving the strongest attack success rate (79.80) among all steering baselines. Second, larger models show greater robustness across general capabilities: across tinyArc, tinyGSM8k-strict, tinyMMLU, tinyTruthQA, tinyHellaSwag, and tinyWinoGrande, the performance variation induced by steering is noticeably smaller than in smaller models. This indicates that larger architectures tend to preserve their base competence even under strong steering interventions.
>
> Together, these results suggest that (i) the performance gap between PID and other baselines widens at larger scales, and (ii) large models exhibit improved resilience on general benchmarks, making PID’s stability and reduced overshoot especially beneficial in high-capacity settings.

---

> ### Author Response · Authors · 2025-11-19
> **Response to Reviewer hCNg (6)**
>
> **References**
>
> [1] Andy Arditi et al. "Refusal in language models is mediated by a single direction", NeurIPS, 2024.
>
> [2] Vu \& Nguyen. "Angular Steering: Behavior Control via Rotation in Activation Space", NeurIPS, 2025.
>
> [3] Rodrigue et al. "Controlling Language and Diffusion Models by Transporting Activations", ICLR, 2025,.
>
> [4] Li et al. “Inference-Time Intervention: Eliciting Truthful Answers from a Language Model”, NeurIPS, 2024.
>
> [5] Zou et al. "Representation Engineering: A Top-Down Approach to AI Transparency", Arxiv, 2023.
>
> -----
> We hope we have cleared your concerns about our work. We have also revised our manuscript according to your comments, and we would appreciate it if we could get your further feedback at your earliest convenience.

---

> ### Author Response · Authors · 2025-11-22
> **Regarding Your Q1: More Discussions and Empirical Evidence for the Stability Interval**
>
> Proposition 4 and Remark 3 (Appendix B.6) show that the PI gains that maximise the asymptotic convergence rate of the linearised error dynamics (the ``fastest convergence'' choice of $K_p$ and $K_i$) also induce a large overshoot in the correlation trajectory $\langle \bar e(0), \bar e(k)\rangle$. Empirically, this manifests as poorer steering performance: aggressive integral action speeds up convergence but increases oscillation and overshoot, which is consistent with classical PI tuning principles (see especially Ch. 3.3, p. 68 of Åström et al., "PID Controllers: Theory, Design, and Tuning", 2nd ed., ISA, 1995).
>
> Theorem 2 and Appendix B.7.2 further show that the derivative term $K_d$ provides damping: for fixed $K_p$ and $K_i$, increasing $K_d$ reduces the overshoot of the error trajectory. Although we do not establish a formal optimality theorem, it is natural to expect that the theoretically fastest PI gains, when combined with a suitably chosen derivative term, can outperform more conservative $K_i$ values that lie strictly inside the stability interval (under the same $K_p$, e.g., $K_p = 1$. Intuitively, because transformer depth is finite and we do not know at which layer the error will effectively settle, it is desirable to drive the error down quickly while preventing excessive overshoot, with $K_d$ acting as the compensating damping term.
>
> Our empirical results corroborate this interpretation. The stability intervals for $K_i$ at $K_p=1$ are $(-0.23\,0.23)$ for Gemma-2-9B-it and $(-0.1355\,0.1355)$ for Gemma-2-2B. In Figures 6(a) and 6(b) in our revised manuscript, we sweep across multiple $K_i$ values in these ranges. For example, for Gemma-2-9B-it using Llamaguard3 evaluation metrics (the upper-left panel of Figure 6), the $K_i$ that yields the fastest theoretical convergence rate is $0.056$. Holding $K_p=1$ and $K_i =0.056$, increasing $K_d$ from $0.0$ to $0.01$ raises the Llamaguard3 score from $76.61$ to $78.53$ (note that these additional results are not shown in Figure 6). This pattern suggests that with $K_d=0$, the aggressive integral term produces noticeable overshoot that harms performance, whereas adding a small derivative term introduces sufficient damping to recover, and in some cases improve, steering performance. The same pattern appears across the remaining three settings in Figure 6: the LLM-Judge metric for Gemma-2-9B-it and the CLS Toxicity and zero-shot Toxicity metrics for Gemma-2-2B. We have included these results in Appendix C.3 of our revised manuscript.
>
> **References**
>
> [6] Åström et al. "PID Controllers: Theory, Design, and Tuning", 1995.
>
> ---
> We hope our responses have addressed your concerns, and we would greatly appreciate any further feedback. We are happy to engage in follow-up discussions or clarify any additional points.

---

> ### Author Response · Authors · 2025-11-27
> **Reminder for Reviewer hCNg's Feedback**
>
> Dear Reviewer hCNg,
>
> Thank you again for the thoughtful feedback you provided earlier. Your comments have helped us strengthen the clarity and presentation of our work.
>
> We would like to gently remind you that we submitted our main rebuttal on November 18 (AoE) and subsequently added further experimental results on November 22 (AoE). As the discussion phase will close soon, specifically at 11:59 pm AoE on December 2, we want to ensure there is sufficient time to address any additional questions you may have. After this deadline, reviewer replies will no longer be possible, and we will not be able to respond after 11:59 pm AoE on December 3.
>
> We would appreciate it if you could let us know if our responses have addressed your concerns and whether you still have any other questions about our rebuttal, while the discussion is still open. We would be happy to do any follow-up discussion or address any additional comments.
>
> If you agree that our responses to your reviews have addressed the concerns you listed, we kindly ask that you consider whether raising your score would more accurately reflect your updated evaluation of our paper. Thank you again for your time and thoughtful comments!
>
> Sincerely,
>
> Authors

---

> > ### Comment · Reviewer_hCNg · 2025-11-27
> > **Response to the authors**
> >
> > Thank you for the responses and the new experiments. I believe these new experiments make your work more solid.
> >
> >  So if I understand correctly the optimal values for PID gains (Kp, Ki, Kd) shall remain the same across models and tasks? I was also wondering how you set these values before conducting the ablation study?

---

> ### Author Response · Authors · 2025-11-27
> **Additional Experimental Results: Further Comparison with State-of-the-Art Activation Steering Methods on Gemma2-9B-Instruct-With-Deeper-Safety-Alignment for the Jailbreaking Task**
>
> Dear Reviewer hCNg,
>
> We would like to thank the reviewer again for your thoughtful reviews and feedback.
>
> In Table 2 below, we further compare our PID steering with recent steering-vector generation methods on a better safety-aligned variant of Gemma-9B-IT, namely Gemma2-9B-Instruct-With-Deeper-Safety-Alignment [7]. Deeper safety alignment here trains the model to sustain refusal behavior beyond the first few tokens: instead of only shaping the opener, it conditions on partially harmful or misleading prefixes and optimizes the model to "recover" back to safe behavior later in the sequence. Practically, this extends safety pressure across positions so refusals remain stable under mild coercion, prefilling, or decoding variance, while preserving general-task utility. Under this strong safety-aligned regime, PID-based steering still achieves a non-trivial attack success rate and outperforms other state-of-the-art activation steering methods considered in our study.
>
> We have added these new results into Table 3 in Appendix C.2 of our revision.
>
> **Table 2**. Jailbreaking results on Gemma2-9B-Instruct-With-Deeper-Safety-Alignment.
>
> | Method     | ASR $\uparrow$        | tinyArc $\uparrow$  | tinyGSM8k strict $\uparrow$  | tinyMMLU $\uparrow$  | tinyTruthQA $\uparrow$  | tinyHellaSwag $\uparrow$  | tinyWinoGrande $\uparrow$  |
> | ---------- | ------------ | --------- | ------------------ | ---------- | ------------- | --------------- | ---------------- |
> | Original   | 1.22         | 69.07     | 82.93              | 76.18      | 54.82         | 82.27           | 72.11            |
> | DIM        | 11.91         | 67.88     | 79.76              | 72.12      | 51.83         | 81.33           | 71.47            |
> | ITI        | $\underline{23.12}$ | 68.19     | 81.41              | 75.22      | 53.05         | 81.58           | 71.79            |
> | RePE       | 11.23        | 67.37     | 78.96              | 70.92      | 51.06         | 81.08           | 71.09            |
> | PID (ours) | **34.75**    | 67.83     | 79.11              | 74.77      | 52.44         | 81.46           | 71.33            |
>
> We would be grateful if you could let us know whether our responses have satisfactorily addressed your concerns or if any questions remain regarding our submission or rebuttal.
>
> We would be happy to engage in any follow-up discussion or address any additional comments by the reviewer.
>
> If you feel our responses have fully addressed the concerns raised in your review, we would appreciate it if you could consider increasing your score to better reflect your current assessment. Thank you again for your time and thoughtful feedback.
>
>
> **References**
>
> [7] Xiangyu Qi et al. "Safety alignment should be made more than just a few tokens deep". ICLR, 2025.

---

> ### Author Response · Authors · 2025-11-27
>
> Thank you for the encouraging comments and the follow-up questions.
>
> **On your first question:** In general, the optimal PID gains $(K_p, K_i, K_d)$ are not universal since they depend on both the model architecture and the specific task or dataset. However, in our ablation study, which spans two model families and two tasks, we found that the near-optimal ranges for $(K_p, K_i, K_d)$ were remarkably similar across all settings (see Figure 6, Section 5.4 of the revised manuscript). This suggests a reasonable degree of robustness, though we do not claim that one fixed triplet will be optimal for all models or tasks beyond those we tested.
>
> **On how we selected values prior to the full ablation:** The main challenge was to obtain principled numerical ranges for $K_p$, $K_i$, and $K_d$ guided by theory rather than relying solely on trial-and-error tuning. As detailed in our earlier responses to your Q1 and in the manuscript (Proposition 1 in Appendix B.4, Proposition 3 in Appendix B.5, and Theorem 1 in Appendix B.7), we derived stability ranges for the PID gains. Since the complexity of deriving stability ranges grows from $K_p$ to $K_i$ to $K_d$, our methodology transitions accordingly from analytic derivations to more empirical selection. Our procedure for each term is explained below.
>
>
> For the proportional term $K_p$, the stability condition
>
> $$1 - \frac{1}{M} < K_p < 1 + \frac{1}{M},$$
>
> where $M > 0$ is defined in our reply to your Q1, implies that $K_p = 1$ lies comfortably within the stable region. This motivated us to center our ablation around values near $1$.
>
> For the integral gain $K_i$, the fastest convergence in the PI closed loop (in the homogeneous setting) occurs when
>
> $$|K_i| = \frac{(1 - M|1-K_p|)^2}{4M},$$
>
> as discussed in our reply to Q1 and Remark 2 (Appendix B.5). Fixing $K_p = 1$ gives $|K_i| = 1/(4M)$. Based on preliminary sweeps of $K_p$, we estimated that $M$ lies roughly between $1$ and $2$. This led us to initially explore $K_i$ in the range $(0.125, 0.25)$, avoiding values that would make the integral correction too weak or overly aggressive, and then refine via a small grid search. As a reminder, PID gains $(K_p, K_i, K_d)$ must be non-negative.
>
> We later computed the exact value of $K_i$ that achieves the fastest theoretical convergence rate, as well as the stability intervals for $K_i$, as detailed in our response *"Regarding Your Q1: More Discussions and Empirical Evidence for the Stability Interval."* For example, for Gemma-2-9B-it evaluated with Llamaguard3 (top-left panel of Figure 6), when $K_p = 1$,  the theoretically optimal value is $K_i = 0.056$, which lie within the $K_i$ sweep range used in our ablation study.
>
> For the derivative gain $K_d$, the theoretically optimal range involves solving a more complex system, so in practice, we began at $K_d = 0$ and gradually increased it to observe its empirical effects.

---

### Official Review · Reviewer_NQp6 · 2025-11-02

**Soundness:** 3
**Presentation:** 3
**Contribution:** 3
**Rating:** 4
**Confidence:** 3

**Summary:**

This paper presents a novel perspective on activation steering for large language models and diffusion models by introducing a control-theoretic framework for managing model behavior. The paper provides theoretical analysis using input-to-state stability ISS and demonstrates empirical gains on toxicity reduction, jailbreak resistance, and style control tasks across several models.

**Strengths:**

1. It is novel to connect the control theory and activation steering in LLMs.
2. The paper provides a solid mathematical formalization of steering as a dynamic control process.
3. Experiments are comprehensive, covering both text (toxicity, jailbreak) and vision (style transfer) domains.

**Weaknesses:**

1. Lack of baselines: The experiments compare primarily against the sequential steering vector ****baseline and, to a lesser extent, static activation addition. This is an insufficiently broad comparison. Many other steering methods, such as CAA and ITI, could serve as competitive baselines.
2. The paper evaluates toxicity using LLaMA-3-8B both as the generator and as the evaluator, which introduces a self-evaluation bias: the same model family that produces text also judges its toxicity. Such evaluations are not independent and can underreport toxicity. A separate toxicity classifier such as GPT-4 or at least cross-model correlation analysis should be used for reliable measurement.
3. The paper does not report ablations on the $K_P, K_I, K_D$, though the performance of control systems is typically sensitive to them. Similarly, computational overhead, such as inference latency and memory increase is unreported, which is important for large models.
4. The theoretical analysis assumes locally linearized activation dynamics (Eq. 29–32). While this yields tractable ISS proofs, the approximation error may be significant in deep nonlinear transformers, especially under strong steering perturbations.

**Questions:**

See above

---

> ### Author Response · Authors · 2025-11-19
> **Response to Reviewer NQp6 (1)**
>
> Thank you for your thoughtful review and valuable feedback. Below, we address your concerns.
>
> -----
>
> **Q1. Lack of baselines: The experiments compare primarily against the sequential steering vector baseline and, to a lesser extent, static activation addition. This is an insufficiently broad comparison. Many other steering methods, such as CAA and ITI, could serve as competitive baselines.**
>
> **Answer:** Thanks for the suggestion. Following your recommendation, we have added the results for ITI-C [1], ActAdd [2], and AURA [3] to Table 1 in the revised manuscript. We also copied those results in Tables 1 and 2 below.
>
> PID-AcT not only outperforms Mean-AcT [4] and Linear-AcT [4], but also exceeds stronger activation-editing baselines such as ActADD, AURA, and ITI-C. Whereas these baseline methods either plateau, raise perplexity sharply, or degrade accuracy, our PID-AcT delivers substantially larger reductions, up to **8.2x** on Gemma2-2B and **8.1x** on LLaMA3-8B, while keeping utility comparable. PID-AcT is the only method that achieves the best scores across both classifier-based and LLM-judge toxicity metrics, highlighting its robustness. This improvement stems from PID’s dynamic correction: the proportional term drives strong mitigation, while small integral and derivative terms stabilize the update, drive the steady-state error to 0, and avoid overshooting. Overall, PID-AcT provides the most stable and effective trade-off among all activation-intervention baselines. We will provide results for CAA in the next response round.

---

> ### Author Response · Authors · 2025-11-19
> **Response to Reviewer NQp6 (2)**
>
> **Table 1**:  Toxicity mitigation results for Gemma-2B, averaged over 10 runs. Lower is better for toxicity and perplexity; higher is better for MMLU. Best and second-best exclude the original baseline. (-) means no steering, (&#10003;) means sequential steering, and (&#10007;) means no sequential steering.
>
> | Method      | Seq. | CLS Tox $\downarrow$                             | 0-shot Tox $\downarrow$                           | PPL Wiki $\downarrow$                                  | PPL Mistral $\downarrow$                               | MMLU $\uparrow$                                       |
> |-------------|------|--------------------------------------------------|---------------------------------------------------|--------------------------------------------------------|--------------------------------------------------------|--------------------------------------------------------|
> | Original    | --   | $4.17 \scriptscriptstyle{\pm 0.32}$              | $13.42 \scriptscriptstyle{\pm 1.08}$              | $13.98$                                                | $6.68$                                                 | $53.10$                                                |
> | ActADD      | &#10007; | $3.96 \scriptscriptstyle{\pm 0.24}$          | $13.43 \scriptscriptstyle{\pm 1.42}$              | $14.69 \scriptscriptstyle{\pm 0.22}$                   | $\mathbf{6.67 \scriptscriptstyle{\pm 0.15}}$           | $\mathbf{53.00 \scriptscriptstyle{\pm 0.51}}$          |
> | AURA        | &#10007; | $2.12 \scriptscriptstyle{\pm 0.27}$          | $9.04 \scriptscriptstyle{\pm 0.66}$               | $\mathbf{14.18 \scriptscriptstyle{\pm 0.14}}$          | $7.04 \scriptscriptstyle{\pm 0.34}$                    | $\mathbf{53.00 \scriptscriptstyle{\pm 0.30}}$          |
> | ITI-C       | &#10007; | $0.74 \scriptscriptstyle{\pm 0.18}$          | $5.36 \scriptscriptstyle{\pm 0.91}$               | $14.90 \scriptscriptstyle{\pm 0.29}$                   | $7.44 \scriptscriptstyle{\pm 0.19}$                    | $52.60 \scriptscriptstyle{\pm 0.55}$                   |
> | Mean-Act    | &#10007; | $1.12 \scriptscriptstyle{\pm 0.23}$          | $5.20 \scriptscriptstyle{\pm 0.42}$               | $\underline{14.53 \scriptscriptstyle{\pm 0.21}}$       | $\underline{6.81 \scriptscriptstyle{\pm 0.19}}$        | $51.74 \scriptscriptstyle{\pm 0.55}$                   |
> | Linear-Act  | &#10007; | $0.95 \scriptscriptstyle{\pm 0.36}$          | $5.37 \scriptscriptstyle{\pm 0.80}$               | $14.75 \scriptscriptstyle{\pm 0.22}$                   | $7.24 \scriptscriptstyle{\pm 0.24}$                    | $51.63 \scriptscriptstyle{\pm 0.50}$                   |
> | Mean-Act    | &#10003; | $\underline{0.68 \scriptscriptstyle{\pm 0.21}}$ | $\underline{3.23 \scriptscriptstyle{\pm 0.44}}$ | $14.92 \scriptscriptstyle{\pm 0.25}$                   | $6.97 \scriptscriptstyle{\pm 0.74}$                    | $\underline{51.80 \scriptscriptstyle{\pm 0.55}}$       |
> | Linear-Act  | &#10003; | $1.00 \scriptscriptstyle{\pm 0.27}$          | $4.13 \scriptscriptstyle{\pm 0.89}$               | $14.98 \scriptscriptstyle{\pm 0.22}$                   | $7.13 \scriptscriptstyle{\pm 0.70}$                    | $51.47 \scriptscriptstyle{\pm 0.50}$                   |
> | PID-Act (Ours) | &#10003; | $\mathbf{0.51 \scriptscriptstyle{\pm 0.21}}$ | $\mathbf{2.90 \scriptscriptstyle{\pm 0.55}}$   | $15.22 \scriptscriptstyle{\pm 0.24}$                   | $7.02 \scriptscriptstyle{\pm 0.65}$                    | $51.30 \scriptscriptstyle{\pm 0.52}$                   |

---

> > ### Author Response · Authors · 2025-11-19
> > **Response to Reviewer NQp6 (3)**
> >
> > **Table 2**: Toxicity mitigation results for Llama-8B, averaged over 10 runs. Lower is better for toxicity and perplexity; higher is better for MMLU. Best and second-best exclude the Original baseline. (-) means no steering, (&#10003;) means sequential steering, and (&#10007;) means no sequential steering.
> >
> > | Method      | Seq. | CLS Tox $\downarrow$                             | 0-shot Tox $\downarrow$                           | PPL Wiki $\downarrow$                                  | PPL Mistral $\downarrow$                               | MMLU $\uparrow$                                       |
> > |-------------|------|--------------------------------------------------|---------------------------------------------------|--------------------------------------------------------|--------------------------------------------------------|--------------------------------------------------------|
> > | Original    | --   | $5.80$                                           | $15.00$                                           | $9.06$                                                 | $5.68$                                                 | $65.30$                                                |
> > | ActADD      | ✗    | $5.57 \scriptscriptstyle{\pm 0.45}$              | $15.73 \scriptscriptstyle{\pm 0.21}$              | $9.71 \scriptscriptstyle{\pm 0.46}$                    | $5.85 \scriptscriptstyle{\pm 0.26}$                    | $\mathbf{65.50 \scriptscriptstyle{\pm 0.34}}$          |
> > | AURA        | ✗    | $1.90 \scriptscriptstyle{\pm 0.61}$              | $8.12 \scriptscriptstyle{\pm 0.85}$               | $9.52 \scriptscriptstyle{\pm 0.32}$                    | $6.05 \scriptscriptstyle{\pm 0.30}$                    | $\mathbf{65.50 \scriptscriptstyle{\pm 0.33}}$          |
> > | ITI-C       | ✗    | $1.60 \scriptscriptstyle{\pm 0.22}$              | $6.53 \scriptscriptstyle{\pm 0.66}$               | ${9.48 \scriptscriptstyle{\pm 0.24}}$        | $6.17 \scriptscriptstyle{\pm 0.14}$                    | $\underline{64.70 \scriptscriptstyle{\pm 0.44}}$       |
> > | Mean-AcT    | ✗    | $1.78 \scriptscriptstyle{\pm 0.33}$              | $6.56 \scriptscriptstyle{\pm 0.54}$               | $\underline{9.36 \scriptscriptstyle{\pm 0.28}}$        | $\mathbf{5.45 \scriptscriptstyle{\pm 0.34}}$           | $64.35 \scriptscriptstyle{\pm 0.39}$                   |
> > | Linear-AcT  | ✗    | $1.87 \scriptscriptstyle{\pm 0.39}$              | $6.55 \scriptscriptstyle{\pm 0.21}$               | $\mathbf{9.35 \scriptscriptstyle{\pm 0.17}}$           | ${5.56 \scriptscriptstyle{\pm 0.33}}$        | $64.55 \scriptscriptstyle{\pm 0.33}$                   |
> > | Mean-AcT    | ✓    | $\underline{1.21 \scriptscriptstyle{\pm 0.41}}$  | $\underline{5.09 \scriptscriptstyle{\pm 0.64}}$   | $9.83 \scriptscriptstyle{\pm 0.21}$                    | $5.71 \scriptscriptstyle{\pm 0.33}$                    | $64.22 \scriptscriptstyle{\pm 0.40}$                   |
> > | Linear-AcT  | ✓    | $1.68 \scriptscriptstyle{\pm 0.48}$              | $6.47 \scriptscriptstyle{\pm 0.38}$               | $9.48 \scriptscriptstyle{\pm 0.19}$                    | $\underline{5.46 \scriptscriptstyle{\pm 0.44}}$        | ${64.49 \scriptscriptstyle{\pm 0.38}}$       |
> > | PID-AcT (Ours) | ✓ | $\mathbf{0.72 \scriptscriptstyle{\pm 0.49}}$     | $\mathbf{4.36 \scriptscriptstyle{\pm 0.81}}$      | $9.56 \scriptscriptstyle{\pm 0.20}$                    | $6.08 \scriptscriptstyle{\pm 0.37}$                    | $\mathbf{64.50 \scriptscriptstyle{\pm 0.36}}$          |

---

> ### Author Response · Authors · 2025-11-19
> **Response to Reviewer NQp6 (4)**
>
> **Q2. The paper evaluates toxicity using LLaMA-3-8B both as the generator and as the evaluator, which introduces a self-evaluation bias: the same model family that produces text also judges its toxicity. Such evaluations are not independent and can underreport toxicity. A separate toxicity classifier such as GPT-4 or at least cross-model correlation analysis should be used for reliable measurement.**
>
>
> **Answer**: Thanks to the reviewer for pointing this out. In Table 1 of our paper, we follow the toxicity mitigation experiments set up in [4], which evaluate toxicity using an ROBERTA-based classifier, as in [3]. In addition, they also measure toxicity in a 0-shot manner by querying Llama3-8B-instruct as LLM-as-a-judge. We will provide the results for LLama-3-8B in Table 1 of our revised manuscript, using QVQ-72B-Preview as the evaluator in the next response round.

---

> ### Author Response · Authors · 2025-11-19
> **Response to Reviewer NQp6 (5)**
>
> **Q3. The paper does not report ablations on the $K_p, K_i, K_d$, though the performance of control systems is typically sensitive to them.**
>
>
> **Answer:** Following the reviewer's suggestion, we have conducted a full ablation study on the gain parameters $K_p, K_i, K_d$ for the LLM jailbreaking task on Gemma-2-9B-it and the toxicity mitigation task for Gemma2-2B. We included the results in Figure 6 (Section 5.4) of the revised manuscript. The ablation heatmaps show a clear, consistent pattern across both the jailbreaking and the toxicity mitigation tasks. First, $K_p$ is the primary driver of effect strength, and larger values (e.g., $K_p = 1.5$) consistently improve performance. In contrast, changes in $K_i$ and $K_d$ produce only small and smooth variations. Small non-zero  $K_i$ and $K_d$ help reduce disturbances and stabilize the steering dynamics, especially when $K_p$ is large.
>
>
> This behavior aligns with Theorems 1 and 2 in our manuscript, which show that the integral and derivative components dampen oscillations and smooth the update trajectory. Importantly, the ablation heatmaps demonstrate that PID-AcT is not hypersensitive to its gain parameters. Even relatively large values such as $K_i = 0.15$ or $K_d = 0.10$ do not destabilize the method, and the optimal region forms a broad plateau around $K_p = 1.5,\ K_i \in [0.03, 0.06],\ K_d = 0.01.$ Both tasks, Gemma-2-9B-it for the LLM jailbreaking task and Gemma-2-2B for the LLM toxicity mitigation task, exhibit flat performance landscapes rather than sharp peaks, indicating low sensitivity and a wide operating range.
>
> In addition to the ablation study $K_p, K_i, K_d$, in our original manuscript (Section 4 and Appendix B.3-B.7), we have derived the stability range for $K_p, K_i, K_d$, which are the values of those terms such that the closed-loop averaged error dynamics are input-to-state stable (ISS) with respect to the input disturbance (see Eqn.20 in Section 4.1). We summarize our results below.
>
> Given that $K_p$, $K_i$, $K_d$ are scalars,  $\|\cdot\|$ denotes the spectral norm of a matrix, and $|\cdot|$ denotes the absolute value of a scalar, we study the stability range for $K_p, K_i, K_d$.
>
> **(i) Stable $K_p$**
> Recall our setup and notations: we consider $N$ contrastive prompt pairs. For discrete time (layer) $k$, let $x_{i}^{\pm}(k)\in\mathbb{R}^d$ ($i=1,\dots,N$) denote the corresponding activation vectors. The layer-to-layer evolution is $x_{i}(k+1)=f_{i}^{(k)}\big(x(k)\big)$.
>
> Let $M= \sup\_k \left\|\frac{\sum\_{i=1}^N J_{f^{(k)}\_i}(x_i^+(k))}{N}
> \right\|$, where $J\_{f^{(k)}_i}(x_i^+(k))$ is the Jacobian of $f\_{i}^{(k)}$ at $x\_{i}^{+}(k)$.
>
> According to Proposition 1 - Appendix B.4, **the stability condition of gain $K_p$ is given by**
>     **$$1 - \frac{1}{M} < K_p < 1 + \frac{1}{M}.$$**
>
> Furthermore, $K_p=1$ yields the fastest convergence of the homogeneous averaged-error dynamics (in the disturbance-free case) (Remark 1 - Appendix B.4)
>
> **(ii) Stable $K_i$**
> Given a stable $K_p$. According to Proposition 3 - Appendix B.5, **the stability condition of gain $K_i$ is**
>     **$$M|1-K_p| + M|K_i| < 1,$$**
>
> where $M$ is defined in part ''(i) Stable $K_p$''.
>
> $|K_i| = \frac{(1 - M|1-K_p|)^2}{4M}$ yields the fastest convergence of the PI closed-loop towards zero in the homogeneous system (Remark 2 - Appendix B.5).
>
> **(iii) Stable $K_d$**
> Given stable $K_p$ and $K_i$. Regarding Theorem 1 - Appendix B.7, we establish a stability condition for gain $K_d$ by following these steps:
> - Solving for $P(k)$ in the Lyapunov equation $$M_i(k)^\top P(k)\, M_i(k) - P(k) = -\mu I \qquad (\text{any scalar}\ \mu > 0)$$
> where
> $
> M_i(k) =
> \begin{bmatrix}
> (1-K_p)\bar{A}(k) & -K_i\bar{A}(k) \\\\
> I_d & I_d
> \end{bmatrix}
> $, $\bar{A}(k)=\frac{\sum_{i=1}^N J_{f^{(k)}_i}(x_i^+(k))}{N}.$
>
> - Computing  $\varepsilon =\frac{\mu}{8 \sup_k \|P(k)\| \sup_k \|M_i(k)\|^2}$,
> and $r =\frac{\mu}{8\left(\sup_k \|\bar{A}(k)(1-K_p) - I\|^2+M|K_i|\right)}$, all are scalars. Note that $M$ is defined in part ''(i) Stable $K_p$''
>
> We then obtain **the stability condition of $K_d$, which is $|K_d|^2<\frac{r}{\left(\sup_k \|P(k)\|\left(1 + \frac{1}{\varepsilon}\right)+3r\right) M^2 }$**.

---

> ### Author Response · Authors · 2025-11-19
> **Response to Reviewer NQp6 (6)**
>
> **Q4. Similarly, computational overhead, such as inference latency and memory increase, is unreported, which is important for large models.**
>
> **Answer:** Regarding inference latency and memory usage, our method introduces no additional computational overhead. It is used solely to compute the steering vectors/planes, which can be employed as a drop-in replacement for those used in prior work (e.g., [4, 5, 6]). This computation is performed once, offline, entirely separate from the inference pipeline. During inference, methods such as Directional Ablation [5], Angular Steering [6], and Activation Transport [4] simply utilize the pre-computed steering vectors/planes produced by our approach, thereby incurring no extra cost beyond what these frameworks already require.

---

> ### Author Response · Authors · 2025-11-19
> **Response to Reviewer NQp6 (7)**
>
> **Q5. The theoretical analysis assumes locally linearized activation dynamics (Eq. 29–32). While this yields tractable ISS proofs, the approximation error may be significant in deep nonlinear transformers, especially under strong steering perturbations.**
>
> **Answer:** We thank the reviewer for raising this important concern about the validity of our locally linearized activation dynamics, especially in the presence of strong steering. Below, we explain why we adopt this linearization and how the analysis changes when we retain the nonlinear terms
>
> Recall our setup and notations: we consider $N$ contrastive prompt pairs. For discrete time (layer) $k$, let $x_{i}^{\pm}(k)\in\mathbb{R}^d$ ($i=1,\dots,N$) denote the corresponding activation vectors. The layer-to-layer evolution is $x_{i}(k+1)=f_{i}^{(k)}\big(x_i(k)\big)$.
>
> **1. Why do we linearize?**
>
> **(i)**  We aim to analyse how steered activations $\{x_i^{-}(k)\}$ track the desired activations $\{x_i^{+}(k)\}$, where the steered activations evolve according to
> $$
> x_i^{-}(k) = f_i^{(k-1)}\big(x_i^{-}(k-1) + u(k-1)\big)
> $$
> To obtain a tractable model, we linearize $f_i^{(k)}$ around the desired activation $x_i^{+}(k)$. This local linearization is a natural choice for studying the behaviour of the system around the desired trajectory (see [8],[9]). The influence of higher-order nonlinear terms in P, PI, and PID controllers will be discussed in Part 3 of our response.
>
> **(ii)**  Our steering law employs a linear PID controller, meaning that the control input is linear in the averaged error. Classical linear control methods, including P, PI and PID, are based on linear models of system dynamics around the desired operating point (see [10, Ch. 2.1, p. 4]; [11, Ch. 1.2, p. 5]; [12, Ch. 1, p. 1]; [13, Ch. 4.1.1, p. 14]). In fact, the standard assumption when designing a linear PID controller for a nonlinear plant is first to obtain the linearized approximation around the chosen operating conditions (see [14]). Following this established practice, we analyse the effect of the PID steering law by linearizing the dynamics around desired setpoints.
>
> **(iii)**  As the reviewer noted, ISS theory is clean for a linear time-varying system with an additive disturbance. Without this local linear model of the activation dynamics, it becomes extremely difficult to derive explicit and interpretable stability conditions on the controller gains $K_p, K_i, K_d$.
>
> **2. What would change if we do not ignore the nonlinear terms?**
>
> Consider the exact Taylor expansion of the activation map:
> $$
> f_i^{(k)} \big(x_i^{+}(k) + \delta_i(k)\big)
> = f_i^{(k)}\big(x_i^{+}(k)\big)+ J_{f_i^{(k)}}\big(x_i^{+}(k)\big) \delta_i(k)+ O(\lVert\delta_i(k)\rVert^{2}).
> $$
>
> This yields the averaged error dynamics:
> $$
> \bar e(k+1) = \bar A(k) \bar e(k) - \bar A(k) u(k) + w(k),
> $$
>
> where
> $$
> \bar e(k)=\frac{1}{N}\sum_{i=1}^N e_i(k)=\frac{1}{N}\sum_{i=1}^N x_i^+(k)-x_i^-(k),$$
>
> $$
> \widetilde{e}_i(k)=e_i(k)-\bar e(k),
> \qquad
> \delta_i(k)=-e_i(k)+u(k),$$
>
> $$
> w(k) =
> \frac{1}{N}\sum\_{i=1}^{N} \widetilde{A}\_i(k)\,\widetilde{e}\_i(k)
> +
> \frac{1}{N}\sum_{i=1}^{N} O(\lVert -e_i(k)+u(k)\rVert^{2})
> $$
>
> $$
> \bar A(k) = \frac{1}{N}\sum_{i=1}^N A_i(k)
> = \frac{1}{N}\sum_{i=1}^N J_{f_i^{(k)}}(x_i^{+}(k)),
> \quad
> \widetilde{A}_i(k) = A_i(k) - \bar A(k).
> $$
>
>
>
> The Taylor remainder terms simply contribute to the disturbance $w(k)$. Even if these higher-order terms are not small, as long as they remain bounded across layers, all the ISS results continue to hold. Their only effect is that the ISS upper bounds on $\lVert\bar e(k)\rVert$ increase proportionally to $\lVert w\rVert_{\infty}$.
>
> **3. Is $\lVert\delta_i(k)\rVert = \lVert -e_i(k) + u(k)\rVert$ significant, especially with strong steering?**
>
> One might worry that when strong steering is applied, $\lVert e_i(k)\rVert$ become large, which could in turn make $\lVert\delta_i(k)\rVert$ large, where $\delta_i(k) = -e_i(k) + u(k)$ as in line 918 in our manuscript. However, the key conceptual point is
> $$
> \delta_i(k) = -e_i(k) + u(k)
> = -\big(x_i^{+}(k) - x_i^{-}(k)\big) + u(k),
> $$
> implying
> $$
> x_i^{-}(k) + u(k) = x_i^{+}(k) + \delta_i(k).
> $$
> Thus, $\delta_i(k)$ *measures* the residual mismatch between the desired activation $x_i^{+}(k)$ and the steered activation $x_i^{-}(k) + u(k)$ at the same layer. When the controller functions properly, the steered activation remains close to the desired trajectory, and therefore $\delta_i(k)$ is small in norm by design. We analyse $\delta_i(k)$ in each control type as follows.
>
> **(i)  P control (scalar gain for simplicity):**
> Following equations 26 and 33 in our manuscript, we have:
> $$
> u(k)=K_p \bar e(k),
> \qquad
> \widetilde{e}_i(k)=e_i(k)-\bar e(k).
> $$
> Therefore,
> $$
> \delta_i(k) = (K_p - 1)\bar e(k) - \tilde e_i(k),
> $$
> where $\tilde e_i(k)$ is the heterogeneity component.

---

> ### Author Response · Authors · 2025-11-19
> **Response to Reviewer NQp6 (8)**
>
> Our stability condition requires
> $$
> 1 - \frac{1}{M} < K_p < 1 + \frac{1}{M},
> $$
>
> so $K_p - 1$ is necessarily small. This cancels most of the contribution of $\bar e(k)$, meaning that $\lVert\delta_i(k)\rVert$ is of the same order as the heterogeneity $\lVert\tilde e_i(k)\rVert$, not the overall steering magnitude.
>
> **(ii)  PI and PID control:**
> For PI control,
> $$
> u(k) = K_p \bar e(k) + K_i s(k),
> \qquad
> s(k) = \sum_{j=0}^{k-1} \bar e(j),
> $$
> so
> $$
> \delta_i(k)
> = (K_p - 1) \bar e(k) + K_i s(k) - \tilde e_i(k).
> $$
>
> For PID control,
> $$
> u(k) = K_p \bar e(k) + K_i s(k) + K_d \Delta\bar e(k),
> \qquad
> \Delta\bar e(k)=\bar e(k)-\bar e(k-1),
> $$
> so
> $$
> \delta_i(k)
> = (K_p - 1) \bar e(k)+ K_i s(k)+ K_d \Delta\bar e(k)- \tilde e_i(k).
> $$
>
> $u(k)$ contains additional terms depending on $s(k)$ and  $\Delta \bar e(k)$. All of these contributions remain linear in $\bar e(\cdot)$. Our gain conditions guarantee closed-loop stability, ensuring that $\bar e$, $s$, and $\Delta \bar e$ remain bounded. Thus, $u(k)$ tracks the mean error and remains in a comparable range, and it does not explode. So is $\delta_i(k)$.
>
> We have added an analysis without using the local linearizations in (2) and (3) above in Appendix B.8 of the revised manuscript.

---

> ### Author Response · Authors · 2025-11-19
> **Response to Reviewer NQp6 (9)**
>
> **References**
>
> [1] Li et al. “Inference-Time Intervention: Eliciting Truthful Answers from a Language Model”, NeurIPS, 2024.
>
> [2] Turner et al. “Steering Language Models with Activation Engineering”, OpenReview preprint, 2025.
>
> [3] Suau et al. “Whispering Experts: Neural Interventions for Toxicity Mitigation in Language Models”, ICML, 2024.
>
> [4] Rodriguez et al. “Controlling Language and Diffusion Models by Transporting Activations”, ICLR, 2025.
>
> [5] Arditi et al. “Refusal in Language Models Is Mediated by a Single Direction”, NeurIPS, 2024.
>
> [6] Vu and Nguyen “Angular Steering: Behavior Control via Rotation in Activation Space”, ICML Workshop on Reliable and Responsible Foundation Models, 2025.
>
> [8] Seiler et al.  “Trajectory-based robustness analysis for nonlinear systems,”  International Journal of Robust and Nonlinear Control, 2023, 34(2), pp. 910–926.
>
> [9] Biertümpfel et al. “Robustness Analysis of Nonlinear Systems Along Uncertain Trajectories,”  2023, 56, pp. 5831–5836.
>
> [10] Sonugür. "A Review of quadrotor UAV: Control and SLAM methodologies ranging from conventional to innovative approaches", Robotics and Autonomous Systems, 2023.
>
> [11] Wu et al. “Advanced control of power electronic systems-An overview of methods”,  In: Control of Power Electronic Converters and Systems, Academic Press, 2021, pp. 1–33.
>
> [12] Eker et al. “Linear control of nonlinear systems: Interplay between nonlinearity and feedback”,  AIChE Journal, 2002.
>
> [13] Boscaino et al. (2024). “Grid-connected photovoltaic inverters: Grid codes, topologies and control techniques,” Renewable and Sustainable Energy Reviews, 2024.
>
> [14] Wu et al. "PID Control of Nonlinear Systems",  In: PID Control System Design and Automatic Tuning using MATLAB/Simulink, 2020, pp. 179-202.
>
> -----
> We hope we have cleared your concerns about our work. We have also revised our manuscript according to your comments, and we would appreciate it if we could get your further feedback at your earliest convenience.

---

> > ### Author Response · Authors · 2025-11-21
> > **Regarding Q1 and Q2: Additional Results on CAA and QVQ-72B-Preview as an Evaluator for LLM Toxicity (1)**
> >
> > Below are the updated versions of Tables 1 and 2 in our answer to your Q1 above, now including results for the CAA method and QVQ-72B-Preview evaluation scores. Overall, the findings remain consistent with our original conclusions: PID-AcT achieves the strongest toxicity reduction with minimal utility loss across both models. In our revised manuscript, we have updated Table 1 and Section 5.1 accordingly to reflect these additions.

---

> > > ### Author Response · Authors · 2025-11-21
> > > **Regarding Q1 and Q2: Additional Results on CAA and QVQ-72B-Preview as an Evaluator for LLM Toxicity (2)**
> > >
> > > **Table 1 (updated)**:  Toxicity mitigation results for Gemma-2B, averaged over 10 runs. Lower is better for toxicity and perplexity; higher is better for MMLU. Best and second-best exclude the original baseline. (-) means no steering, (&#10003;) means sequential steering, and (&#10007;) means no sequential steering.
> > >
> > > | Method    | Seq. | CLS Tox $\downarrow$    | 0-shot Tox $\downarrow$    | QVQ $\downarrow$    | PPL Wiki $\downarrow$    | PPL Mistral $\downarrow$    | MMLU $\uparrow$    |
> > > |-|-|-|-|-|-|-|-|
> > > | Original    | --    | $4.17 \scriptscriptstyle{\pm 0.32}$    | $13.42 \scriptscriptstyle{\pm 1.08}$    | $14.17 \scriptscriptstyle{\pm 0.08}$    | $13.98$    | $6.68$    | $53.10$    |
> > > | ActADD    | &#10007; | $3.96 \scriptscriptstyle{\pm 0.24}$    | $13.43 \scriptscriptstyle{\pm 1.42}$    | $14.17 \scriptscriptstyle{\pm 0.08}$    | $14.69 \scriptscriptstyle{\pm 0.22}$                   | $\mathbf{6.67 \scriptscriptstyle{\pm 0.15}}$    | $\mathbf{53.00 \scriptscriptstyle{\pm 0.51}}$    |
> > > | CAA    | &#10007; | $1.20 \scriptscriptstyle{\pm 0.25}$    | $5.35 \scriptscriptstyle{\pm 0.50}$    | $5.88 \scriptscriptstyle{\pm 0.36}$    | $\mathbf{14.60 \scriptscriptstyle{\pm 0.20}}$    | $6.85 \scriptscriptstyle{\pm 0.22}$    | 51.70 $\scriptscriptstyle{\pm 0.48}$    |
> > > | AURA    | &#10007; | $2.12 \scriptscriptstyle{\pm 0.27}$    | $9.04 \scriptscriptstyle{\pm 0.66}$    | $9.72 \scriptscriptstyle{\pm 0.27}$    | $\mathbf{14.18 \scriptscriptstyle{\pm 0.14}}$    | $7.04 \scriptscriptstyle{\pm 0.34}$    | $\mathbf{53.00 \scriptscriptstyle{\pm 0.30}}$    |
> > > | ITI-C    | &#10007; | $0.74 \scriptscriptstyle{\pm 0.18}$    | $5.36 \scriptscriptstyle{\pm 0.91}$    | $6.10 \scriptscriptstyle{\pm 0.13}$    | $14.90 \scriptscriptstyle{\pm 0.29}$    | $7.44 \scriptscriptstyle{\pm 0.19}$    | $52.60 \scriptscriptstyle{\pm 0.55}$    |
> > > | Mean-Act    | &#10007; | $1.12 \scriptscriptstyle{\pm 0.23}$    | $5.20 \scriptscriptstyle{\pm 0.42}$    | $5.80 \scriptscriptstyle{\pm 0.15}$    | $\underline{14.53 \scriptscriptstyle{\pm 0.21}}$    | $\underline{6.81 \scriptscriptstyle{\pm 0.19}}$    | $51.74 \scriptscriptstyle{\pm 0.55}$    |
> > > | Linear-Act  | &#10007; | $0.95 \scriptscriptstyle{\pm 0.36}$          | $5.37 \scriptscriptstyle{\pm 0.80}$               | $5.92 \scriptscriptstyle{\pm 0.11}$    | $14.75 \scriptscriptstyle{\pm 0.22}$                   | $7.24 \scriptscriptstyle{\pm 0.24}$                    | $51.63 \scriptscriptstyle{\pm 0.50}$                   |
> > > | Mean-Act    | &#10003; | $\underline{0.68 \scriptscriptstyle{\pm 0.21}}$ | $\underline{3.23 \scriptscriptstyle{\pm 0.44}}$ | $\underline{3.70 \scriptscriptstyle{\pm 0.14}}$    | $14.92 \scriptscriptstyle{\pm 0.25}$                   | $6.97 \scriptscriptstyle{\pm 0.74}$                    | $\underline{51.80 \scriptscriptstyle{\pm 0.55}}$       |
> > > | Linear-Act  | &#10003; | $1.00 \scriptscriptstyle{\pm 0.27}$          | $4.13 \scriptscriptstyle{\pm 0.89}$               | $4.64 \scriptscriptstyle{\pm 0.04}$    | $14.98 \scriptscriptstyle{\pm 0.22}$                   | $7.13 \scriptscriptstyle{\pm 0.70}$                    | $51.47 \scriptscriptstyle{\pm 0.50}$                   |
> > > | PID-Act (Ours) | &#10003; | $\mathbf{0.51 \scriptscriptstyle{\pm 0.21}}$ | $\mathbf{2.90 \scriptscriptstyle{\pm 0.55}}$   | $\mathbf{3.40 \scriptscriptstyle{\pm 0.04}}$    | $15.22 \scriptscriptstyle{\pm 0.24}$                   | $7.02 \scriptscriptstyle{\pm 0.65}$                    | $51.30 \scriptscriptstyle{\pm 0.52}$                   |

---

> > > > ### Author Response · Authors · 2025-11-21
> > > > **Regarding Q1 and Q2: Additional Results on CAA and QVQ-72B-Preview as an Evaluator for LLM Toxicity (3)**
> > > >
> > > > **Table 2 (updated)**: Toxicity mitigation results for Llama-8B, averaged over 10 runs. Lower is better for toxicity and perplexity; higher is better for MMLU. Best and second-best exclude the Original baseline. (–) means no steering, (✓) means sequential steering, and (✗) means no sequential steering.
> > > >
> > > > | Method           | Seq. | CLS Tox $\downarrow$                             | 0-shot Tox $\downarrow$                           | QVQ $\downarrow$                                       | PPL Wiki $\downarrow$                                  | PPL Mistral $\downarrow$                               | MMLU $\uparrow$                                        |
> > > > |------------------|------|--------------------------------------------------|---------------------------------------------------|--------------------------------------------------------|--------------------------------------------------------|--------------------------------------------------------|---------------------------------------------------------|
> > > > | Original         | --   | $5.80$                                           | $15.00$                                           | $15.81 \scriptscriptstyle{\pm 0.09}$                   | $9.06$                                                 | $5.68$                                                 | $65.30$                                                 |
> > > > | ActADD           | ✗    | $5.57 \scriptscriptstyle{\pm 0.45}$              | $15.73 \scriptscriptstyle{\pm 0.21}$              | $16.48 \scriptscriptstyle{\pm 0.19}$                   | $9.71 \scriptscriptstyle{\pm 0.46}$                    | $5.85 \scriptscriptstyle{\pm 0.26}$                    | $\mathbf{65.50 \scriptscriptstyle{\pm 0.34}}$           |
> > > > | CAA              | ✗    | $1.82 \scriptscriptstyle{\pm 0.36}$              | $6.70 \scriptscriptstyle{\pm 0.58}$               | $7.40 \scriptscriptstyle{\pm 0.06}$                    | $9.40 \scriptscriptstyle{\pm 0.25}$                    | $5.50 \scriptscriptstyle{\pm 0.30}$                    | $64.30 \scriptscriptstyle{\pm 0.37}$                    |
> > > > | AURA             | ✗    | $1.90 \scriptscriptstyle{\pm 0.61}$              | $8.12 \scriptscriptstyle{\pm 0.85}$               | $8.80 \scriptscriptstyle{\pm 0.17}$                    | $9.52 \scriptscriptstyle{\pm 0.32}$                    | $6.05 \scriptscriptstyle{\pm 0.30}$                    | $\mathbf{65.50 \scriptscriptstyle{\pm 0.33}}$           |
> > > > | ITI-C            | ✗    | $1.60 \scriptscriptstyle{\pm 0.22}$              | $6.53 \scriptscriptstyle{\pm 0.66}$               | $7.19 \scriptscriptstyle{\pm 0.06}$                    | $9.48 \scriptscriptstyle{\pm 0.24}$                    | $6.17 \scriptscriptstyle{\pm 0.14}$                    | $\underline{64.70 \scriptscriptstyle{\pm 0.44}}$        |
> > > > | Mean-AcT         | ✗    | $1.78 \scriptscriptstyle{\pm 0.33}$              | $6.56 \scriptscriptstyle{\pm 0.54}$               | $7.30 \scriptscriptstyle{\pm 0.25}$                    | $\underline{9.36 \scriptscriptstyle{\pm 0.28}}$        | $\mathbf{5.45 \scriptscriptstyle{\pm 0.34}}$           | $64.35 \scriptscriptstyle{\pm 0.39}$                    |
> > > > | Linear-AcT       | ✗    | $1.87 \scriptscriptstyle{\pm 0.39}$              | $6.55 \scriptscriptstyle{\pm 0.21}$               | $7.30 \scriptscriptstyle{\pm 0.15}$                    | $\mathbf{9.35 \scriptscriptstyle{\pm 0.17}}$           | $5.56 \scriptscriptstyle{\pm 0.33}$                    | $64.55 \scriptscriptstyle{\pm 0.33}$                    |
> > > > | Mean-AcT         | ✓    | $\underline{1.21 \scriptscriptstyle{\pm 0.41}}$  | $\underline{5.09 \scriptscriptstyle{\pm 0.64}}$   | $\underline{5.73 \scriptscriptstyle{\pm 0.05}}$        | $9.83 \scriptscriptstyle{\pm 0.21}$                    | $5.71 \scriptscriptstyle{\pm 0.33}$                    | $64.22 \scriptscriptstyle{\pm 0.40}$                    |
> > > > | Linear-AcT       | ✓    | $1.68 \scriptscriptstyle{\pm 0.48}$              | $6.47 \scriptscriptstyle{\pm 0.38}$               | $7.12 \scriptscriptstyle{\pm 0.26}$                    | $9.48 \scriptscriptstyle{\pm 0.19}$                    | $\underline{5.46 \scriptscriptstyle{\pm 0.44}}$        | $64.49 \scriptscriptstyle{\pm 0.38}$                    |
> > > > | PID-AcT (Ours)   | ✓    | $\mathbf{0.72 \scriptscriptstyle{\pm 0.49}}$     | $\mathbf{4.36 \scriptscriptstyle{\pm 0.81}}$      | $\mathbf{4.90 \scriptscriptstyle{\pm 0.14}}$           | $9.56 \scriptscriptstyle{\pm 0.20}$                    | $6.08 \scriptscriptstyle{\pm 0.37}$                    | $64.50 \scriptscriptstyle{\pm 0.36}$                    |

---

> ### Author Response · Authors · 2025-11-22
> **Regarding Your Q3: More Discussions and Empirical Evidence for the Stability Interval**
>
> Proposition 4 and Remark 3 (Appendix B.6) show that the PI gains that maximise the asymptotic convergence rate of the linearised error dynamics (the ``fastest convergence'' choice of $K_p$ and $K_i$) also induce a large overshoot in the correlation trajectory $\langle \bar e(0), \bar e(k)\rangle$. Empirically, this manifests as poorer steering performance: aggressive integral action speeds up convergence but increases oscillation and overshoot, which is consistent with classical PI tuning principles (see especially Ch. 3.3, p. 68 of Åström et al., "PID Controllers: Theory, Design, and Tuning", 2nd ed., ISA, 1995).
>
> Theorem 2 and Appendix B.7.2 further show that the derivative term $K_d$ provides damping: for fixed $K_p$ and $K_i$, increasing $K_d$ reduces the overshoot of the error trajectory. Although we do not establish a formal optimality theorem, it is natural to expect that the theoretically fastest PI gains, when combined with a suitably chosen derivative term, can outperform more conservative $K_i$ values within its stability interval (under  $K_p = 1$ for the fastest convergence P- control, see Remark 1 in Appendix B.4) . Intuitively, because transformer depth is finite and we do not know at which layer the error will effectively settle, it is desirable to drive the error down quickly while preventing excessive overshoot, with $K_d$ acting as the compensating damping term.
>
> Our empirical results corroborate this interpretation. The stability intervals for $K_i$ at $K_p=1$ are $(-0.23\,0.23)$ for Gemma-2-9B-it and $(-0.1355\,0.1355)$ for Gemma-2-2B. In Figures 6(a) and 6(b) in our revised manuscript, we sweep across multiple $K_i$ values in these ranges. For example, for Gemma-2-9B-it, which uses Llamaguard3 evaluation metrics (the upper-left panel of Figure 6), the $K_i$ that yields the fastest theoretical convergence rate is $0.056$. Holding $K_p=1$ and $K_i =0.056$, increasing $K_d$ from $0.0$ to $0.01$ raises the Llamaguard3 score from $76.61$ to $78.53$ (note that these additional results are not shown in Figure 6). This pattern suggests that with $K_d=0$, the aggressive integral term produces noticeable overshoot that harms performance, whereas adding a small derivative term introduces sufficient damping to recover, and in some cases improve, steering performance. The same pattern appears across the remaining three settings in Figure 6: the LLM-Judge metric for Gemma-2-9B-it and the CLS Toxicity and zero-shot Toxicity metrics for Gemma-2-2B. We have included these results in Appendix C.3 of our revised manuscript.
>
> **References**
>
> [15] Åström et al. "PID Controllers: Theory, Design, and Tuning", 1995.
>
> ---
> We hope our responses have addressed your concerns, and we would greatly appreciate any further feedback. We are happy to engage in follow-up discussions or clarify any additional points.

---

> ### Author Response · Authors · 2025-11-27
> **Reminder for Reviewer NQp6's Feedback**
>
> Dear Reviewer NQp6,
>
> Thank you again for the thoughtful feedback you provided earlier. Your comments have helped us strengthen the clarity and presentation of our work.
>
> We would like to gently remind you that we submitted our main rebuttal on November 18 (AoE) and subsequently added further experimental results on November 22 (AoE). As the discussion phase will close soon, specifically at 11:59 pm AoE on December 2, we want to ensure there is sufficient time to address any additional questions you may have. After this deadline, reviewer replies will no longer be possible, and we will not be able to respond after 11:59 pm AoE on December 3.
>
> We would appreciate it if you could let us know if our responses have addressed your concerns and whether you still have any other questions about our rebuttal, while the discussion is still open. We would be happy to do any follow-up discussion or address any additional comments.
>
> If you agree that our responses to your reviews have addressed the concerns you listed, we kindly ask that you consider whether raising your score would more accurately reflect your updated evaluation of our paper. Thank you again for your time and thoughtful comments!
>
> Sincerely,
>
> Authors

---

> ### Author Response · Authors · 2025-11-27
> **Additional Experimental Results: Further Comparison with State-of-the-Art Activation Steering Methods on Gemma2-9B-Instruct-With-Deeper-Safety-Alignment for the Jailbreaking Task**
>
> Dear Reviewer NQp6,
>
> We would like to thank the reviewer again for your thoughtful reviews and feedback.
>
> In Table 3 below, we further compare our PID steering with recent steering-vector generation methods on a better safety-aligned variant of Gemma-9B-IT, namely Gemma2-9B-Instruct-With-Deeper-Safety-Alignment [16]. Deeper safety alignment here trains the model to sustain refusal behavior beyond the first few tokens: instead of only shaping the opener, it conditions on partially harmful or misleading prefixes and optimizes the model to "recover" back to safe behavior later in the sequence. Practically, this extends safety pressure across positions so refusals remain stable under mild coercion, prefilling, or decoding variance, while preserving general-task utility. Under this strong safety-aligned regime, PID-based steering still achieves a non-trivial attack success rate and outperforms other state-of-the-art activation steering methods considered in our study.
>
> We have added these new results into Table 3 in Appendix C.2 of our revision.
>
> **Table 3**. Jailbreaking results on Gemma2-9B-Instruct-With-Deeper-Safety-Alignment.
>
> | Method     | ASR $\uparrow$        | tinyArc $\uparrow$  | tinyGSM8k strict $\uparrow$  | tinyMMLU $\uparrow$  | tinyTruthQA $\uparrow$  | tinyHellaSwag $\uparrow$  | tinyWinoGrande $\uparrow$  |
> | ---------- | ------------ | --------- | ------------------ | ---------- | ------------- | --------------- | ---------------- |
> | Original   | 1.22         | 69.07     | 82.93              | 76.18      | 54.82         | 82.27           | 72.11            |
> | DIM        | 11.91         | 67.88     | 79.76              | 72.12      | 51.83         | 81.33           | 71.47            |
> | ITI        | $\underline{23.12}$ | 68.19     | 81.41              | 75.22      | 53.05         | 81.58           | 71.79            |
> | RePE       | 11.23        | 67.37     | 78.96              | 70.92      | 51.06         | 81.08           | 71.09            |
> | PID (ours) | **34.75**    | 67.83     | 79.11              | 74.77      | 52.44         | 81.46           | 71.33            |
>
> We would be grateful if you could let us know whether our responses have satisfactorily addressed your concerns or if any questions remain regarding our submission or rebuttal.
>
> We would be happy to engage in any follow-up discussion or address any additional comments by the reviewer.
>
> If you feel our responses have fully addressed the concerns raised in your review, we would appreciate it if you could consider increasing your score to better reflect your current assessment. Thank you again for your time and thoughtful feedback.
>
> **References**
>
> [16] Xiangyu Qi et al. "Safety alignment should be made more than just a few tokens deep". ICLR, 2025.

---

### Author Response · Authors · 2025-11-19
**Summary of Revision**

Incorporating the comments and suggestions from all reviewers, besides fixing typos and notations, we have made the following main changes in the revised paper:

**1. New baselines and expanded comparisons:** We have added additional activation-steering baselines (ActADD [1], AURA [2], ITI-C [3], CAA [4]), as well as results using QVQ-72B-Preview evaluation scores, to the toxicity experiments (Table 1).

**2. Additional model-size experiment:** We have added a new experiment on Gemma2-27B-Instruct for the jailbreak task, reported in Table 2 (Section 5.2). Overall, ITI and RePE degrade strongly on large models, whereas PID scales better and achieves the highest ASR.

**3. New Ablation Studies on $K_p, K_i, K_d$:** We conducted a full ablation over $K_p, K_i, K_d$ for Jailbreaking on Gemma-2-9B-it and Toxicity mitigation on Gemma2-2B. The results are reported in Figure 6 of Section 5.4.

**4. Improved Explanation of Figure 3 (P vs PI vs PID):** We have added a clearer explanation for Figure 3, from line 249 to line 265.

**5. Analysis Beyond Local Linearization:** We added a new section in Appendix B.8 that analyzes how the higher-order nonlinear terms enter the error dynamics and discusses their impact on the closed-loop behaviour.

**6. Empirical Evidence for the Stability Ranges of $K_p, K_i, K_d$:** We have numerically computed the stability ranges of $K_p, K_i, K_d$ in our PID steering and reported the results in Appendix C.3 of our revised manuscript.

**7. Further Comparison with State-of-the-Art Activation Steering Methods under the Strong Safety-aligned Regime:** We have further compared our PID steering with state-of-the-art steering-vector generation methods on a better safety-aligned variant of Gemma-9B-IT, namely Gemma2-9B-Instruct-With-Deeper-Safety-Alignment [5], and included these new results in Table 3 in Appendix C.2 of our revision.

**References**

[1] Li et al. “Inference-Time Intervention: Eliciting Truthful Answers from a Language Model”, NeurIPS, 2024.

[2] Turner et al. “Steering Language Models with Activation Engineering”, OpenReview preprint, 2025.

[3] Suau et al. “Whispering Experts: Neural Interventions for Toxicity Mitigation in Language Models”, ICML, 2024.

[4] Rimsky et al. “Steering Llama 2 via Contrastive Activation Addition”, ACL, 2024.

[5] Xiangyu Qi et al. "Safety alignment should be made more than just a few tokens deep". ICLR, 2025.

---

### Author Response · Authors · 2025-11-20
**General Response (1)**

Dear AC and Reviewers,

Thanks for your thoughtful reviews and valuable comments, which have helped us improve the paper significantly. We are encouraged by the endorsements that: **(1)** the connection between activation steering and classical control theory via a PID feedback formulation is novel (Reviewer NQp6) and interesting (Reviewer hCNg); **(2)** the paper provides a rigorous (Reviewer A2rW), solid (Reviewer NQp6) mathematical framework that maps proportional, integral, and derivative terms to steering dynamics, including analysis on error dynamic modeling and PID controller discretization; and **(3)** the empirical evaluation is comprehensive, showing consistent improvements across models and tasks (Reviewers NQp6, hCNg) with a lightweight implementation that can be seamlessly integrated into various existing steering frameworks (Reviewer A2rW). We have updated our submission based on the reviewers' feedback, and **we have highlighted our revision in magenta**.

One of the main concerns from the reviewers is that tuning PID controller parameters $(K_p, K_i, K_d)$ is nontrivial and somewhat cumbersome, and it is unclear how these parameters generalize across model architectures or domains. Another concern is that experimental comparisons with more baseline methods are needed. We address these concerns here.

**Concern 1: Tuning PID controller parameters $(K_p, K_i, K_d)$**

To understand the impact of the controller parameters on steering performance, **we conducted a full ablation study of the gain parameters $K_p, K_i, K_d$** for the LLM jailbreaking task on Gemma-2-9B-it and the toxicity mitigation task on Gemma2-2B. We included the results in Figure 6 (Section 5.4) of the revised manuscript. The ablation heatmaps show a clear, consistent pattern across both the jailbreaking and the toxicity mitigation tasks. First, $K_p$ is the primary driver of effect strength, and larger values (e.g., $K_p = 1.5$) consistently improve performance. In contrast, changes in $K_i$ and $K_d$ produce only small and smooth variations. Small non-zero  $K_i$ and $K_d$ help reduce disturbances and stabilize the steering dynamics, especially when $K_p$ is large.


This behavior aligns with Theorems 1 and 2 in our manuscript, which show that the integral and derivative components dampen oscillations and smooth the update trajectory. Importantly, the ablation heatmaps demonstrate that PID-AcT is not hypersensitive to its gain parameters. Even relatively large values such as $K_i = 0.15$ or $K_d = 0.10$ do not destabilize the method, and the optimal region forms a broad plateau around $K_p = 1.5,\ K_i \in [0.03, 0.06],\ K_d = 0.01.$ Both tasks, Gemma-2-9B-it for the LLM jailbreaking task and Gemma-2-2B for the LLM toxicity mitigation task, exhibit flat performance landscapes rather than sharp peaks, indicating low sensitivity and a wide operating range.

In addition to the ablation study $K_p, K_i, K_d$, in our original manuscript (Section 4 and Appendix B.3-B.7), **we have derived the stability range for $K_p, K_i, K_d$**, which are the values of those terms such that the closed-loop averaged error dynamics are input-to-state stable (ISS) with respect to the input disturbance (see Eqn. 20 in Section 4.1). We have numerically computed these stability ranges and reported the results in Appendix C.3 of our revised manuscript.

**Concern 2: Experimental comparisons with more baseline methods**

We have added the results for ITI-C [1], ActAdd [2], AURA [3], and CAA [4], as well as results using QVQ-72B-Preview evaluation scores, to Table 1 in the revised manuscript. We also copied those results in Tables 1 and 2 below.

PID-AcT not only outperforms Mean-AcT [5] and Linear-AcT [5], but also exceeds stronger activation-editing baselines such as ActADD, CAA, AURA, and ITI-C. Whereas these baseline methods either plateau, raise perplexity sharply, or degrade accuracy, our PID-AcT delivers substantially larger reductions, up to **8.2x** on Gemma2-2B and **8.1x** on LLaMA3-8B, while keeping utility comparable. PID-AcT is the only method that achieves the best scores across both classifier-based and LLM-judge toxicity metrics, highlighting its robustness. This improvement stems from PID’s dynamic correction: the proportional term drives strong mitigation, while small integral and derivative terms stabilize the update, drive the steady-state error to 0, and avoid overshooting. Overall, PID-AcT provides the most stable and effective trade-off among all activation-intervention baselines.

---

> ### Author Response · Authors · 2025-11-20
> **General Response (2)**
>
> **Table 1**:  Toxicity mitigation results for Gemma-2B, averaged over 10 runs. Lower is better for toxicity and perplexity; higher is better for MMLU. Best and second-best exclude the original baseline. (-) means no steering, (&#10003;) means sequential steering, and (&#10007;) means no sequential steering.
>
> | Method    | Seq. | CLS Tox $\downarrow$    | 0-shot Tox $\downarrow$    | QVQ $\downarrow$    | PPL Wiki $\downarrow$    | PPL Mistral $\downarrow$    | MMLU $\uparrow$    |
> |-|-|-|-|-|-|-|-|
> | Original    | --    | $4.17 \scriptscriptstyle{\pm 0.32}$    | $13.42 \scriptscriptstyle{\pm 1.08}$    | $14.17 \scriptscriptstyle{\pm 0.08}$    | $13.98$    | $6.68$    | $53.10$    |
> | ActADD    | &#10007; | $3.96 \scriptscriptstyle{\pm 0.24}$    | $13.43 \scriptscriptstyle{\pm 1.42}$    | $14.17 \scriptscriptstyle{\pm 0.08}$    | $14.69 \scriptscriptstyle{\pm 0.22}$                   | $\mathbf{6.67 \scriptscriptstyle{\pm 0.15}}$    | $\mathbf{53.00 \scriptscriptstyle{\pm 0.51}}$    |
> | CAA    | &#10007; | $1.20 \scriptscriptstyle{\pm 0.25}$    | $5.35 \scriptscriptstyle{\pm 0.50}$    | $5.88 \scriptscriptstyle{\pm 0.36}$    | $\mathbf{14.60 \scriptscriptstyle{\pm 0.20}}$    | $6.85 \scriptscriptstyle{\pm 0.22}$    | 51.70 $\scriptscriptstyle{\pm 0.48}$    |
> | AURA    | &#10007; | $2.12 \scriptscriptstyle{\pm 0.27}$    | $9.04 \scriptscriptstyle{\pm 0.66}$    | $9.72 \scriptscriptstyle{\pm 0.27}$    | $\mathbf{14.18 \scriptscriptstyle{\pm 0.14}}$    | $7.04 \scriptscriptstyle{\pm 0.34}$    | $\mathbf{53.00 \scriptscriptstyle{\pm 0.30}}$    |
> | ITI-C    | &#10007; | $0.74 \scriptscriptstyle{\pm 0.18}$    | $5.36 \scriptscriptstyle{\pm 0.91}$    | $6.10 \scriptscriptstyle{\pm 0.13}$    | $14.90 \scriptscriptstyle{\pm 0.29}$    | $7.44 \scriptscriptstyle{\pm 0.19}$    | $52.60 \scriptscriptstyle{\pm 0.55}$    |
> | Mean-Act    | &#10007; | $1.12 \scriptscriptstyle{\pm 0.23}$    | $5.20 \scriptscriptstyle{\pm 0.42}$    | $5.80 \scriptscriptstyle{\pm 0.15}$    | $\underline{14.53 \scriptscriptstyle{\pm 0.21}}$    | $\underline{6.81 \scriptscriptstyle{\pm 0.19}}$    | $51.74 \scriptscriptstyle{\pm 0.55}$    |
> | Linear-Act  | &#10007; | $0.95 \scriptscriptstyle{\pm 0.36}$          | $5.37 \scriptscriptstyle{\pm 0.80}$               | $5.92 \scriptscriptstyle{\pm 0.11}$    | $14.75 \scriptscriptstyle{\pm 0.22}$                   | $7.24 \scriptscriptstyle{\pm 0.24}$                    | $51.63 \scriptscriptstyle{\pm 0.50}$                   |
> | Mean-Act    | &#10003; | $\underline{0.68 \scriptscriptstyle{\pm 0.21}}$ | $\underline{3.23 \scriptscriptstyle{\pm 0.44}}$ | $\underline{3.70 \scriptscriptstyle{\pm 0.14}}$    | $14.92 \scriptscriptstyle{\pm 0.25}$                   | $6.97 \scriptscriptstyle{\pm 0.74}$                    | $\underline{51.80 \scriptscriptstyle{\pm 0.55}}$       |
> | Linear-Act  | &#10003; | $1.00 \scriptscriptstyle{\pm 0.27}$          | $4.13 \scriptscriptstyle{\pm 0.89}$               | $4.64 \scriptscriptstyle{\pm 0.04}$    | $14.98 \scriptscriptstyle{\pm 0.22}$                   | $7.13 \scriptscriptstyle{\pm 0.70}$                    | $51.47 \scriptscriptstyle{\pm 0.50}$                   |
> | PID-Act (Ours) | &#10003; | $\mathbf{0.51 \scriptscriptstyle{\pm 0.21}}$ | $\mathbf{2.90 \scriptscriptstyle{\pm 0.55}}$   | $\mathbf{3.40 \scriptscriptstyle{\pm 0.04}}$    | $15.22 \scriptscriptstyle{\pm 0.24}$                   | $7.02 \scriptscriptstyle{\pm 0.65}$                    | $51.30 \scriptscriptstyle{\pm 0.52}$                   |

---

> ### Author Response · Authors · 2025-11-20
> **General Response (3)**
>
> **Table 2**: Toxicity mitigation results for Llama-8B, averaged over 10 runs. Lower is better for toxicity and perplexity; higher is better for MMLU. Best and second-best exclude the Original baseline. (–) means no steering, (✓) means sequential steering, and (✗) means no sequential steering.
>
> | Method           | Seq. | CLS Tox $\downarrow$                             | 0-shot Tox $\downarrow$                           | QVQ $\downarrow$                                       | PPL Wiki $\downarrow$                                  | PPL Mistral $\downarrow$                               | MMLU $\uparrow$                                        |
> |------------------|------|--------------------------------------------------|---------------------------------------------------|--------------------------------------------------------|--------------------------------------------------------|--------------------------------------------------------|---------------------------------------------------------|
> | Original         | --   | $5.80$                                           | $15.00$                                           | $15.81 \scriptscriptstyle{\pm 0.09}$                   | $9.06$                                                 | $5.68$                                                 | $65.30$                                                 |
> | ActADD           | ✗    | $5.57 \scriptscriptstyle{\pm 0.45}$              | $15.73 \scriptscriptstyle{\pm 0.21}$              | $16.48 \scriptscriptstyle{\pm 0.19}$                   | $9.71 \scriptscriptstyle{\pm 0.46}$                    | $5.85 \scriptscriptstyle{\pm 0.26}$                    | $\mathbf{65.50 \scriptscriptstyle{\pm 0.34}}$           |
> | CAA              | ✗    | $1.82 \scriptscriptstyle{\pm 0.36}$              | $6.70 \scriptscriptstyle{\pm 0.58}$               | $7.40 \scriptscriptstyle{\pm 0.06}$                    | $9.40 \scriptscriptstyle{\pm 0.25}$                    | $5.50 \scriptscriptstyle{\pm 0.30}$                    | $64.30 \scriptscriptstyle{\pm 0.37}$                    |
> | AURA             | ✗    | $1.90 \scriptscriptstyle{\pm 0.61}$              | $8.12 \scriptscriptstyle{\pm 0.85}$               | $8.80 \scriptscriptstyle{\pm 0.17}$                    | $9.52 \scriptscriptstyle{\pm 0.32}$                    | $6.05 \scriptscriptstyle{\pm 0.30}$                    | $\mathbf{65.50 \scriptscriptstyle{\pm 0.33}}$           |
> | ITI-C            | ✗    | $1.60 \scriptscriptstyle{\pm 0.22}$              | $6.53 \scriptscriptstyle{\pm 0.66}$               | $7.19 \scriptscriptstyle{\pm 0.06}$                    | $9.48 \scriptscriptstyle{\pm 0.24}$                    | $6.17 \scriptscriptstyle{\pm 0.14}$                    | $\underline{64.70 \scriptscriptstyle{\pm 0.44}}$        |
> | Mean-AcT         | ✗    | $1.78 \scriptscriptstyle{\pm 0.33}$              | $6.56 \scriptscriptstyle{\pm 0.54}$               | $7.30 \scriptscriptstyle{\pm 0.25}$                    | $\underline{9.36 \scriptscriptstyle{\pm 0.28}}$        | $\mathbf{5.45 \scriptscriptstyle{\pm 0.34}}$           | $64.35 \scriptscriptstyle{\pm 0.39}$                    |
> | Linear-AcT       | ✗    | $1.87 \scriptscriptstyle{\pm 0.39}$              | $6.55 \scriptscriptstyle{\pm 0.21}$               | $7.30 \scriptscriptstyle{\pm 0.15}$                    | $\mathbf{9.35 \scriptscriptstyle{\pm 0.17}}$           | $5.56 \scriptscriptstyle{\pm 0.33}$                    | $64.55 \scriptscriptstyle{\pm 0.33}$                    |
> | Mean-AcT         | ✓    | $\underline{1.21 \scriptscriptstyle{\pm 0.41}}$  | $\underline{5.09 \scriptscriptstyle{\pm 0.64}}$   | $\underline{5.73 \scriptscriptstyle{\pm 0.05}}$        | $9.83 \scriptscriptstyle{\pm 0.21}$                    | $5.71 \scriptscriptstyle{\pm 0.33}$                    | $64.22 \scriptscriptstyle{\pm 0.40}$                    |
> | Linear-AcT       | ✓    | $1.68 \scriptscriptstyle{\pm 0.48}$              | $6.47 \scriptscriptstyle{\pm 0.38}$               | $7.12 \scriptscriptstyle{\pm 0.26}$                    | $9.48 \scriptscriptstyle{\pm 0.19}$                    | $\underline{5.46 \scriptscriptstyle{\pm 0.44}}$        | $64.49 \scriptscriptstyle{\pm 0.38}$                    |
> | PID-AcT (Ours)   | ✓    | $\mathbf{0.72 \scriptscriptstyle{\pm 0.49}}$     | $\mathbf{4.36 \scriptscriptstyle{\pm 0.81}}$      | $\mathbf{4.90 \scriptscriptstyle{\pm 0.14}}$           | $9.56 \scriptscriptstyle{\pm 0.20}$                    | $6.08 \scriptscriptstyle{\pm 0.37}$                    | $64.50 \scriptscriptstyle{\pm 0.36}$                    |

---

> ### Author Response · Authors · 2025-11-21
> **General Response (4)**
>
> **References**
>
> [1] Li et al. “Inference-Time Intervention: Eliciting Truthful Answers from a Language Model”, NeurIPS, 2024.
>
> [2] Turner et al. “Steering Language Models with Activation Engineering”, OpenReview preprint, 2025.
>
> [3] Suau et al. “Whispering Experts: Neural Interventions for Toxicity Mitigation in Language Models”, ICML, 2024.
>
> [4] Rimsky et al. “Steering Llama 2 via Contrastive Activation Addition”, ACL, 2024.
>
> [5] Rodriguez et al. “Controlling Language and Diffusion Models by Transporting Activations”, ICLR, 2025.
>
> ---
>
> We are glad to answer any further questions you have on our submission.

---

### Author Response · Authors · 2025-11-24
**Finalized Rebuttal and Additional Results**

Dear Reviewers and Chairs,

Thank you once again for your thoughtful reviews, valuable feedback, and for the great effort the chairs have invested in coordinating the process and ensuring a fair and constructive discussion.

We have now finalized and posted our rebuttal. As promised, we have also included additional experimental results in our second-round response last week, and all planned results have now been incorporated into the rebuttal and mentioned in our "Summary of Revision".

We would greatly appreciate it if you could let us know whether our responses have adequately addressed your concerns, or if there are any remaining questions regarding our submission or rebuttal.

We are happy to provide any further clarification or engage in additional follow-up discussion as needed.

Best regards,

Authors

---

### Author Response · Authors · 2025-11-27
**Additional Experimental Results: Further Comparison with State-of-the-Art Activation Steering Methods on Gemma2-9B-Instruct-With-Deeper-Safety-Alignment for the Jailbreaking Task**

Dear Reviewers,

We would like to thank all reviewers again for your thoughtful reviews and feedback.

In Table 1 below, we further compare our PID steering with recent steering-vector generation methods on a better safety-aligned variant of Gemma-9B-IT, namely Gemma2-9B-Instruct-With-Deeper-Safety-Alignment [1]. Deeper safety alignment here trains the model to sustain refusal behavior beyond the first few tokens: instead of only shaping the opener, it conditions on partially harmful or misleading prefixes and optimizes the model to "recover" back to safe behavior later in the sequence. Practically, this extends safety pressure across positions so refusals remain stable under mild coercion, prefilling, or decoding variance, while preserving general-task utility. **Under this strong safety-aligned regime, PID-based steering still achieves a non-trivial attack success rate and outperforms other state-of-the-art activation steering methods** considered in our study.

We have added these new results into Table 3 in Appendix C.2 of our revision.

**Table 1**. Jailbreaking results on Gemma2-9B-Instruct-With-Deeper-Safety-Alignment.

| Method     | ASR $\uparrow$        | tinyArc $\uparrow$  | tinyGSM8k strict $\uparrow$  | tinyMMLU $\uparrow$  | tinyTruthQA $\uparrow$  | tinyHellaSwag $\uparrow$  | tinyWinoGrande $\uparrow$  |
| ---------- | ------------ | --------- | ------------------ | ---------- | ------------- | --------------- | ---------------- |
| Original   | 1.22         | 69.07     | 82.93              | 76.18      | 54.82         | 82.27           | 72.11            |
| DIM        | 11.91         | 67.88     | 79.76              | 72.12      | 51.83         | 81.33           | 71.47            |
| ITI        | $\underline{23.12}$ | 68.19     | 81.41              | 75.22      | 53.05         | 81.58           | 71.79            |
| RePE       | 11.23        | 67.37     | 78.96              | 70.92      | 51.06         | 81.08           | 71.09            |
| PID (ours) | **34.75**    | 67.83     | 79.11              | 74.77      | 52.44         | 81.46           | 71.33            |

We would be grateful if you could let us know whether our responses have satisfactorily addressed your concerns or if any questions remain regarding our submission or rebuttal.

We would be happy to engage in any follow-up discussion or address any additional comments by the reviewers.

**References**

[1] Xiangyu Qi et al. "Safety alignment should be made more than just a few tokens deep". ICLR, 2025.

---

### Author Response · Authors · 2025-11-30
**Briefing 1 for the New AC: Summary of Our Key Contributions**

# Main Contributions of Our Submission

**1. Control-Theoretic Formulation for Feature Direction**

We develop a control-theoretic formulation of activation steering by modeling the layer-wise construction of feature directions in LLMs as a dynamical system. Under this framework, we show that widely used steering methods relying on difference-of-means (ActAdd [1], Directional Ablation [2], Mean-AcT [3]) can be interpreted as proportional ($P$) controllers responding to an error signal induced by contrastive prompts. This analysis explains their steady-state bias: perturbations to activation dynamics cannot be fully corrected under a pure $P$-update.

**2. PID-Based Steering**

*We propose the novel Proportional-Integral-Derivative (PID) Steering, a control-theoretic framework for computing feature directions using a PID controller to reduce the steady-state error inherent in existing activation steering methods*.

 * The **$P$-term** pushes activations toward target semantic directions.
  * The **$I$-term** accumulates error across layers to remove steady-state bias.
  * The **$D$-term** reacts to rapid activation changes to mitigate overshoot and oscillations.

Our PID-Based Steering yields a *closed-loop* controller with interpretable error dynamics and classical input-to-state stability (ISS) guarantees.

**3. Rigorous Theoretical Results for Activation Steering**

Our control-theoretical formulation for feature direction in **1.** above allows to *theoretically analyze the PID Steering’s advantages in reducing steady-state error and oscillations*.

* We derive averaged error dynamics for $P$, $PI$, and $PID$ steering.
* We prove ISS-type stability under locally linearized activation dynamics. These results show how $P$-term reduces the error but leaves a bias, $I$-term cancels the bias at the cost of potential overshoot, and $D$-term damps that overshoot.
* We obtain explicit stability ranges for $(K_p,K_i,K_d)$ (Proposition 1, Proposition 3, Theorem 1).

The definitions, analysis, and proofs of the overshoot mechanism in activation steering are presented in detail in Appendix B.6 and Theorem 2 in Appendix B.7.

# Additional Advantages of Our PID Steering

We would also like to highlight here 2 convenient advantages of our PID Steering.

* **Lightweight, plug-and-play implementation.**
  PID Steering is used only in the *construction* of steering vectors/planes; these are then fed into existing frameworks such as Angular Steering[4], Mean-AcT, and CAA [5]. The steering vectors are computed offline, so inference latency and memory cost at test time are unchanged relative to the underlying steering method.
* **Broad empirical gains across models and modalities.**
  Across Gemma2, LLaMA3, and Qwen2.5 (toxicity mitigation, jailbreak resistance, style control), PID Steering consistently improves behavioral control while maintaining utility. Reviewers highlight that the empirical evaluation is comprehensive and that the framework is easily integrated into existing methods.

**References**

[1] Turner et al. “Steering Language Models with Activation Engineering”, arXiv preprint, 2025.

[2] Arditi et al. “Refusal in Language Models Is Mediated by a Single Direction”, NeurIPS, 2024.

[3] Rodriguez et al. “Controlling Language and Diffusion Models by Transporting Activations”, ICLR, 2025.

[4] Vu and Nguyen “Angular Steering: Behavior Control via Rotation in Activation Space”, NeurIPS, 2025.

[5] Rimsky et al. “Steering Llama 2 via Contrastive Activation Addition”, ACL, 2024.

---

### Author Response · Authors · 2025-11-30
**Briefing 2 for the New AC: Main Concerns of Reviewers and Our Replies (1)**

First, we are encouraged by the endorsements from the reviewers that: **(1)** the connection between activation steering and classical control theory via a PID feedback formulation is novel (Reviewer NQp6) and interesting (Reviewer hCNg); **(2)** the paper provides a rigorous (Reviewer A2rW), solid (Reviewer NQp6) mathematical framework that maps proportional, integral, and derivative terms to steering dynamics, including analysis on error dynamic modeling and PID controller discretization; and **(3)** the empirical evaluation is comprehensive, showing consistent improvements across models and tasks (Reviewers NQp6, hCNg) with a lightweight implementation that can be seamlessly integrated into various existing steering frameworks (Reviewer A2rW). We have updated our submission based on the reviewers' feedback, and **we have highlighted our revision in magenta**.

Second, we summarize the reviewers' main concerns and our replies below.

**(1) “Too few baselines / incomplete comparisons” (Reviewers NQp6, A2rW)**

* *Reviewer's Concern:* Initial experiments mainly compared against sequential Mean-AcT/Linear-AcT; reviewers asked for stronger steering baselines such as CAA, ITI, and ActAdd, and for more thorough comparisons with cited methods.
* *Our Response:* We added **ActADD [1], AURA [2], ITI-C [3], and CAA [4]** plus **QVQ-72B-Preview [5]** as an additional LLM-as-a-judge toxicity metric. Updated tables for Gemma-2B and LLaMA-8B show that PID-AcT achieves the strongest toxicity reduction (up to ~8.2× on Gemma2-2B and ~8.1× on LLaMA3-8B) while keeping perplexity/MMLU comparable, and is the only method that is best on both classifier-based and LLM-judge metrics. Please refer to Table 1, Section 5 in our revised manuscripts for further analysis.

**(2) “Self-evaluation bias in toxicity measurement” (NQp6)**

* *Reviewer's Concern:* Using LLaMA-3-8B both as generator and toxicity evaluator could bias results.
* *Our Response:* we clarified that the original setup already uses a **RoBERTa-based toxicity classifier** plus a 0-shot LLaMA-3 judge, and we further added **QVQ-72B-Preview** as an external evaluator. The improved scores of PID-AcT hold across all three metrics, alleviating concerns about self-evaluation. Please refer to Table 1, Section 5, in our revised manuscripts for further analysis.

**(3) “Sensitivity and manual tuning of PID gains” (Reviewers NQp6, A2rW, hCNg)**

* *Reviewer's Concern:* PID control is often sensitive to $(K_p,K_i,K_d)$; reviewers worried about heavy trial-and-error and whether gains generalize across tasks/models.
* *Our Response:*
  * We ran full ablations over $(K_p,K_i,K_d)$ for (i) jailbreaking on Gemma-2-9B-it and (ii) toxicity mitigation on Gemma2-2B. Heatmaps in Figure 6, Section 5.3 of the revised manuscript show:
    * $K_p$ is the main driver of strength; larger $K_p$ improves effect.
    * Small non-zero $K_i, K_d$ stabilize dynamics and reduce steady-state error/overshoot.
    * Performance landscapes are broad and flat rather than sharply peaked, indicating low sensitivity and a wide safe region.
  * We derived analytic stability ranges for $P$, $PI$, and $PID$ (Proposition 1, Proposition 3, Theorem 1), then numerically instantiated them, showing that empirically good gains fall well inside these regions (Figure 6, Section 5.4).
  * In a later exchange, we clarified that optimal gains are *not strictly universal*, but the near-optimal ranges found in the ablation are remarkably similar across two model families and two tasks, suggesting practical robustness.

**(4) “No analysis of independent P/I/D contributions” (Reviewer A2rW)**

* *Reviewer's Concern:* Since we claim a plug-in design, reviewers wanted explicit evidence for the separate and combined effects of $P$, $I$, $D$.
* *Our Response:* The new ablation evaluates **$P$-only, $PI$, $PD$, and full $PID$** variants across gain grids. Empirically,
  * $P$-only removes most errors but leaves a steady-state bias (see Proposition 1 in Section 3.1 and Appendix B.4),
  * adding $I$ reduces this bias and improves consistency, but can cause overshoot (see Proposition 3 in Section 4.2.1 and Appendix B.5, Proposition 4 in Appendix B.6)
  * $D$-term dampens overshoot from aggressive $PI$ settings (see Theorem 1 & 2 in Section 4.2.2 and Appendix B.7.2),
  * Full $PID$ consistently beats $P$-only, $PI$, and $PD$ on jailbreak and toxicity metrics (see Section 5), supporting the claim that the modules are complementary rather than redundant.

---

> ### Author Response · Authors · 2025-11-30
> **Briefing 2 for the New AC: Main Concerns of Reviewers and Our Replies (2)**
>
> **(5) “Computational overhead and practicality” (Reviewer NQp6)**
>
> * *Reviewer's Concern:* Potential extra latency/memory from closed-loop control on large LMs.
> * *Our Response:* We emphasized that all PID computations occur **offline when constructing steering vectors**; at inference time, downstream methods just apply these vectors exactly as in prior work, so there is **no additional runtime overhead** beyond the base steering method.
>
> **(6) “Validity of linearization and higher-order terms” (Reviewer NQp6)**
>
> * *Reviewer's Concern:* The ISS proofs rely on local linearization; reviewers worried about approximation error in deep nonlinear transformers, especially under strong steering.
> * *Our Response:*
>   * In Appendix B.8, we expanded our theoretical results, treating the Taylor remainder as a **bounded disturbance term** in the error dynamics. As long as higher-order terms remain bounded, the ISS guarantees still hold, with bounds that scale with the disturbance magnitude.
>   * We analyze the residual $\delta_i(k)$ between desired and steered activations, showing that under the derived gain conditions, closed-loop dynamics keep $\delta_i(k)$, the integral state, and the derivative term bounded; thus, higher-order effects stay controlled even with strong steering.
>
> **(7) “Do gains/generalization look reasonable in practice?” (Reviewer hCNg, post-discussion)**
>
> After seeing the new experiments, Reviewer hCNg commented that the added results “make our work more solid” and asked whether gains remain the same across models and how they were chosen. We clarified that:
>
>   * Gains are *not* theoretically universal.
>   * However, the **near-optimal band is consistent across the tested models/tasks**.
>   * The search space was **narrowed analytically** using the stability ranges before doing empirical sweeps.
>
>
> **References**
>
> [1] Turner et al. “Steering Language Models with Activation Engineering”, OpenReview preprint, 2025.
>
> [2] Suau et al. “Whispering Experts: Neural Interventions for Toxicity Mitigation in Language Models”, ICML, 2024.
>
> [3] Li et al. “Inference-Time Intervention: Eliciting Truthful Answers from a Language Model”, NeurIPS, 2024.
>
> [4] Rimsky et al. “Steering Llama 2 via Contrastive Activation Addition”, ACL, 2024.
>
> [5] Alibaba Qwen Team. Qvq: To see the world with wisdom, December 2024.

---

### Author Response · Authors · 2025-11-30
**Briefing 3 for the New AC: Summary of Additional Theoretical and Empirical Results Obtained during the Rebuttal & Discussion Period (1)**

# New Theoretical Results

**(1) Extended analysis beyond local linearization**

We first clarify why we linearize the activation dynamics around the desired activation trajectory:(i) this yields a tractable LTV model for tracking steered activations, (ii) our steering law is a **linear** PID controller, classically designed on linearised plants, and (iii) ISS/Lyapunov tools are cleanest in the linear time-varying setting, which lets us derive explicit stability ranges for $(K_p, K_i, K_d)$. Appendix B.8 then shows that keeping the full Taylor expansion simply augments the disturbance $w(k)$; as long as these terms are bounded (which they are under the stability conditions), the ISS results and qualitative conclusions remain valid. This strengthens the theoretical justification of PID Steering under nonlinear dynamics.

**(2) Empirical Evidence for the Stability Ranges of $K_p, K_i, K_d$**

We have numerically computed the stability ranges of $K_p, K_i$ in our PID steering. Since it is computationally expensive to compute the stability range of $K_d$, which involves solving a Lyapunov equation (line 1315) at every layer, we roughly estimate this stability range based on $K_p$ and then sweep through possible values of $K_d$ in that range, as in our ablation study in Figure 6 (Section 5.4) of the revised manuscript. We have included these results and discussion in Appendix C.3 of our revision and in our last reply to Reviewer hCNg.

Furthermore, in Figure 6 (Section 5.4), the measured performance over the gain grids matches the theoretically predicted stable regions: configurations outside the analytic bounds degrade performance or become unstable, whereas gains within the derived intervals yield smooth trajectories and stable control. This empirically validates the ISS-based gain design.

**(3) Clearer interpretation of P vs PI vs PID**

We expanded the explanation around Figure 3 to connect empirical trajectories with the theory: $P$ reduces error but leaves a residual offset; $PI$ cancels the offset but can overshoot; $PID$ trades off convergence speed, residual error, and overshoot in a controllable way.

# New Empirical Results

**(1) Full PID gain ablations on two tasks**

  We performed a comprehensive ablations over 3 hyper-parameters $(K_p,K_i,K_d)$ for

  * jailbreak on **Gemma-2-9B-it**, and
  * toxicity mitigation on **Gemma2-2B**.

  The heatmaps in Figure 6, Section 5.4, show a broad plateau of strong performance, indicating that PID Steering is robust rather than finely tuned. They consistently reveal that

  * $K_p$ primarily controls the overall steering strength;
  * small non-zero $K_i$ and $K_d$ damp oscillations, reduce steady-state error, and curb overshoot;
  * performance varies smoothly across a wide region of gains instead of peaking at a single narrow optimum.

**(2) Expanded toxicity baselines on Gemma-2B and LLaMA-8B**

In Table 1, Section 5 in the revised manuscript, we expanded the toxicity experiments on **Gemma-2B** and **LLaMA-8B** to include stronger activation-steering baselines (ActADD [1], AURA [2], ITI-C [3], CAA [4], Mean-AcT [5], Linear-AcT [5]) and to evaluate with both classifier-based metrics and the QVQ-72B LLM-as-a-judge scores. Across all three toxicity measures (CLS, 0-shot, QVQ) and on both models, **PID-AcT** achieves the lowest toxicity while keeping perplexity and MMLU comparable to or better than competing methods. As a result, PID-AcT is the only method that is consistently strongest across *all* toxicity metrics, rather than excelling on just a single one.


**(3) Larger-model jailbreak experiment (Gemma2-27B-Instruct)**

We added a new experiment on a substantially larger model (Table 2 Section 5.2, and Table 3, Appendix C.3), Gemma2-27B-Instruct. As model size grows, **ITI and RePE [6] degrade strongly**, while PID continues to achieve the highest attack-success rate, indicating that PID Steering scales more favorably with model capacity.

**(4) Deeper safety-aligned Gemma2-9B with stronger defenses**

We evaluated PID-based jailbreaking on **Gemma2-9B-Instruct-With-Deeper-Safety-Alignment** (Table 3, Appendix C.3), which is explicitly trained to maintain refusals deeper into the sequence. Even under this stronger safety regime, PID Steering:

  * Attains a non-trivial ASR (~35% vs. 1.2% for the original model),
  * **Outperforms other steering-vector generators (DIM, ITI, RePE)** on attack success while maintaining competitive performance on tinyArc, tinyGSM8K, tinyMMLU, tinyTruthQA, tinyHellaSwag, and tinyWinoGrande.

**(5) Cross-evaluator robustness for toxicity.**

The updated Table 1 in Section 5 includes **QVQ-72B-Preview** as a third toxicity evaluator, alongside the RoBERTa classifier and the LLaMA-3-based judge. PID-AcT remains the best method under all evaluators, demonstrating that improvements are not an artifact of any single classifier or model family.

---

> ### Author Response · Authors · 2025-11-30
> **Briefing 3 for the New AC: Summary of Additional Theoretical and Empirical Results Obtained during the Rebuttal & Discussion Period (2)**
>
> **References**
>
> [1] Turner et al. “Steering Language Models with Activation Engineering”, OpenReview preprint, 2025.
>
> [2] Suau et al. “Whispering Experts: Neural Interventions for Toxicity Mitigation in Language Models”, ICML, 2024.
>
> [3] Li et al. “Inference-Time Intervention: Eliciting Truthful Answers from a Language Model”, NeurIPS, 2024.
>
> [4] Rimsky et al. “Steering Llama 2 via Contrastive Activation Addition”, ACL, 2024.
>
> [5] Rodriguez et al. “Controlling Language and Diffusion Models by Transporting Activations”, ICLR, 2025.
>
> [6] Zou et al. "Representation Engineering: A Top-Down Approach to AI Transparency", Arxiv, 2023.

---

### Author Response · Authors · 2025-11-30
**Submission Briefing for the New AC: Thank You for Taking on the Oversight of Our Submission**

Dear new AC,

Thank you very much for stepping in during this difficult moment in the review process and taking over the handling of our submission. We sincerely appreciate your support.

To make it easier for you to review the paper, we have prepared three brief summaries, provided in the messages below:

**(Briefing 3) Summary of Additional Theoretical and Empirical Results Obtained during the Rebuttal & Discussion Period**

**(Briefing 2) Summary of Main Concerns of Reviewers and Our Replies**

**(Briefing 1) Summary of Our Key Contributions**

We submitted our initial rebuttal on November 18 (AoE), followed by additional experimental results on November 22 (AoE). Despite several reminders, only Reviewer hCNg engaged with us during the discussion phase. We appreciated his/her feedback, especially the remark “I believe these new experiments make your work more solid.” However, since the discussion with reviewers ended early, we were unable to continue the discussion with Reviewer hCNg or receive further feedback from the other reviewers.


We would be grateful if the AC could carefully read our paper, rebuttal, and further replies with additional results, and take into account that we have addressed all of the reviewers' concerns. We believe the reviewers would have updated their evaluations had the discussion continued.

We trust that, with your careful review, our submission will be assessed fairly and accurately by the AC, SAC, and PC. Please let us know if you have any questions regarding our submission and rebuttal. We are more than happy to provide clarification or engage in further discussion.

Best regards,

Authors

---

### Meta-Review · Area_Chair_Rhzh · 2026-01-10

**Summary:**

This paper reframes activation steering as a feedback control problem. Theoretically proves that most existing methods implicitly act as proportional (P) controllers, which leads to steady-state error and instability across layers.
Proposes PID-based activation steering, achieving more stable and effective control in LLM.

**Reviewer Concerns:**

Reviews request baseline comparisons, ablation studies on the control parameters K. Authors provide extensive experimental evaluations on the review response. I believe that most of concerns are addressed.

**Reviewer Scores:**

Review NQp6 would raise the score since the most of questions requested additional information which is provided by authors.

---

### Decision · Program_Chairs · 2026-01-26

Accept (Poster)